# ARTICLES

# Viruses affect picocyanobacterial abundance and biogeography in the North Pacific Ocean

Michael. C. G. Carlson[1], François Ribalet[2], Ilia Maidanik[1], Bryndan P. Durham[2,4], Yotam Hulata[1], Sara Ferrón[3], Julia Weissenbach[1], Nitzan Shamir[1], Svetlana Goldin[1], Nava Baran[1], B. B. Cael[3,5], David M. Karl[3], Angelicque E. White[3], E. Virginia Armbrust[2] and Debbie Lindell[1]✉

The photosynthetic picocyanobacteria *Prochlorococcus* and *Synechococcus* are models for dissecting how ecological niches are defined by environmental conditions, but how interactions with bacteriophages affect picocyanobacterial biogeography in open ocean biomes has rarely been assessed. We applied single-virus and single-cell infection approaches to quantify cyanophage abundance and infected picocyanobacteria in 87 surface water samples from five transects that traversed approximately 2,200 km in the North Pacific Ocean on three cruises, with a duration of 2–4 weeks, between 2015 and 2017. We detected a 550-km-wide hotspot of cyanophages and virus-infected picocyanobacteria in the transition zone between the North Pacific Subtropical and Subpolar gyres that was present in each transect. Notably, the hotspot occurred at a consistent temperature and displayed distinct cyanophage-lineage composition on all transects. On two of these transects, the levels of infection in the hotspot were estimated to be sufficient to substantially limit the geographical range of *Prochlorococcus*. Coincident with the detection of high levels of virally infected picocyanobacteria, we measured an increase of 10–100-fold in the *Synechococcus* populations in samples that are usually dominated by *Prochlorococcus*. We developed a multiple regression model of cyanophages, temperature and chlorophyll concentrations that inferred that the hotspot extended across the North Pacific Ocean, creating a biological boundary between gyres, with the potential to release organic matter comparable to that of the sevenfold-larger North Pacific Subtropical Gyre. Our results highlight the probable impact of viruses on large-scale phytoplankton biogeography and biogeochemistry in distinct regions of the oceans.

Shifts in microbial distributions occur along environmental gradients in the ocean, with changes in light, temperature and nutrient availability altering growth rates[1–5]. Viruses are the most abundant mortality agents on Earth and further shape microbial abundances, diversity and evolution[6–8]. The density and metabolic status of host cells impact infection by these obligate parasites, as viruses rely on the intracellular resources and cellular machinery of the host for their replication[9,10]. Thus, varying environmental conditions may shape virus populations and the degree to which they control host abundances and influence ecosystem functioning.

Marine picocyanobacteria of the genera *Prochlorococcus* and *Synechococcus* are the most abundant phototrophs on the planet and contribute a major fraction of global primary production[11]. Abiotic factors, particularly temperature, light and nutrient availability, are widely considered to control their geographical distribution[12–16]. Mortality factors, such as grazing and viral infection, are regarded as important regulators of picocyanobacterial abundance and diversity[8,17] but are seldom considered as factors impacting cyanobacterial biogeography.

Picocyanobacteria are infected by several lineages of viruses belonging to the order *Caudovirales*[18–20]. Each lineage has distinct traits with infection characteristics that lie on a spectrum of virulence and host range, and differ in genome size, the core replication and morphogenesis genes they encode as well as the number and type of genes captured from their hosts[10,18,19,21–24]. Such traits affect their fitness and presumably influence their abundance under different environmental conditions[9]. Although viral diversity continues to be extensively characterized via global-scale genomic surveys[25,26], the actual abundance and extent of infection by distinct virus lineages is largely unknown.

## Results and discussion

To explore how environmental gradients shape the distribution of cyanophages and picocyanobacteria, we conducted high-resolution surveys in surface waters along five oceanic transects on three cruises covering thousands of kilometres in the North Pacific Ocean in the spring or early summer of 2015, 2016 and 2017 (Fig. 1a–c). These cruises, two of which were out-and-back, passed through distinct regimes from warm, saline and nutrient-poor waters of the North Pacific Subtropical Gyre to cooler, less saline and nutrient-rich waters of higher latitudes influenced by the subpolar gyre (Fig. 1d–i)[27]. The shift between the two gyres was marked by abrupt changes in trophic indicators such as particulate carbon concentrations (Fig. 1g) and a chlorophyll front (defined as the 0.2 mg m⁻³ chlorophyll contour[28]; Fig. 1a–c). As such, the inter-gyre transition zone, defined by salinity and temperature thresholds[29] (Fig. 1d), was distinct from both the subtropical and subpolar gyre ecosystems[28].

**Unexpected *Prochlorococcus* decline.** *Prochlorococcus* concentrations in the oligotrophic waters of the subtropical gyre were $1.5$–$3.0 \times 10^5$ cells ml⁻¹, comprising an average of approximately 29% of the total bacteria (Extended Data Fig. 1) and numerically dominating the phytoplankton community in all three cruises (Extended Data Fig. 2). *Prochlorococcus* abundance remained high in the

[1]Faculty of Biology, Technion–Israel Institute of Technology, Haifa, Israel. [2]School of Oceanography, University of Washington, Seattle, WA, USA. [3]Department of Oceanography, University of Hawaiʻi at Mānoa, Honolulu, HI, USA. [4]Present address: Department of Biology, Genetics Institute, University of Florida, Gainesville, FL, USA. [5]Present address: National Oceanography Centre, European Way, Southampton, UK. ✉e-mail: dlindell@technion.ac.il

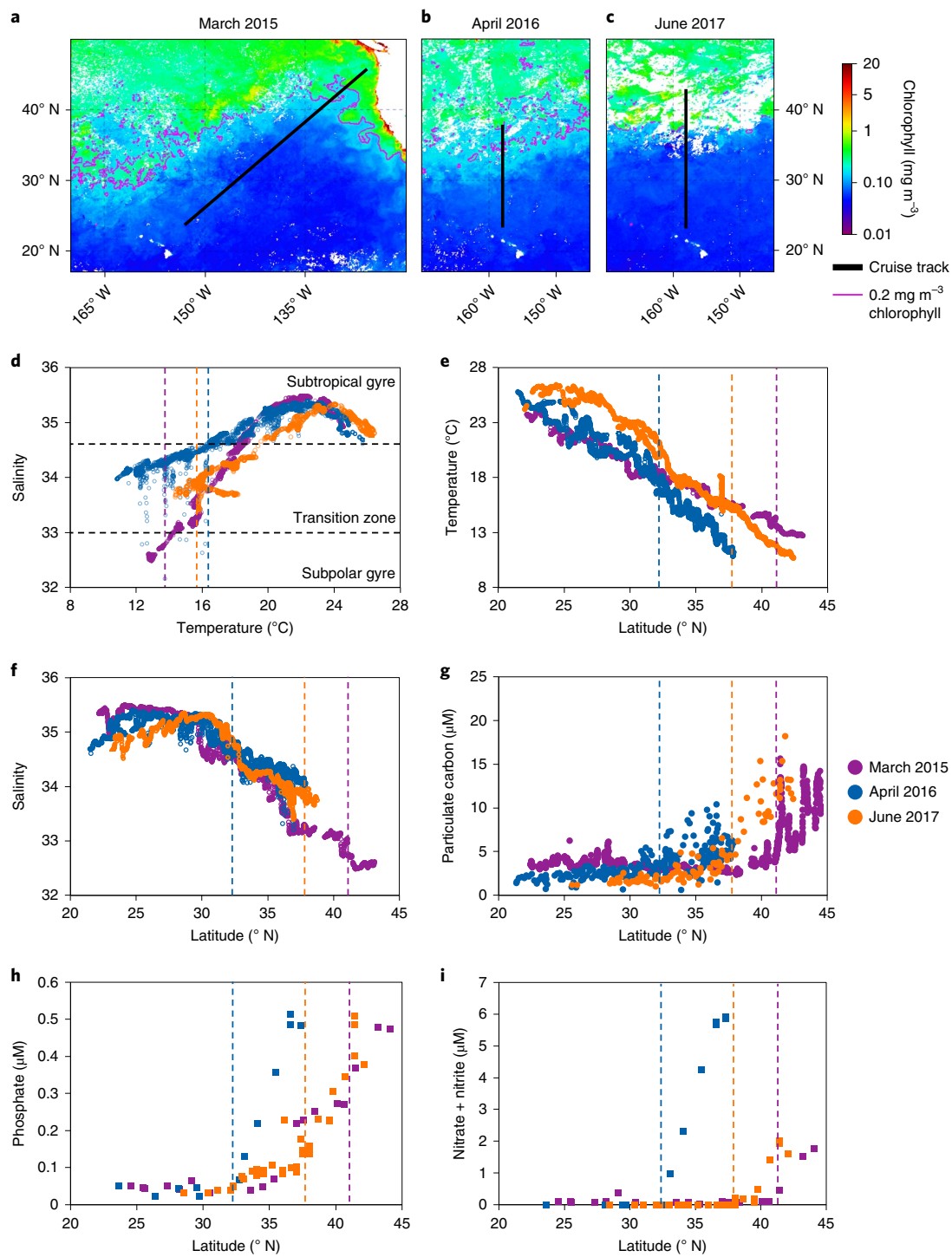

**Fig. 1 | Gradients in environmental conditions across the North Pacific gyres. a–c**, Transects of three cruises overlaid on monthly averaged satellite-derived sea-surface chlorophyll in March 2015 (**a**), April 2016 (**b**) and June 2017 (**c**). **d**, Temperature–salinity diagram showing the boundaries of the subtropical and subpolar gyres (black dashed lines) based on the salinity thresholds reported by Roden[29]. **e–i**, Temperature (**e**), salinity (**f**) as well as the levels of particulate carbon (**g**), phosphate (**h**) and nitrate + nitrite (**i**) as a function of latitude. The coloured dashed lines show the position of the 0.2 mg m$^{-3}$ chlorophyll contour. For environmental variables plotted against temperature, see Supplementary Fig. 3.

southern region of the transition zone in 2015 and 2016, decreasing precipitously to less than 2,000 cells ml$^{-1}$ north of the chlorophyll front, generally constituting <1% of the total bacteria (Fig. 2 and Extended Data Figs. 1b,f and 2). This decline occurred at temperatures of about 12 °C (Fig. 2) and is consistent with the thermal limits on *Prochlorococcus* growth determined for cultures and in numerous

field observations[11,12,15,30] (see Supplementary Discussion). Conversely, *Synechococcus* was 10–100-fold more abundant in the transition zone relative to the subtropics and gradually decreased northwards towards the subpolar waters (Fig. 2 and Extended Data Fig. 2).

The picocyanobacterial abundance patterns differed dramatically in June 2017. The decline in *Prochlorococcus* occurred

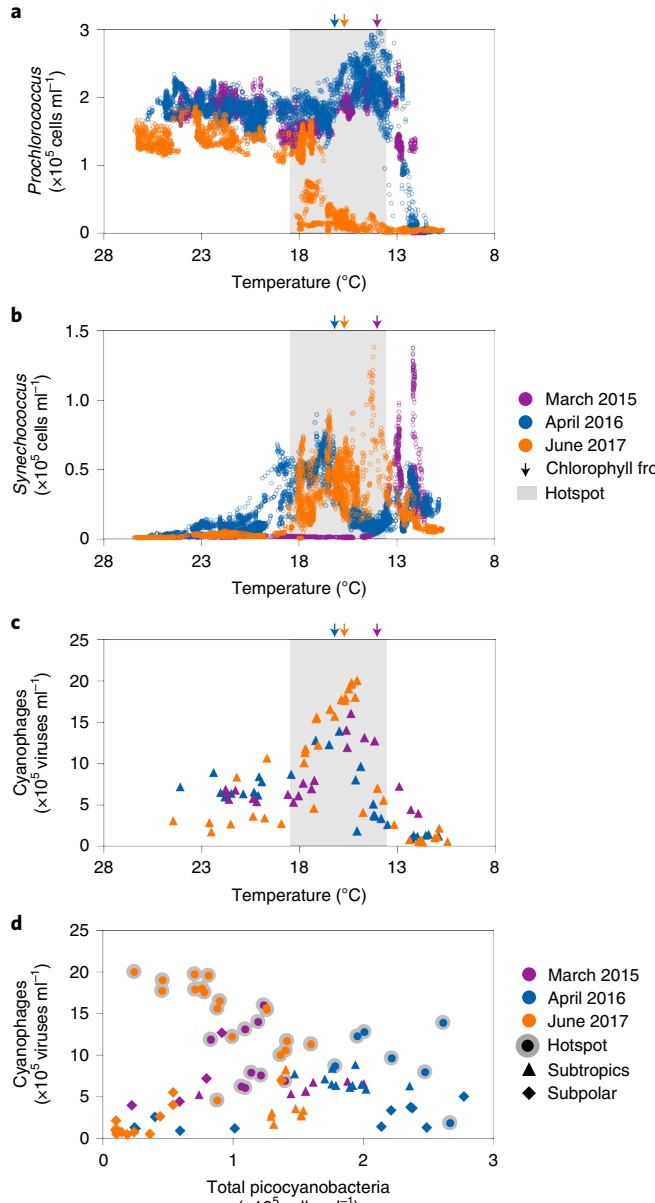

**Fig. 2 | Shifts in the distributions of *Prochlorococcus*, *Synechococcus* and cyanophages in the North Pacific Ocean. a–c,** The distributions of *Prochlorococcus* (**a**), *Synechococcus* (**b**) and cyanophages (**c**) were measured at high resolution in the surface waters along the transects of the three cruises in this study and plotted as a function of temperature. **d,** The relationship between the total numbers of picocyanobacteria and cyanophages. There was no relationship across the data from all regimes (Pearson's $r = -0.008$, two-sided $P = 0.9$, $n = 87$). Picocyanobacteria correlated positively with cyanophages in the subtropics (Pearson's $r = 0.54$, two-sided $P = 0.02$, $n = 26$). In the hotspot, the picocyanobacteria abundance correlated negatively with that of cyanophages across all three cruises (Pearson's $r = -0.56$, two-sided $P = 0.0005$, $n = 34$). There was no relationship found in the subpolar region (Pearson's $r = 0.2$, two-sided $P = 0.2$, $n = 27$).

approximately two latitudinal degrees south (about 220 km) of the chlorophyll front (Extended Data Figs. 1b,f and 2), where the water temperatures were nearly 18 °C (Fig. 2a). This implicated factors other than temperature as responsible for substantially restricting the geographical distribution of *Prochlorococcus*. In contrast,

the geographical range of *Synechococcus* was broader by approximately three degrees of latitude (about 330 km), and their integrated abundances across the transects were twofold higher than those observed in 2015 and 2016 (Fig. 2b and Extended Data Fig. 2). Thus, a more southern decline in *Prochlorococcus* and a broader distribution of *Synechococcus* was observed in 2017 relative to the 2015 and 2016 transects (Fig. 2 and Extended Data Fig. 2) as well as nine previous transects across this transition zone conducted in different years and seasons[31–33] (Supplementary Discussion).

Total bacterial abundances were stable in the subtropics and increased 1.4–3.2-fold in the transition zone on all three cruises (Extended Data Fig. 1a,e). This increase occurred north of the chlorophyll front in 2015 and 2016, whereas abundances increased south of this feature in June 2017. Thus, the 2017 increase in total bacteria happened despite the anomalous loss of *Prochlorococcus*, which made up only 5% of the total bacteria south of the chlorophyll front relative to 20–30% in the equivalent region in 2015 and 2016 (Extended Data Fig. 1b,f).

A suite of abiotic variables beyond temperature—many of which are considered important determinants for the biogeography of *Prochlorococcus*[11–16,34]—were assessed for their potential role in restricting the geographical distribution of *Prochlorococcus* on the 2017 transects (Extended Data Fig. 3). *Prochlorococcus* populations in 2017 were low compared with previous observations at similar macronutrient levels (phosphate and nitrate + nitrite; Extended Data Fig. 3a,c)[11]. Micronutrient (iron[27]) concentrations were within the range for optimal *Prochlorococcus* growth[35] and lead concentrations[27] were below levels toxic to *Prochlorococcus*[36]. *Prochlorococcus* populations in 2017 declined despite similar salinity (Extended Data Fig. 3e) and particulate carbon values (Extended Data Fig. 3g) to the 2016 transect; furthermore, *Prochlorococcus* populations thrived across a wider range of values for these variables in 2015. *Prochlorococcus* maintained high abundances across a range of mixed-layer depths of up to 100 m in 2015 and 2016 but declined at unusually shallow mixing depths in 2017 (Extended Data Fig. 3i). Collectively, the abiotic conditions observed in 2017 in the region of the *Prochlorococcus* decline supported large populations of *Prochlorococcus* in 2015 and 2016 (Extended Data Fig. 3). Thus, none of the physical or chemical factors investigated here can alone explain the unexpected decline in *Prochlorococcus* in 2017. However, we cannot rule out that a unique combination of these factors, or additional abiotic factors, led to the decline in *Prochlorococcus*.

**A virus hotspot affects picocyanobacterial distributions.** The lack of an identifiable abiotic variable differentiating the 2017 transects from the other transects and the overall high abundances in total bacteria for all transects (Extended Data Fig. 1a,e) led us to hypothesize that a mortality factor specific to picocyanobacteria, such as infection by viruses, played a role in precipitating the observed shifts in picocyanobacterial geographical ranges. This was investigated through quantification of the abundances of cyanophages and the extent to which they infected *Prochlorococcus* and *Synechococcus* using single-virus[37,38] and single-cell-infection[39] polony methods in surface waters of the three different regimes across the cruise transects—that is, the subtropical and subpolar gyres as well as the transition zone between them. We targeted the T7-like clade A and clade B cyanopodoviruses and the T4-like cyanomyoviruses— three major cyanophage lineages based on isolation studies[18–20,24,40], single-cell genomics[41] and global metagenomic surveys[25,26,42–44] (Supplementary Discussion)—as well as a more recently discovered group, the TIM5-like cyanomyoviruses[42,44]. Cyanophages from other lineages that are less common in metagenomic surveys[39,42,43] were not investigated.

In the subtropical gyre, cyanophages were major components of the planktonic virus community, averaging $5.7 \pm 1.9 \times 10^5$ cyanophages ml$^{-1}$ and ranging between 0.7 and 21% of the total

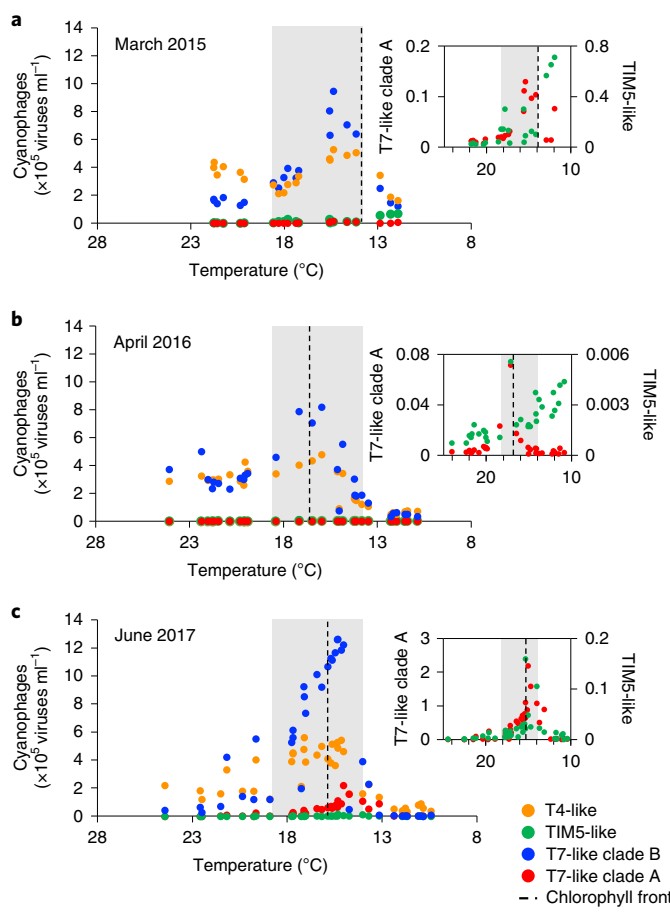

**Fig. 3 | Cyanophage community composition across the North Pacific gyres. a–c,** Cyanophage abundance for the March 2015 (**a**), April 2016 (**b**) and June 2017 (**c**) transects. Insets: T7-like clade A and TIM5-like cyanophage abundances on an expanded scale (similar to the main images, the units for the vertical axes are ×10⁵ viruses ml⁻¹). The grey shaded regions show the position of the virus hotspot. See Extended Data Fig. 4 for the confidence intervals and out-and-back reproducibility and Supplementary Fig. 4 for cyanophage lineages plotted against latitude.

double-stranded DNA viruses (Fig. 2c and Extended Data Fig. 1d,h, 2), consistent with earlier observations for this region[38,39]. T4-like cyanophages dominated the subtropical cyanophage community and were generally twofold more abundant than the T7-like clade B cyanophages, the second-most abundant group (Fig. 3 and Extended Data Fig. 4). Together, these two clades constituted >80% of cyanophages measured, with the remainder consisting of T7-like clade A and TIM5-like cyanophages (Fig. 3 and Extended Data Fig. 4). Cyanophage abundances correlated positively with total picocyanobacteria in the subtropical gyre (Pearson's coefficient of multiple correlation ($r$) = 0.54, $P$ = 0.02, $n$ = 26; Fig. 2d), suggesting that cyanophages were limited by the availability of susceptible hosts in this region and were not regulating picocyanobacterial populations. On average, less than 1% of the cyanobacterial populations were infected (Fig. 4), with higher infection rates by T4-like cyanophages than T7-like cyanophages (Extended Data Figs. 5 and 6). These instantaneous measurements of infection were used to estimate the daily rates of mortality[39] (Methods and Supplementary Discussion), which suggests that 0.5–6% of picocyanobacterial populations were lysed by viruses each day (Extended Data Fig. 7). This implicates other factors, such as grazing[45], as the major causes of cyanobacterial mortality in the North Pacific Subtropical Gyre.

Within the transition zone we observed a steep latitudinal increase in the abundance of cyanophages for every transect, which we define as a cyanophage hotspot (Fig. 2c and Extended Data Figs. 2 and 4). The cyanophage abundances in this hotspot were between three- and tenfold greater than in the subtropical gyre (Fig. 2c). Notably, cyanophages were approximately 25% more abundant (an increase of approximately 5×10⁵ viruses ml⁻¹) in the hotspot on the 2017 cruise relative to the other two cruises, reaching a maximum of 2×10⁶ viruses ml⁻¹. The hotspot peaked at temperatures of 15–16 °C on all transects, regardless of the geographical location, season or the exact pattern of the *Prochlorococcus* and *Synechococcus* distributions (Fig. 2c). Notably, the numbers of T7-like clade B cyanophages increased sharply in the transition zone to become the most abundant lineage, whereas T4-like cyanophages increased more modestly (Fig. 3 and Extended Data Fig. 4). The change in the cyanophage community structure was particularly pronounced in June 2017, when T7-like cyanophages were up to 2.3-fold more abundant than T4-like cyanophages (Fig. 3c). The switch in the relative abundance of T4-like and T7-like clade B cyanophages was diagnostic of the cyanophage hotspot compared with patterns in the subtropical and subpolar gyres.

To begin assessing whether cyanophages negatively affected cyanobacterial populations in the hotspot, we tested the relationship between the abundance of cyanophages and total cyanobacteria. This showed a significant negative correlation between cyanophage and cyanobacterial abundances across all three cruises (Pearson's $r$ = −0.56, two-sided $P$ = 0.0005, $n$ = 34). This relationship was particularly distinct in 2017, when cyanobacteria were at their overall lowest abundances and cyanophages at their highest (Pearson's $r$ = −0.65, two-sided $P$ = 0.004, $n$ = 18). This suggests that viruses are one of the key regulators of picocyanobacteria in the region of the hotspot. However, no significant correlation was found across all regimes and all years (Pearson's $r$ = −0.008, two-sided $P$ = 0.9, $n$ = 87; Fig. 2d), indicating that factors other than viruses are likely to be more important in regulating the abundances of cyanobacteria in other regimes.

Our single-cell infection measurements allowed us to directly evaluate active viral infection and its impact on picocyanobacteria in the transition zone. Viral infection spiked in this region each year with infection levels that were an average of two- to ninefold higher than those in the subtropical gyre (Fig. 4 and Extended Data Figs. 5,6 and 8). Infection peaked within the temperature range of 12–18 °C and was associated with a concomitant dip in *Prochlorococcus* abundances in all three cruises (Fig. 4 and Extended Data Fig. 5). These findings provide independent support for the strong negative correlation between cell and virus abundances (Fig. 2d) being the result of virus-induced mortality.

Lineage-specific infection was also distinct in the transition zone relative to the subtropical gyre. Infection by T7-like clade B cyanophages generally increased to reach (2015 and 2016) or exceed (2017) those of T4-like cyanophages (Extended Data Figs. 5 and 6). In addition, the ratio of the abundances of T7-like clade B cyanophages to the number of cells they infected was 2.6-fold greater in the hotspot than the subtropics, whereas this ratio was similar in both regions for T4-like cyanophages. Together, these results indicate that, within the hotspot, the T4-like cyanophages displayed increased levels of infection, whereas the T7-like cyanophages displayed both increased levels of infection and produced more viruses per infection, suggesting that T7-like clade B cyanophages are better adapted to conditions in the transition zone (see below).

Of the three cruises, the highest levels of viral infection were observed in June 2017, with up to 9.5% and 8.9% of *Prochlorococcus* and *Synechococcus* infected, respectively (Fig. 4e,f). This dramatic increase in infection mirrored the massive decline in *Prochlorococcus* abundances (Fig. 4e and Extended Data Fig. 5i). We estimate that viruses killed 10–30% of *Prochlorococcus* and *Synechococcus* cells

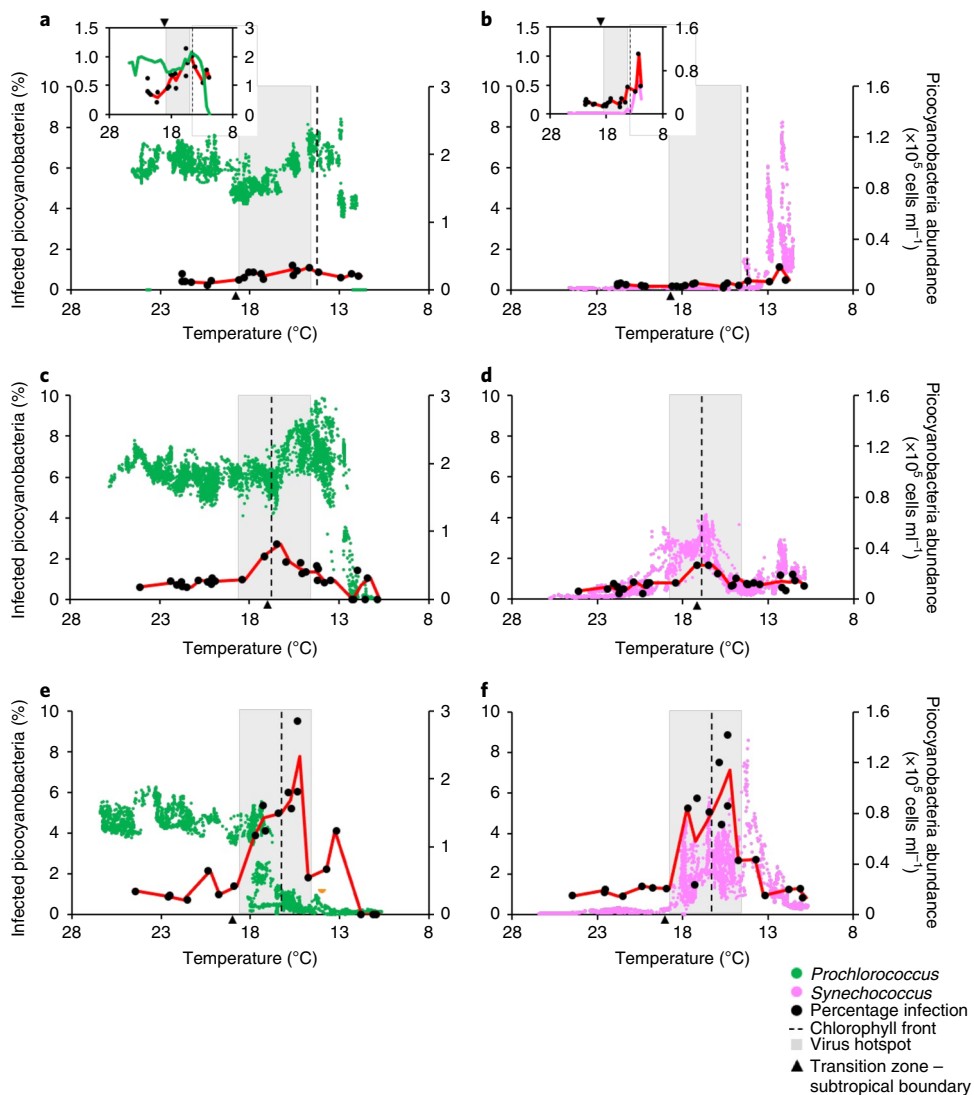

**Fig. 4 | Viral infection patterns of picocyanobacteria in the North Pacific Ocean. a–f**, Viral infection levels (black) of *Prochlorococcus* (**a,c,e**) and *Synechococcus* (**b,d,f**) plotted against temperature for the March 2015 (**a,b**), April 2016 (**c,d**) and June 2017 (**e,f**) transects. Insets: infection levels on an expanded scale. The solid lines show infection (red), *Prochlorococcus* (green) and *Synechococcus* (pink) averaged and plotted for every 0.5 °C. The dashed lines and shaded regions show the position of the chlorophyll front and the virus hotspot, respectively. For plots by latitude and the upper and lower bounds of infection, see Extended Data Figs. 5 and 6.

daily at these high instantaneous levels of infection (Extended Data Fig. 6) based on the expected number of infection cycles cyanophages were able to complete at the light and temperature conditions in the transition zone (Methods and Supplementary Discussion). Given that *Prochlorococcus* is estimated to double every $2.8 \pm 0.8$ d at the low temperatures in this region[12], we estimate that 21–51% of the population was infected and killed in the interval before cell division. *Synechococcus* is expected to have faster growth rates at these temperatures, doubling every $1.1 \pm 0.2$ d (refs. [12,46]). Thus, we estimate that less of the *Synechococcus* population (9–31%) was killed before division.

Under quasi-steady state conditions, abiotic controls on the growth rate of *Prochlorococcus* are balanced by mortality due to viral lysis, grazing and other mortality agents[39,45,47]. Based on the high levels of virus-mediated mortality, the parallel pattern between *Prochlorococcus'* death and viral infection, and the negative correlation between cyanophage and picocyanobacterial abundances in the transition zone, we propose that enhanced viral infection in 2017 disrupted this balance, leading to the unexpected decline in

*Prochlorococcus* populations. Grazing and other mortality agents not investigated here could also have contributed to additional mortality beyond the steady state, resulting in further losses of *Prochlorococcus*. In contrast to *Prochlorococcus*, *Synechococcus* maintained large populations despite high levels of infection (Fig. 4f), presumably due to their faster growth rates enabling them to maintain a positive net growth despite enhanced mortality. These findings suggest that virus-mediated mortality in 2017 was an important factor in limiting the geographic range of *Prochlorococcus* that resulted in a massive loss of habitat of approximately 550 km.

Cyanophage abundances and infection levels dropped sharply in the higher-latitude waters north of the hotspot (Figs. 2c, 4 and Extended Data Figs. 1d,h and 2). The abundances of both T7-like clade B and T4-like cyanophages declined precipitously, yet T4-like cyanophages were the dominant cyanophage lineage (Fig. 3). T7-like clade A cyanophages generally increased locally at the northern border of the hotspot and became the dominant T7-like lineage in two samples between 38 and 39.2° N in 2017 (Fig. 3c and Extended Data Fig. 4). In contrast to all other cyanophages, the abundances

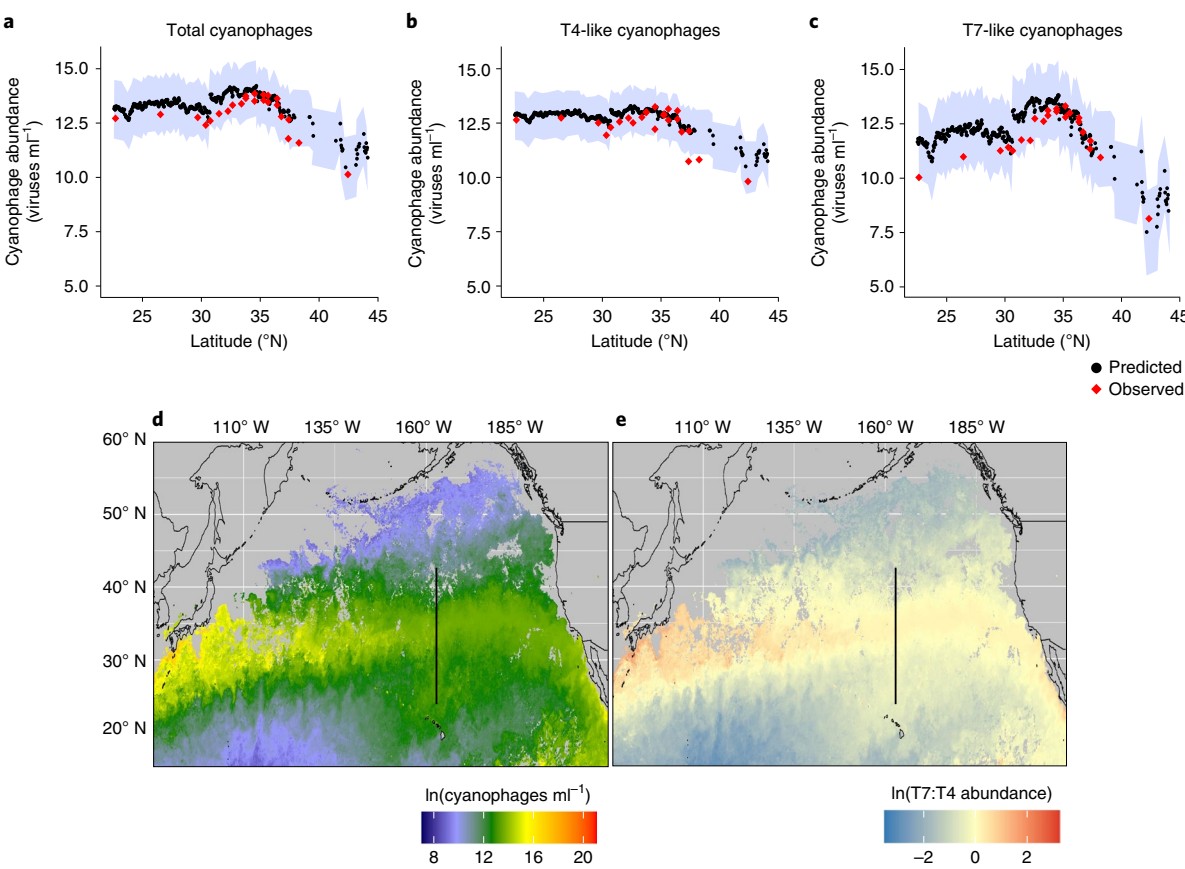

**Fig. 5 | Prediction of cyanophage abundances. a–c**, Model-based predictions of cyanophage abundances corresponding to the empirically measured total (**a**), T4-like (**b**) and T7-like clade B (**c**) cyanophage abundances along a transect in the North Pacific in April 2019. The shaded regions show the 95% confidence interval for the model predictions. **d,e**, Predicted total cyanophages (**d**) and the ratio of T4-like/T7-like clade B cyanophages (**e**) in June 2017 in the North Pacific Ocean. The black lines indicate the cruise track. The grey areas represent regions with no values due to cloud cover or that were beyond the limits of the predictive model. The hotspot peak corresponds to yellow regions in **d** and red regions in **e**.

of TIM5-like cyanophages increased in waters north of the hotspot (Fig. 3 and Extended Data Fig. 4d,i,m) but remained a minor component of the cyanophage community. No relationship was found between cyanophage and cyanobacterial abundances (Fig. 2d), and less than 1.5% of picocyanobacteria were infected by all cyanophage lineages in these waters (Fig. 4).

The cyanophage hotspot in the transition zone is a ridge of high virus activity that separates the subtropical and subpolar gyres. The reproducibility of our observations, which were separated by days to weeks within each cruise (2016 and 2017) and by years among the three cruises (Extended Data Fig. 4), indicates that this virus hotspot is a recurrent feature at the boundary of these two major gyres in the North Pacific Ocean. This suggests that the hotspot forms due to the distinctive environment of the inter-gyre transition zone creating conditions that enhance infection of picocyanobacteria and proliferation of cyanophages. *Prochlorococcus* in the transition zone may be prone to stress due to being close to the limits of their temperature growth range[5,6], which has the potential to increase susceptibility to viral infection. Alternatively, there may be temperature-dependent trade-offs between virus decay and production that lead to replication optima within a narrow temperature range[48]. Cyanophage infectivity has been observed to decay more slowly at colder temperatures[49], which may allow for the accumulation of infective viruses, leading to increased infection. In addition, cyanophage infections may be more productive due to enhanced nutrient supply in the transition zone[27] (Fig. 1h,i) relative to the subtropics, given that the cyanophages replicate in hosts with

presumably greater intracellular nutrient quota and obtain more extracellular nutrients, both of which may increase progeny production[9,10]. The environmental factors influencing the production and removal of viruses probably vary in intensity at different times, leading to variability in cyanophage abundance and infection levels. Thus, the putative cyanophage replication optimum in the hotspot may reflect the combined effects of temperature and nutrient conditions that are intrinsically linked to the oceanographic forces that shape the transition zone itself.

Changes in the cyanophage community structure over environmental gradients are likely to reflect differences in host range, infection properties and genomic potential to remodel host metabolism[9]. Our data, together with previous measurements in the North Pacific Subtropical Gyre[38,39], indicate that the T4-like cyanophages are the lineage best adapted to the low-nutrient waters of the subtropics (Fig. 2d–f). As these waters are inhabited by hundreds of genomically diverse subpopulations of *Prochlorococcus*[50], the broad host range of many T4-like cyanophages[18,19,22,51] may be advantageous for finding a suitable host. T4-like cyanophages also have a large and diverse repertoire of host-derived genes[21,51]—such as nutrient acquisition, photosynthesis and carbon-metabolism genes—that augment host metabolism[52] and may increase fitness in nutrient-poor conditions in the subtropics[51]. In contrast, T7-like clade A and B cyanophages seem to be better adapted to conditions in the transition zone (Fig. 3). T7-like cyanophages have narrow host ranges[19,22,40], with smaller genomes and fewer genes to manipulate the host metabolism[23], which may allow them to

replicate and produce more progeny in regions with elevated nutrient concentrations relative to subtropical conditions. The maximal abundances of TIM5-like cyanophages were found in the most productive waters at the northern end of the transects where the cyanobacterial abundances were lowest and *Synechococcus* was the dominant picocyanobacterium. This may be partially due to the narrow host range of TIM5-like cyanophages and their specificity for *Synechococcus*[40,44]. Our findings of reproducible lineage-specific responses to changing ocean regimes indicate that cyanophage lineages occupy distinct ecological niches.

Temperature and nutrient changes occurring in the transition zone are expected to result in shifts in picocyanobacterial diversity at the sub-genus level (Supplementary Discussion), which we speculate may affect community susceptibility to viral infection. One mechanism for this may be that the picocyanobacteria that thrive in the transition zone are intrinsically more susceptible to viral infection. Another scenario may be related to trade-offs associated with the evolution of resistance to viral infection. The horizontal advection of nutrient-rich waters to the transition zone[28] may select for rapidly growing cells adapted for efficient resource utilization. Viral resistance in picocyanobacteria often incurs the cost of reduced growth rates[53,54]. Thus, competition for nutrients in this region may favour cells with faster growth rates but increased susceptibility to viral infection. Thus, it is probable that the cyanophage distributions do not always follow the cyanobacterial patterns (Extended Data Fig. 2) because of complex interactions between lineage-specific cyanophage traits, host community structure and environmental variables, which may vary seasonally or annually as a result of inter-annual variability in environmental conditions (see below).

Despite consistent features in cyanophage distributions across the North Pacific Ocean, cyanophage infection was higher (Fig. 4 and Extended Data Fig. 7), whereas *Prochlorococcus* abundances were consistently lower (Fig. 2a), across the June 2017 transects relative to the March 2015 and April 2016 transects. Seasonality and/or climate variability could explain this interannual variability, although the data currently available to assess this are sparse. Viral infection of picocyanobacteria in the subtropical gyre increased from early spring to summer, suggesting a potential seasonal pattern that may extend across the transect (Extended Data Fig. 9a). In addition, the June 2017 transect occurred during a neutral-to-negative El Niño phase with lower sea-surface temperatures relative to the 2015 and 2016 transects, which were in years of a record marine heatwave, followed by a strong El Niño[55] (Extended Data Fig. 9b). In 2015 and 2016, the *Prochlorococcus* abundances were found to be higher than usual in the North Pacific Ocean in this (Fig. 2a) and other studies[56,57]. Irrespective of the underlying drivers for the observed interannual variability, we speculate that an ecosystem tipping point was reached in the hotspot under the prevailing conditions in June 2017, aided by the higher cyanophage abundances yet smaller *Prochlorococcus* population sizes. In this scenario, picocyanobacterial populations were subjected to high infection levels that resulted in an accumulation of cyanophages, initiating a stronger than usual positive-feedback loop between infection and virus production, and precipitating the unexpected *Prochlorococcus* decline. Continued observations in the North Pacific Ocean are needed to evaluate the potential link between seasonality and/or large-scale climate forcing as ultimate drivers affecting virus–host interactions.

**Predicting basin-scale virus dynamics.** Measurements of cyanobacterial and cyanophage abundances rely on discrete sample collection from shipboard oceanographic expeditions, which limits the geographical and seasonal extent of available data. Therefore, we developed a multiple regression model based on high-resolution satellite data of temperature and chlorophyll to predict cyanophage abundances, a key proxy of cyanobacterial infection (Pearson's $r = 0.61$, two-sided $P = 1.7 \times 10^{-8}$, degrees of freedom = 68, $n = 70$).

We used the model to estimate the geographical extent of the virus hotspot. The model accurately predicted the location of the hotspot and cyanophage abundances along a fourth transect in April 2019 (Supplementary Table 1), with the majority of observations falling within the 95% confidence intervals of the model predictions (Fig. 5a–c). Application of the model to the larger region predicted that the virus hotspot formed a boundary extending across the North Pacific Ocean, with lower cyanophage abundances on both sides (Fig. 5d,e and Supplementary Fig. 1). This boundary had the hallmarks of the hotspot with a core that was dominated by T7-like cyanophages and the flanking gyre regions dominated by T4-like cyanophages. Thus, this feature may be more appropriately termed a 'hot-zone' due to its substantial projected aerial extent. Assuming the infection levels observed in the hotspot in June 2017 were similar throughout the hot-zone, the potential habitat loss for *Prochlorococcus* would be about $3.2 \times 10^6 \, \text{km}^2$, approximately half of the cumulative area loss of the Amazonian rainforest to date[58].

**Virus hotspot biogeochemistry.** With the ability to predict biogeographic patterns of cyanophages, we evaluated the potential biogeochemical implications of virus-mediated picocyanobacterial lysis and release of organic material in sustaining the bacterial community[6–9]. The aerial extent of the hot-zone (approximately $4 \times 10^6 \, \text{km}^2$) is only 14% of the size of the subtropical gyre ($2.9 \times 10^7 \, \text{km}^2$), and yet the total virus-mediated organic matter released from picocyanobacteria in the hot-zone in June 2017 was estimated to be on par with that for the entire North Pacific Subtropical Gyre (Methods and Supplementary Discussion). We estimate that viral lysate released from picocyanobacteria in the subtropical gyre could sustain $4.4 \pm 0.8\%$ of the calculated bacterial carbon demand there (Extended Data Fig. 10). In contrast, viral lysate released in the transition zone could sustain an average of $21 \pm 12\%$ of the bacterial carbon demand, reaching 33% in some regions (Extended Data Fig. 10), assuming that the bacterial assimilation and growth efficiencies were similar between the subtropical gyre and the hotspot. Thus, local generation of cyanobacterial viral lysate in the transition zone is likely to be an important source of carbon for the heterotrophic bacterial community that can rapidly utilize large molecular weight dissolved organic matter[59] and may have contributed to the increase in their abundances south of the chlorophyll front in 2017 (Extended Data Fig. 1a,e).

## Conclusions

The oceans have few geographical boundaries, yet marine ecosystems exhibit clear changes in species distributions. Although the habitat of a species is constrained by a combination of abiotic and biotic interactions, the distributions of marine microbes have classically been linked only to abiotic conditions. Here we show that the region between the North Pacific Ocean gyres harbours a previously hidden virus hotspot with reproducible and distinct community composition and host–virus dynamics. This hotspot is superimposed on gradients in abiotic conditions, and together they influence important processes that shape the ecological succession of major marine primary producers and the cycling of organic matter in this region. The formation of the hotspot and the variation within was probably a result of distinct combinations of environmental conditions that ensued at different times, potentially having differential effects on virus diversity, infectivity and production as well as on host diversity and susceptibility to co-occurring viruses. Our modelling enabled predictions of viruses and their potential impact on picocyanobacterial distributions and biogeochemistry at a large geographical scale. Expansion of this model to other ocean regions, determination of population traits that lead to these ecosystem features and the development of population models for cyanophages and other autotroph-virus systems will allow us to gain a global view of the impacts of viruses on marine ecosystems in both present-day and future oceans[11,12,14–16,30].

## Methods

**Sample collection.** Samples were collected in the North Pacific Ocean during 21–29 March 2015 and 10–29 April 2019 aboard the RV *Kilo Moana*, 20 April–4 May 2016 aboard the RV *Ka'imikai-O-Kanaloa* and 26 May–13 June 2017 aboard the RV *Marcus G. Langseth*. The three cruises conducted in 2016–2019 were out-and-back along the 158° W longitudinal line, whereas the 2015 cruise transited unidirectionally from Oregon to Hawaii. Temperature and salinity measurements were collected continuously by a shipboard Sea-Bird Electronics SBE-21 thermosalinograph. Nutrients were collected into acid-cleaned high-density polyethylene bottles using a CTD rosette equipped with Niskin bottles closed at 15 m depth at latitude intervals of approximately 0.5°. The samples were immediately frozen at −20 °C until analysis in the laboratory. All samples used in the virus analyses (virus-like particles, cyanophage abundances, and infection) were collected every 4–8 h from the flow-through seawater system of the ship, which was located at depths between 5 m and 8 m.

For the analyses of viral abundances and infection levels, water was first filtered through a 20 μm mesh. Ten-millilitre samples were collected for iPolony infected-cell analysis, amended with glutaraldehyde (0.1% final concentration), incubated at 4 °C in the dark for 15–30 min, frozen in liquid nitrogen and stored at −80 °C. Forty-millilitre samples were filtered through a 0.2 μm syringe top filter to collect the virus-containing filtrate (Millipore, Durapore). For samples used in polony analyses of free cyanophages, the 0.2 μm filtrate was frozen as is at −80 °C. For samples used in the analyses of virus-like particles, formaldehyde (2% final concentration) was added to the filtrate, incubated for 15–30 min in the dark and stored at −80 °C.

Discrete samples used for the validation of picocyanobacteria abundances determined by SeaFlow ('Picocyanobacteria and heterotrophic bacteria analysis' section) and for enumeration of heterotrophic bacteria abundances were collected approximately every 8–12 h or at latitude intervals of approximately 0.5° and amended with glutaraldehyde (0.2% final concentration), incubated for 15–30 min in the dark and frozen as above.

**Environmental conditions.** The concentrations of soluble reactive phosphorus and nitrate + nitrite were measured as described by Foreman et al.[60] using a segmented flow SEAL AutoAnalyzer III with high-resolution detectors and following the colorimetric reactions of Murphy and Riley[61], and Strickland and Parsons[62], respectively. The limit of quantification of these methods is approximately 30 nmol l$^{-1}$ for soluble reactive phosphorus and 9 nmol l$^{-1}$ for nitrate + nitrite, and the average precision is 0.2% and 0.4%, respectively. The accuracies, determined from daily analyses of Wako CSK standard phosphate and nitrate solutions, were within 2%. Low nitrate + nitrite concentrations (<0.5 μmol l$^{-1}$) were determined using a high-sensitivity chemiluminiscent method[63] with a detection limit of 1 nmol l$^{-1}$. The precision of the high-sensitivity method ranges from 0.4% at 1,000 nmol l$^{-1}$ to 7% at 2 nmol l$^{-1}$. Underway particulate beam attenuation measurements were collected using a transmissometer (Wetlabs C-star), calibrated to discrete measurements of particulate organic carbon made via the standard high combustion method as described by White et al.[64]. Data were merged to an average resolution of 3 min for comparison to underway flow cytometry. Mixed-layer depth was calculated assuming the mixed-layer threshold is the depth at which the surface-referenced potential density is 0.1 kg m$^{-3}$ greater than at the surface[65] from CTD profiles taken across the 2016 and 2017 transects as well as from the March climatology based on Argo float profiles for the 2015 cruise given that no CTD profiles were taken on this transect. Sea-surface chlorophyll levels were derived from 4 km resolution, 32 d rolling composite MODIS-Aqua satellite products.

**Virus analysis.** Cyanophage abundances and virally infected picocyanobacteria were analysed using the polony[37] and iPolony[39] methods, respectively. Briefly, the virus-containing fraction of seawater or *Prochlorococcus* and *Synechococcus* that had been flow cytometrically sorted ('Picocyanobacteria and heterotrophic bacteria analysis' section) were mixed with 10% polyacrylamide and an acrydite-modified primer specific to the cyanophage group of interest. Gels were poured on 40 μm deep Blind-Silane pre-treated microscope slides (Thermo Fisher Scientific), polymerized for 30 min under argon gas to embed and spatially separate the viruses or cells, washed with sterile MilliQ water and 0.025% Tween-20 and dried. Polymerase chain reaction mixes containing 1×Taq buffer, 0.25 mM deoxyribonucleotide triphosphate mix, non-modified primer and 0.67 U μl$^{-1}$ Jumpstart Taq polymerase (Sigma-Aldrich) were then diffused into the gels and the gels were covered with mineral oil. For a list of primers, their concentrations in reactions and the thermal cycling conditions refer to Supplementary Table 2 (refs. [37–39]). After thermal cycling, the slides were washed with buffer E (10 mM Tris pH 7.5, 50 mM KCl, 2 mM EDTA pH 8 and 0.01% Triton X-100) to remove oil and the PCR reagents. The amplicons were denatured with heat and 70% formamide and then washed with buffer E to remove unbound template. A hybridization mix (900 mM NaCl, 60 mM NaH$_2$PO$_4$, 6 mM EDTA and 0.01% Triton X-100) containing degenerate probes specific to the virus family of interest and internal to the PCR primers were applied to the gels and hybridized. See Supplementary Table 2 for the probe sequences, concentrations and hybridization conditions[37,38]. After hybridization, excess probe was washed off with three rinses of buffer E. The slides were scanned using a Genepix 4000A or B microarray scanner and the Genepix Pro v5.0 software. All reactions were performed with a minimum of two technical replicates. Amplicons, also known as polonies, were enumerated using the ImageJ v1.0 software and normalized to the input volume of seawater for the free cyanophage analyses or to the number of cells in each reaction for the infected-cell analyses. The polony concentrations were then converted to cyanophage abundances or infected-cell percentages based on the previously determined efficiencies of detection[37–39]. In the case of infected cells, an additional correction for co-sorted free cyanophages was determined empirically for several samples or using a correction factor based on the number of free cyanophages[39]. For the free cyanophage analyses, a bootstrapping with resampling approach was used to determine the 95% confidence intervals based on the efficiencies of polony formation[37]. For infected cells, the upper and lower bounds were determined based on the efficiency of detection, assuming that all cells were in the early stages of infection when the detection efficiency is lowest or that all cells were in late-stage infections when detection was maximal[39]. The infected-cell abundances were derived from multiplying the per cent infection by the abundance of picocyanobacteria. Log-transformed cyanophage and infected-cell abundances were used to determine the linear correlation between the variables.

To convert instantaneous measurements of infection to daily virus-mediated picocyanobacteria mortality, we assumed that mortality is relative to the number of infection cycles that can occur in one day, as done previously[39]. The virus-mediated daily mortality rates were estimated by multiplying the instantaneous measurements of infection by the number of infection cycles that could be completed in 24 h based on the average latent period for cyanophage–cyanobacteria pairs from previously reported literature values (Supplementary Table 3). These culture-based values were typically determined in culture at approximately 21 °C at light intensities in the range of 10–50 μmol photons m$^{-2}$ s$^{-1}$—much dimmer than those in the upper mixed layer (see Supplementary Table 3 references). Thus, we further refined these estimates to be more conservative by adjusting for the impact of day length, light levels and temperature. First, we applied a correction for temperature, assuming that the latent period lengthened by 25% for every 3 °C decrease in temperature from 21 °C and similarly shortened for increasing temperatures (I. Pekarski and D. L., unpublished data). Second, cyanophage infections in high light intensities (210 μmol photons m$^{-2}$ s$^{-1}$) are 40% shorter than those conducted at low light intensities (15 μmol photons m$^{-2}$ s$^{-1}$)[66]. Third, we applied this light correction such that cyanophages had latent periods that were 60% shorter during daylight hours and that no lysis occurred during the night, as suggested previously[39]. These assumptions yielded estimates of about 3–4 cycles per day for the dominant cyanophage types in the subtropical gyre[39] and 2–3 infection cycles per day considering the cooler temperatures and shorter day length in the transition zone. These assumptions yield conservative estimates in mortality. Using fewer assumptions, such as not adjusting for temperature or not considering day length and light levels, resulted in higher levels of mortality (a maximum of 66% versus the 51% reported here).

The viral-induced mortality per generation was calculated using the cumulative distribution function:

$$M = 1 - e^{-I \times \frac{\ln(2)}{\mu}},$$

where $M$ is the fraction of the total population that would be expected to be infected before division, $I$ is the instantaneous infection level and $\mu$ is the estimated division rate of *Prochlorococcus* based on temperature using the relationships for high light I (strains MED4 and MIT9515) and high light II ecotypes[67] (strains MIT9312 and MIT9215) or *Synechococcus* clades I (strain MVIR-16–2) and II (strain M16.1)[46]. We assumed that the doubling time of a natural population at any given temperature is equal to the division rate of the faster growing ecotype at that temperature. As such, *Prochlorococcus* high light II ecotypes were assumed to dominate at temperatures above 14.7 °C in the subtropics and the southern transition zone, whereas high light I ecotypes were assumed to dominate at temperatures below that threshold in the northern transition zone, consistent with empirical observations of *Prochlorococcus* community structure along thermal gradients in the field[12,14,15]. Similarly, *Synechococcus* clade II were assumed to dominate at temperatures above 23 °C in the subtropics, whereas clade I was assumed to dominate at temperatures below that threshold in the northern transition zone and subpolar waters.

The abundances of virus-like particles were enumerated after staining the formaldehyde-fixed 0.2 μm filtrate with 100× SYBR Green I, filtration onto a 0.02 μm 25 mm filter (Whatman, Anodisc) and visualization through epifluorescence microscopy using a Leica Application Suite X system[68]. At least 20 random fields of view per filter were captured. All slides were analysed with at least two technical replicates. The virus-like particles were enumerated using the ImageJ v1.0 software.

**Picocyanobacteria and heterotrophic bacteria analysis.** The optical properties of cyanobacteria cells were measured continuously using SeaFlow, an underway flow cytometer[69]. The cell abundance, size and carbon content of each cell were calculated as in Ribalet et al.[70] using the R package popcycle v1.1.

Heterotrophic bacteria were analysed from discrete samples by staining each sample with SYBR Green (1×final concentration) for 15 min in the dark on ice. The stained samples were run on a BD Influx flow cytometer equipped with a small-particle detector and the cells were enumerated from a known volume of at least 50 μl. The unstained samples were also analysed to enumerate the picocyanobacteria. The concentration of picocyanobacteria was subtracted from the concentration of total bacteria to give the concentration of heterotrophic bacteria. The analyses were performed in triplicate and analysed using the FACSDiva v8 or Spigot software.

See Supplementary Fig. 2 for examples of the flow cytometry gating strategies.

**Defining oceanic regimes and the virus hotspot.** Oceanic regimes in the North Pacific Ocean were delineated based on definitions previously described in Roden[29] where the subtropical gyre was limited to waters with salinity >34.6, the subpolar gyre was limited to waters with salinity <33 and the intermediate waters designated the North Pacific Transition Zone. These boundaries were based on shipboard salinity measurements or, in the absence of these measurements, satellite-based sea-surface salinity measurements derived from the NASA Soil Moisture Active Passive platform at 9 km[2] resolution during the cruise periods[71]. In addition, we used chlorophyll concentrations from the MODIS-Aqua satellite at 4 km resolution over a rolling 32-d average during the cruise periods to serve as a biological marker of the different oceanic regimes (cloud cover substantially reduced the usefulness of satellite observations over shorter time scales)[72]. The 0.2 mg m[−3] chlorophyll contour used to define the Transition Zone Chlorophyll Front is based on previous observations reported by Polovina and colleagues[28].

The virus hotspot was defined as the position where the rates of change of the cyanophage abundances was most rapid. To remove the variability related to the seasonal migration of the transition zone, temperature was used as the independent variable against which to plot the cyanophage abundances and because the relationship was consistent between all four cruises (Fig. 2c). First, the cyanophage abundances ($n = 119$) were scaled to each cruise-wide maximum and then normalized to the average abundance in the subtropics to centre the data. The abundances were smoothed over a 2.5 °C sliding window. The second derivative of cyanophage abundances as a function of temperature was calculated and the maximum was used to define the temperature range of the hotspot (14.7−18.4 °C). This hotspot range was then translated to geospatial coordinates for each cruise independently according to the relationship between temperature and latitude.

**Model projection of cyanophage abundances.** We used a train-test splitting approach to identify the variables that best explained the observed T7-like, T4-like and total cyanophage abundances. The data from the 2015–2017 cruises ($n = 87$) were used as a training set and the data from the 2019 cruise ($n = 24$) were used as a testing set. The log-transformed abundances of *Prochlorococcus*, *Synechococcus* and total cyanobacteria; log-transformed chlorophyll concentration; the *Prochlorococcus*:*Synechococcus* ratio and their interaction with temperature were tested as predictors. The best model was chosen by comparison of the root-mean-square error (RMSE) that was calculated based on predicted ($P_i$) and observed ($O_i$) cyanophage abundances (equation (1))[73]. The RMSE is proportional to the scale of the dataset, with small values indicating high precision and large values indicating low precision. Although two models commonly had the best predictive value (*Synechococcus* × temperature and *Prochlorococcus*:*Synechococcus* × temperature), these models were not considerably better based on RMSE compared with models using chlorophyll (Supplementary Table 1). Furthermore, there are far fewer publicly available datasets that report global-scale *Prochlorococcus* and/or *Synechococcus* abundances than those that report chlorophyll. For these reasons, we employed the chlorophyll-based models for predictions. For all cyanophage groups, the interaction between temperature ($T$) and the chlorophyll (Chl) concentration (equations (2)–(4)) showed the relatively high predictive powers of the model (Supplementary Table 1). The fitted parameter values ($a$, $b$, $c$) are listed in Supplementary Table 4. All calculations were performed using the 'stats' package v3.6.2 in R.

$$\text{RMSE} = \left( \frac{\sum_{i=1}^{n} (P_i - O_i)^2}{n} \right)^{1/2} \tag{1}$$

$$\text{Total cyanophages} = a_0 + a_1 T + a_2 \text{Chl} + a_3 T \times \text{Chl} \tag{2}$$

$$\text{T7-like cyanophages} = b_0 + b_1 T + b_2 \text{Chl} + b_3 T \times \text{Chl} \tag{3}$$

$$\text{T4-cyanophages} = c_0 + c_1 T + c_2 \text{Chl} + c_3 T \times \text{Chl} \tag{4}$$

For graphing maps of cyanophage distributions (Fig. 5 and Supplementary Fig. 1), we obtained values for the sea-surface chlorophyll concentration and sea-surface temperature from the MODIS-Aqua satellite at 4 km resolution over a 32-d rolling composite[74]. Data that covered a latitudinal and longitudinal range from 15° N to 70° N and 110° E to 100° W were extracted using SeaDAS v7.5.3. The cyanophage abundances were predicted using the above linear regression model for regions warmer than 5 °C as a bound on the thermal range of picocyanobacteria growth and where the chlorophyll concentrations were lower than 0.4 mg m[−3]. The maps were produced using the 'mapdata' v2.3.0 and 'ggplot2' v5.5.5 R packages[75,76].

**Organic matter turnover and bacterial production.** Virus-mediated organic-matter production was estimated by subtracting the carbon in virus particles produced during infection from the total cellular carbon using the following equation:

$$L_{\text{vir}} = PM_{\text{Pro}} \left( C_{\text{Pro}} - C_{\text{vir}} B \right) + SM_{\text{Syn}} \left( C_{\text{Syn}} - C_{\text{vir}} B \right),$$

where $L_{\text{vir}}$ is the lysate generated per day, $P$ is the abundance of *Prochlorococcus*, $S$ is the abundance of *Synechococcus*, $M_{\text{Pro}}$ and $M_{\text{Syn}}$ are the daily estimated mortality for *Prochlorococcus* and *Synechococcus*, respectively, $C_{\text{Pro}}$ and $C_{\text{Syn}}$ are the daily average carbon quota per cell for *Prochlorococcus* and *Synechococcus*, respectively, derived from SeaFlow ('Picocyanobacteria and heterotrophic bacteria analysis' section), $C_{\text{vir}}$ is the carbon content for a virus particle as reported in Jover et al.[77] for either the T7-like cyanophage Syn5 or the coliphage T4 as representatives of T7-like and T4-like cyanophages, respectively, and $B$ is the estimated burst size (Supplementary Table 3). Note that the proportion of total cellular carbon of a single virus particle is <0.5% and <0.1% for T4-like and T7-like cyanophages, respectively. Most reported burst-size measurements for cyanophages are different estimates of virus production (Supplementary Table 3). They measure infective viruses, total particles or free and packaged genomic DNA. Although these are not the same measure, we combined them in our use here. Nevertheless, our calculation was not strongly impacted by these methodological differences, both because the highest burst sizes of about 200 T7-like cyanophages cell[−1] would be approximately 2% of the total carbon and these reports are for T7-like clade A viruses (I.M., I. Pekarski and D.L., unpublished data), which are minor in their contribution to infection in this dataset (Extended Data Figs. 3 and 4). The lysate production was calculated for both T4- and T7-like cyanophages and summed to estimate the total carbon released through the viral lysis of *Prochlorococcus* and *Synechococcus* daily. Average virus-mediated organic-matter production was determined for each of the three regimes, multiplied by the average surface area and average mixed-layer depth of the corresponding regime to estimate the cumulative virus-mediated production of organic matter per regime.

*Prochlorococcus* and *Synechococcus* cultured at 24 °C have elemental stoichiometries of 208 C:P and 6.4 N:P (ref. [78]), and 130.7 C:P and 7.6 N:P (ref. [79]), respectively. These ratios were used to convert cellular carbon to nitrogen and phosphorus production. We assumed that all cellular material released due to lysis, except for the progeny viruses, is biologically available to heterotrophic bacteria and that bacteria assimilate 100% of the organic matter. The average estimated virus-mediated organic-matter production was then divided by previously reported measurements of dissolved organic-matter production in the North Pacific Subtropical Gyre and other gyres. These values were 0.07 μmol C l[−1] d[−1] from Viviani et al.[80], 1.8 nmol P l[−1] d[−1] from Björkman et al.[81] and 33.6 nmol N l[−1] d[−1] from Sipler and Bronk[82].

Bacterial production was determined through the measurement of ³H-leucine incorporation according to previously described methods[83,84]. Briefly, triplicate 30–40 ml samples were collected with Niskin bottles at a depth of 15 m pre-dawn with one sample as a killed control by the addition of trichloroacetic acid (1% final concentration). Each sample was spiked with approximately 3 GBq L-[3,4,5−³H(N)]-leucine (Perkin Elmer) to yield an added concentration of approximately 20 nM ³H-leucine. The samples were incubated for 1–3 h in the dark in on-deck flow-through incubators at in situ temperature. The samples were then filtered by vacuum onto 25 mm GF75 filters and the incorporation of ³H-leucine into protein (insoluble in cold 5% trichloroacetic acid) was quantified in 10 ml of Ultima Gold LLT liquid scintillation cocktail using a scintillation counter. Bacterial production was determined from leucine incorporation as previously described[83], where the fraction of leucine in protein is 7.3% and ratio of cellular carbon per protein is 0.86. Bacterial carbon demand was estimated using the measured bacterial production values and an assumed bacterial growth efficiency of 0.15 that is typical of oligotrophic communities[85–87].

**Statistics.** Pearson correlation analyses between cyanophage and cyanobacterial abundances were performed using RStudio v1.2.5019 and Python v3.8.2.

**Reporting Summary.** Further information on research design is available in the Nature Research Reporting Summary linked to this article.

## Data availability

The data that support the findings of this study are available at https://simonscmap.com/catalog/cruises/ in the directories KM1502, KOK1606, MGL1704 and KM1906 (ref. [88]). Source data are provided with this paper.

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

## Acknowledgements

We thank the captains and crews of the RV *Ka'imikai-O-Kanaloa*, RV *Kilo Moana* and RV *Marcus G. Langseth* as well as O. Sosa for underway phosphate measurements and comments on the manuscript. In addition, we thank K. W. Brandt, R. Morales, M. Schatz, A. Hynes and A. Replogle for technical help and use of equipment and the Gradients team for discussions. Funding was provided by the Simons Foundation (Gradients grant no. 426570 and SCOPE grant no. 329108 to D.L., E.V.A., A.E.W. and D.M.K.; and LIFE grant nos. 529554 and 721254 to D.L.) and European Research Council (grant no. ERC-CoG 646868 to D.L.). M.C.G.C. was supported by a Fulbright Postdoctoral Fellowship. This manuscript is a contribution of the Simons Collaboration on Ocean Processes and Ecology (SCOPE).

## Author contributions

M.C.G.C. and D.L. designed the study. M.C.G.C., F.R., I.M., B.P.D., Y.H., S.F., J.W., N.S., B.B.C., A.E.W. and E.V.A. participated in the sampling efforts. M.C.G.C., Y.H., S.G. and N.B. performed the virus analyses. M.C.G.C., F.R. and E.V.A. analysed the bacterial abundances. S.F., D.M.K. and A.E.W. contributed to the inorganic and organic water chemistry analyses. M.C.G.C., I.M. and B.B.C. conceptualized and developed the predictive models. B.P.D. performed and analysed the bacterial uptake experiments. M.C.G.C. and D.L. wrote the paper with contributions from all authors.

## Competing interests

The authors declare no competing interests.

## Additional information

**Extended data** is available for this paper at https://doi.org/10.1038/s41564-022-01088-x.

**Correspondence and requests for materials** should be addressed to Debbie Lindell.

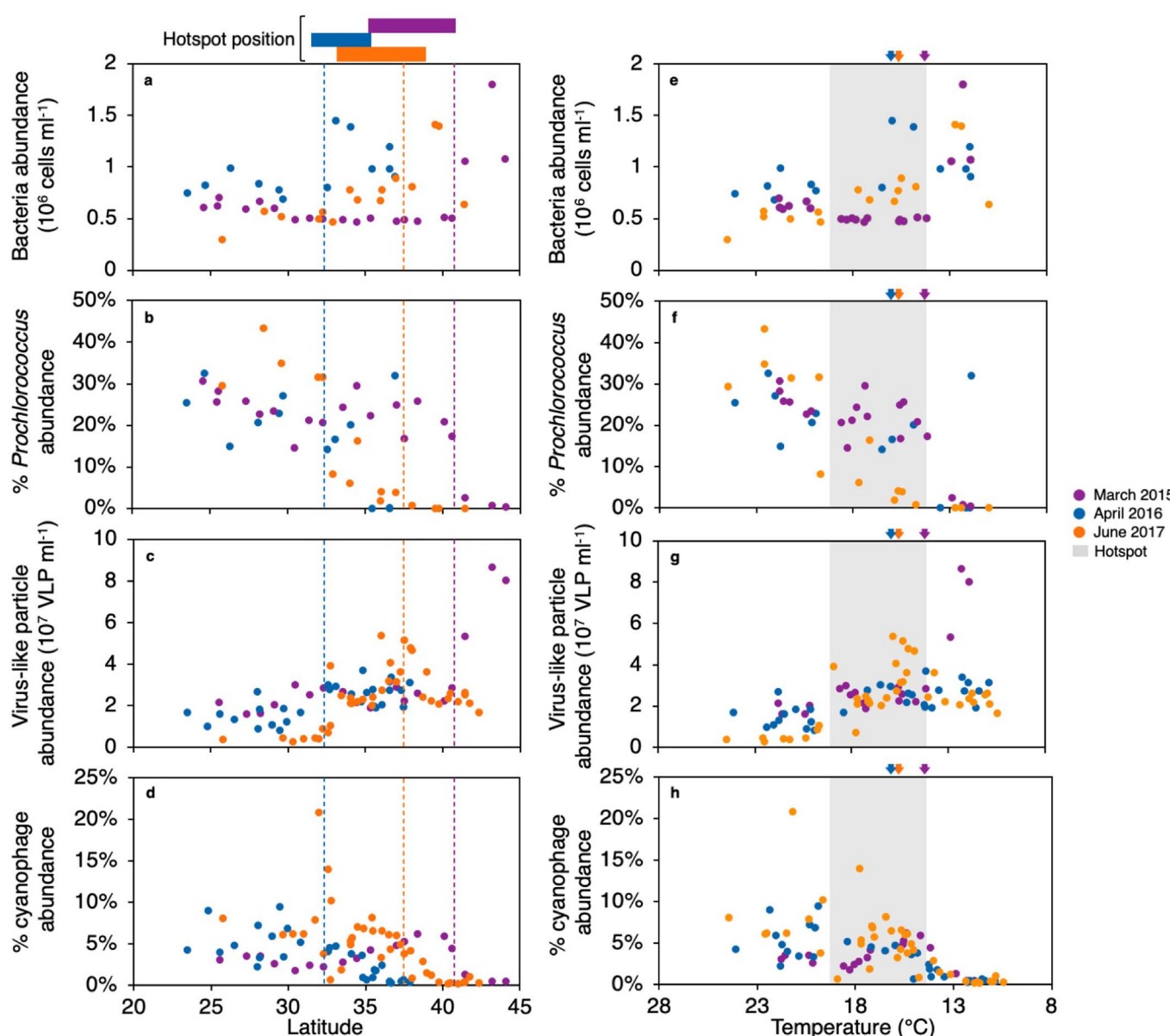

**Extended Data Fig. 1 | Bacteria and virus abundances across the transects.** Total bacteria (heterotrophic and picocyanobacteria) (**a, e**), percent *Prochlorococcus* of total bacteria (**b, f**), total virus-like particles (**c, g**), and percent cyanophages of total virus-like particles (**d, h**) in the surface waters of the North Pacific in March 2015 (purple), April 2016 (blue), and June 2017 (orange). Data are plotted against latitude (**a-d**) or temperature (**e-h**). Dashed lines (**a-d**) or arrows (**e-h**) show the latitude of the 0.2 mg·m⁻³ chlorophyll contour as in Fig. 1. Colour bars at the top (**a-d**) or shaded region (**e-h**) indicate the position of the virus hotspot.

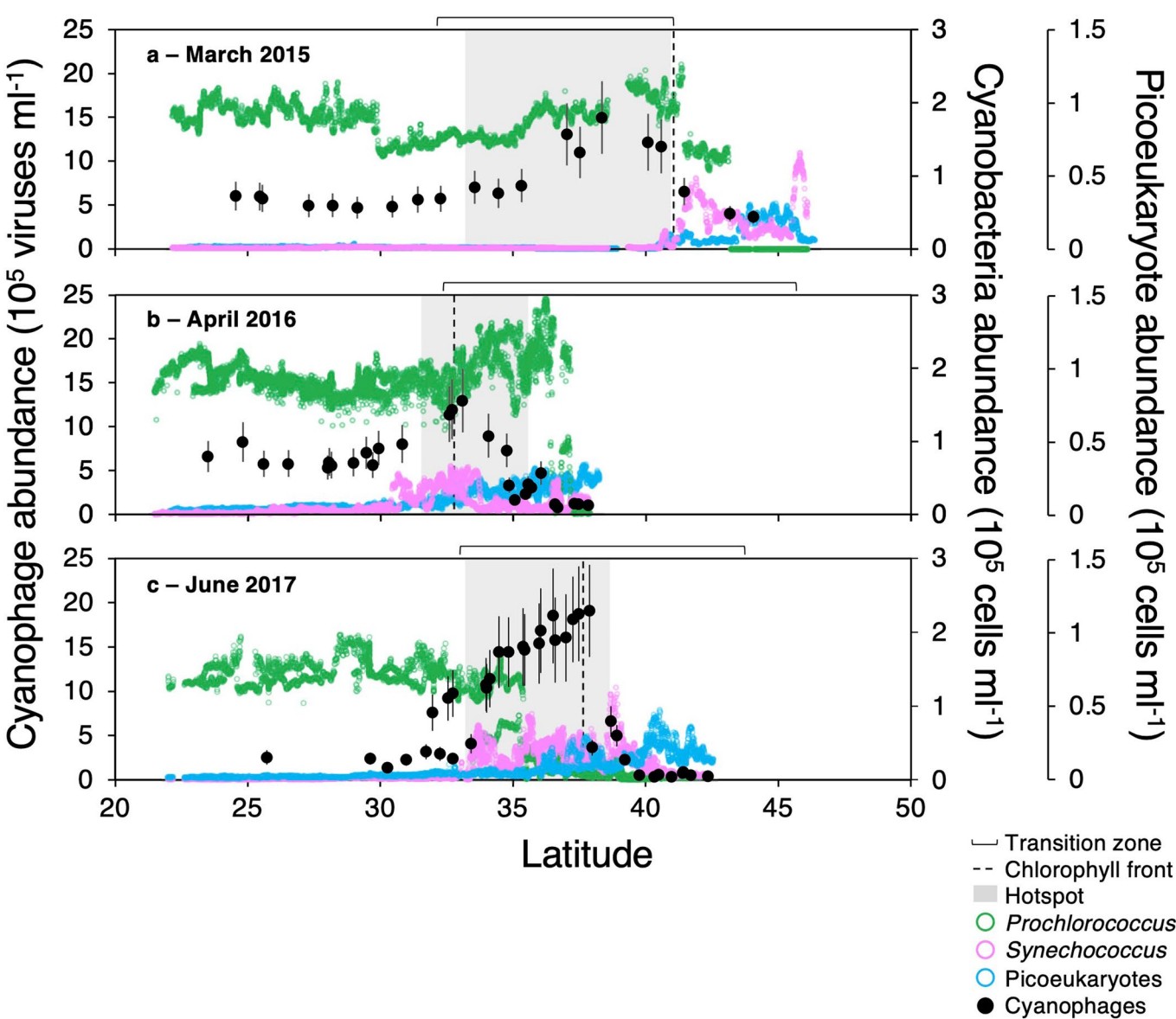

**Extended Data Fig. 2 | Latitudinal distributions of picophytoplankton and cyanophages in the North Pacific Ocean.** Semicontinuous sampling of picophytoplankton across the transects. Note that the scale of picoeukaryote abundances is 2-fold lower than that of the picocyanobacteria. Cyanophage abundances are averages determined from 10,000 bootstrapping and resamplings of the phage-to-polony conversion efficiencies (see Methods). Error bars show 95% confidence intervals for cyanophage abundances which are the 95% quantiles of the bootstrap analyses.

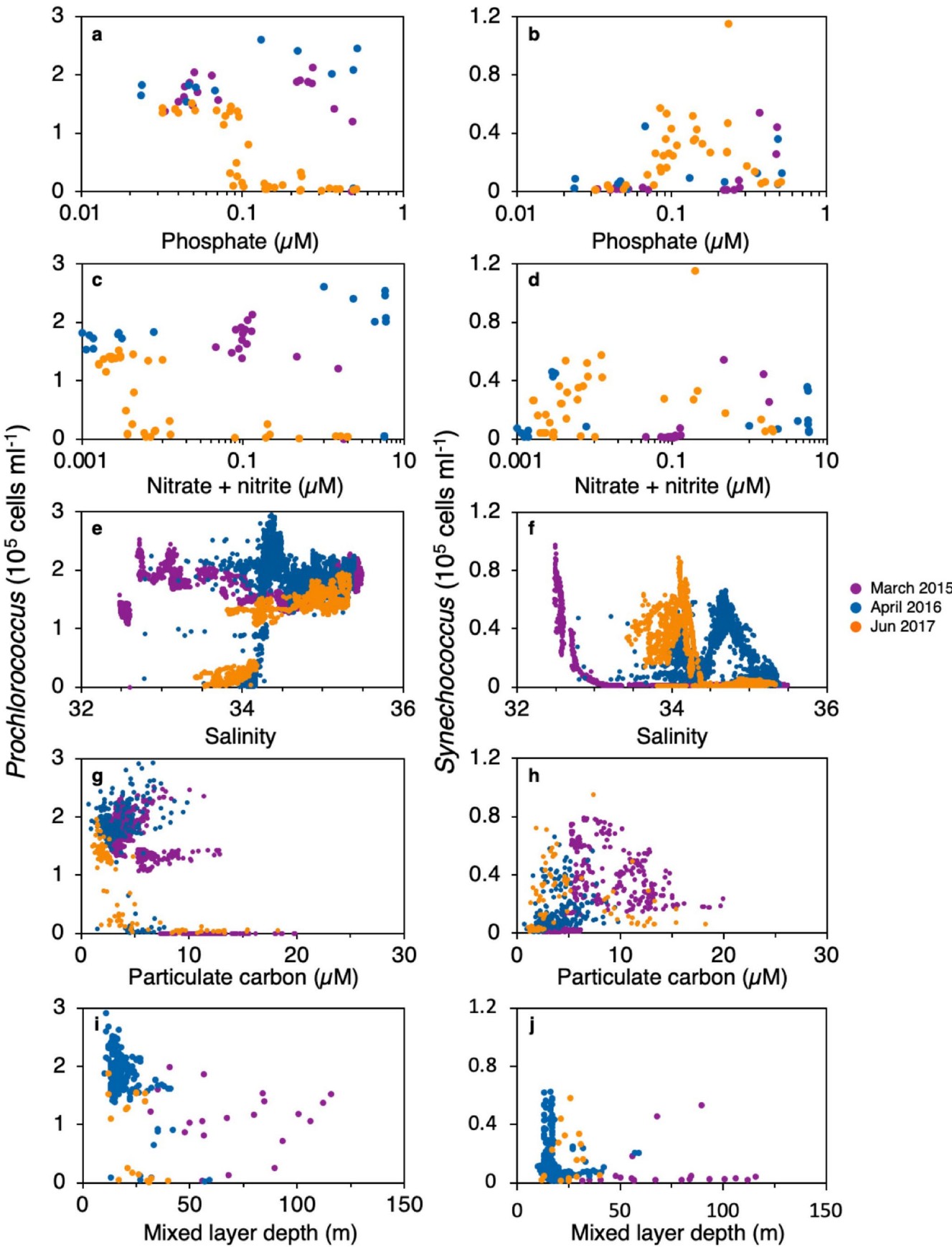

**Extended Data Fig. 3 | Comparison of picocyanobacterial abundances to abiotic factors.** *Prochlorococcus* (left) and *Synechococcus* (right) abundances plotted against concentrations of phosphate (**a,b**), nitrate+nitrite (**c,d**), particulate carbon (**e,f**), salinity (**g,h**), and mixed layer depth (**i,j**) for the March 2015 (purple), April 2016 (blue), and June 2017 (orange) transects.

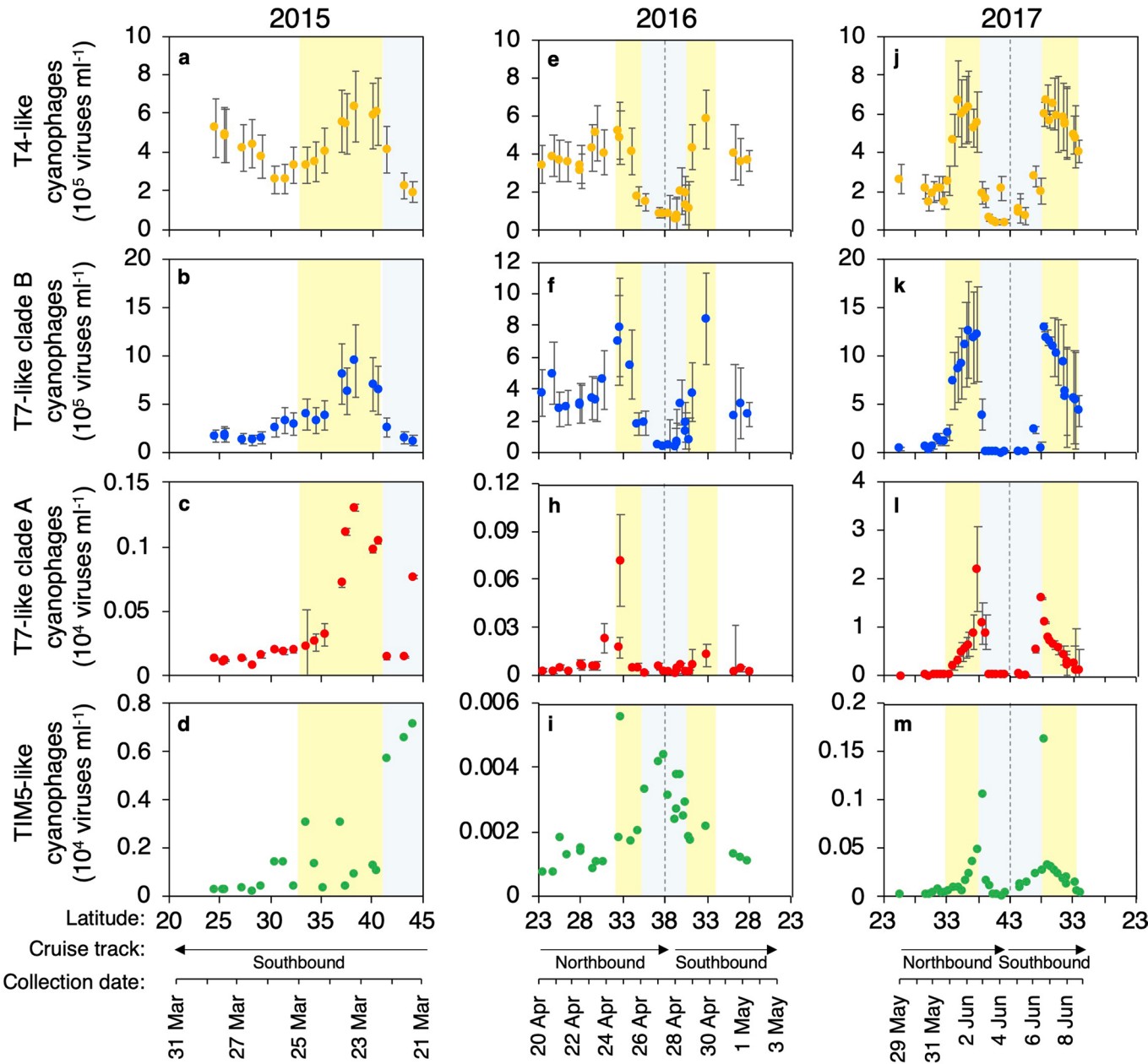

**Extended Data Fig. 4 | Recurrent patterns in the abundances of different cyanophage lineages.** T4-like (orange), T7-like clade B (blue), T7-like clade A (red), and TIM5-like (green) cyanophage abundances are plotted for the 2015 (left panel, **a-d**), 2016 (middle panel, **e-i**), and 2017 (right panel, **j-m**) transects against latitude (primary x-axis). Northbound and southbound legs of the transects in 2016 and 2017, indicated below the axis, are unfolded on the x-axis as they overlap, and dashed lines indicate the switch in direction between the two transect legs. Date of sample collection is shown as a secondary x-axis. Cyanophage abundances are averages determined from 10,000 bootstrapping and resamplings of the phage-to-polony conversion efficiencies (see Methods). Error bars show 95% confidence intervals for cyanophage abundances which are the 95% quantiles of the bootstrap analyses. The location of the hotspot is indicated by yellow shading, sampling occurring northward of the hotspot is indicated by light blue shading. Note the change in the magnitude of the vertical axis for the different cyanophage lineages.

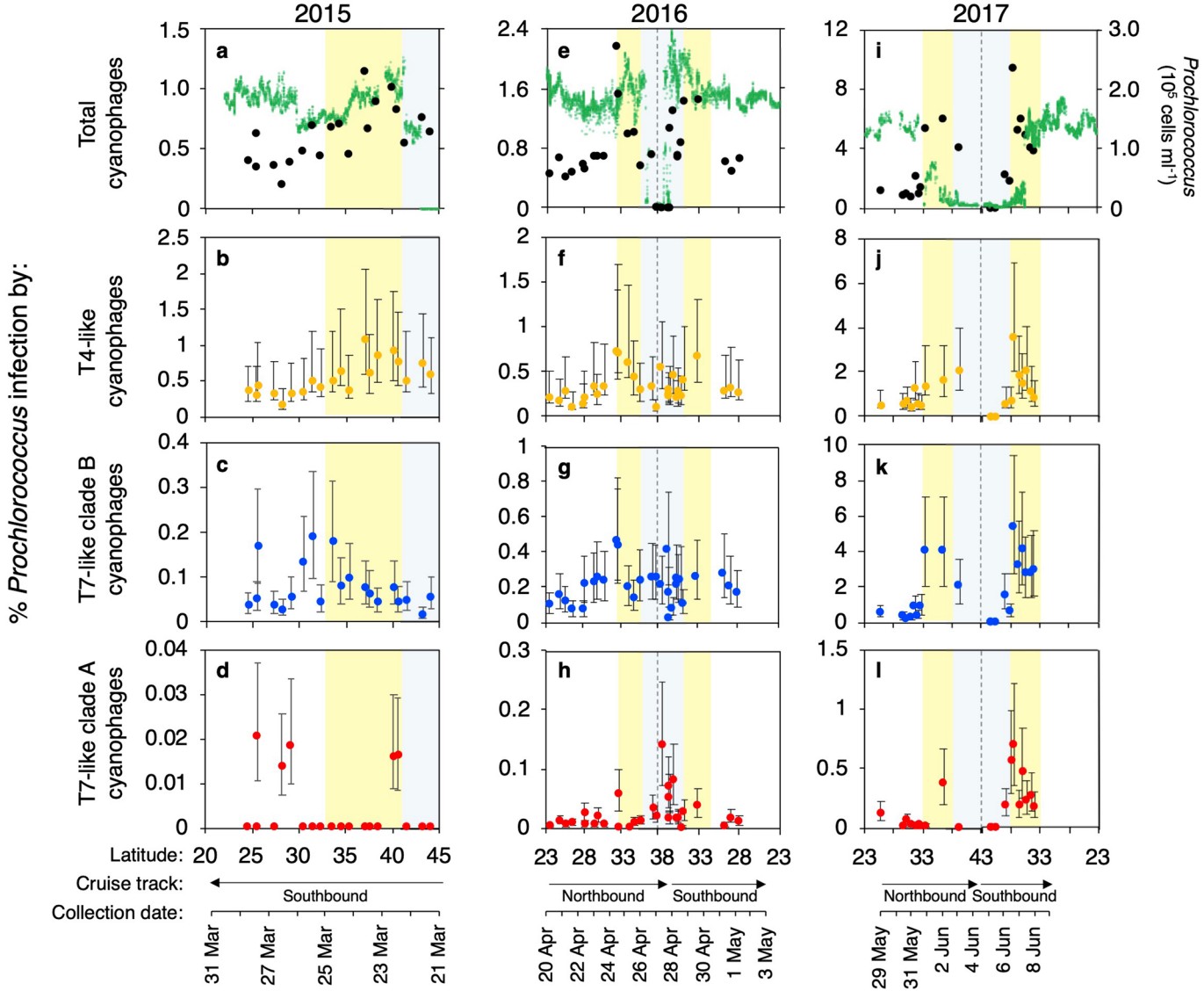

**Extended Data Fig. 5 | Recurrent patterns in the infection of *Prochlorococcus* by different cyanophage lineages.** *Prochlorococcus* abundances are shown in green (top row: **a, e, i**). Infection levels by all cyanophages combined (black: **a, e, i**), T4-like (orange: **b, f, j**), T7-like clade B (blue: **c, g, k**), and T7-like clade A (red: **d, h, l**) cyanophages are plotted against the latitudinal position of the cruise track. Northbound and southbound legs of the transects in 2016 and 2017, indicated below the axis, are unfolded on the x-axis as they overlap, and dashed lines indicate the switch in direction between the two transect legs. Date of sample collection is shown as a secondary x-axis. Infection levels are averages determined from 10,000 bootstrapping and resamplings of the cell-to-polony conversion efficiencies (see Methods). The maximum and minimum bounds of infection are represented by error bars. The limit of accurate detection of infection using this assay is <0.05% infection. The location of the hotspot is indicated by yellow shading, sampling occurring northward of the hotspot is indicated by blue shading.

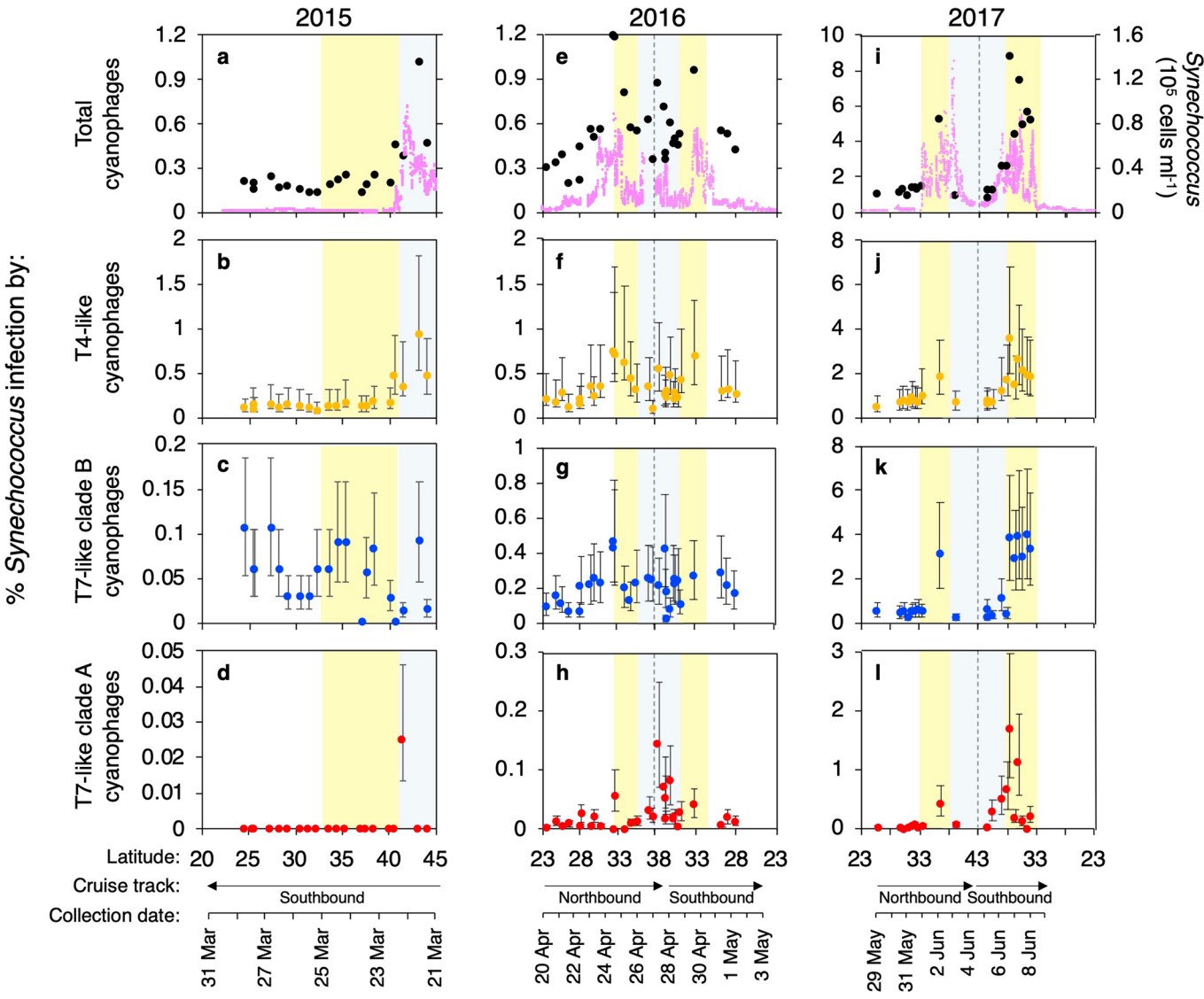

**Extended Data Fig. 6 | Recurrent patterns in the infection of *Synechococcus* by different cyanophage lineages.** *Synechococcus* abundances are shown in pink (top row: **a, e, i**). Infection levels by all cyanophages combined (black: **a, e, i**), T4-like (orange: **b, f, j**), T7-like clade B (blue: **c, g, k**), and T7-like clade A (red: **d, h, l**) cyanophages are plotted against the latitudinal position of the cruise track. Northbound and southbound legs of the transects in 2016 and 2017, indicated below the axis, are unfolded on the x-axis as they overlap, and dashed lines indicate the switch in direction between the two transect legs. Date of sample collection is shown as a secondary x-axis. Infection levels are averages determined from 10,000 bootstrapping and resamplings of the cell-to-polony conversion efficiencies (see Methods). The maximum and minimum bounds of infection are represented by error bars. The limit of accurate detection of infection using this assay is <0.05% infection. The location of the hotspot is indicated by yellow shading, sampling occurring northward of the hotspot is indicated by blue shading.

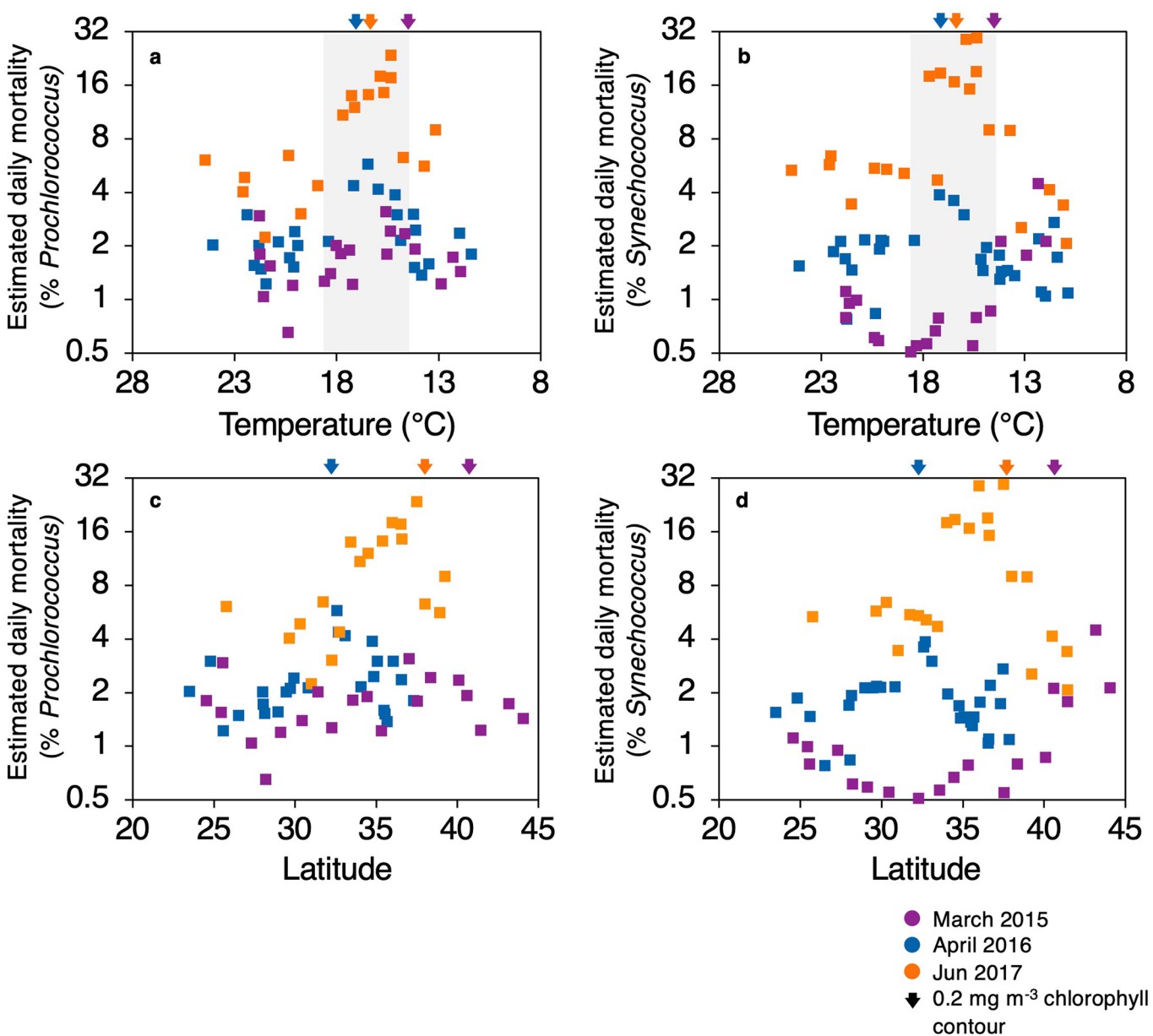

**Extended Data Fig. 7 | Estimated daily virus-mediated mortality of the picocyanobacterial in the North Pacific Ocean.** Mortality of *Prochlorococcus* (**a**, **c**) and *Synechococcus* (**b**, **d**) plotted against temperature (**a, b**) and latitude (**c, d**) was estimated by multiplying the instantaneous infection measurements by the number of infection cycles that could be completed in 24 hours based on temperature and light intensity adjusted average cyanophage latent periods (Supplementary Table 3).

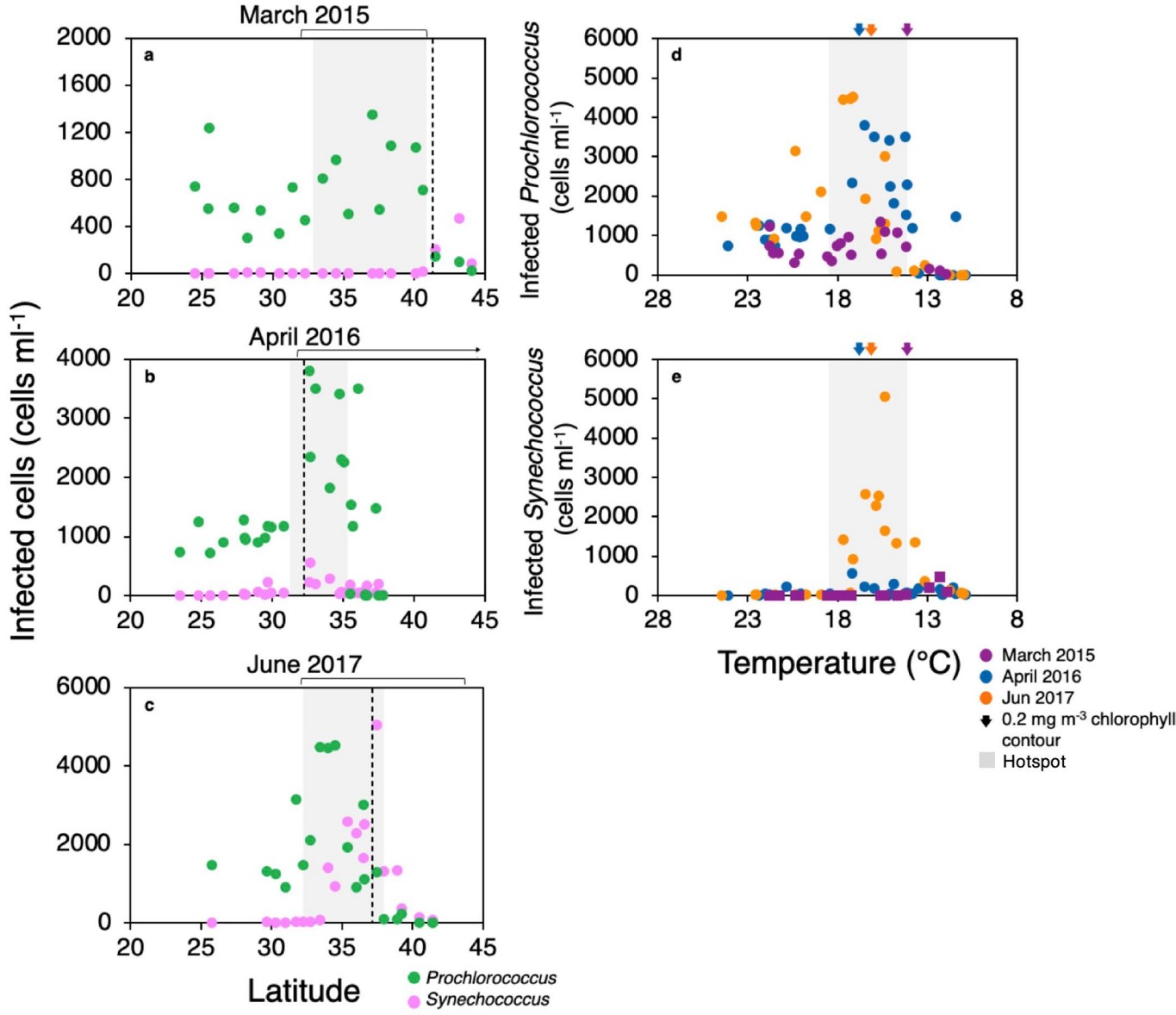

**Extended Data Fig. 8 | Changes in infected cell densities across latitude and temperature.** The density (cells·ml$^{-1}$) of total infected *Prochlorococcus* (green) and *Synechococcus* (pink) are shown by latitude for the March 2015 (**a**), April 2016 (**b**), and June 2017 (**c**) transects. *Prochlorococcus* (**d**) and *Synechococcus* (**e**) infected cell abundances plotted against temperature across the March 2015 (purple), April 2016 (blue), and June 2017 (orange) transects. Shaded regions indicate the hotspot, dashed lines (**a-c**) or arrows (**d**, **e**) indicate the chlorophyll front, and brackets (**a-c**) indicate the transition zone.

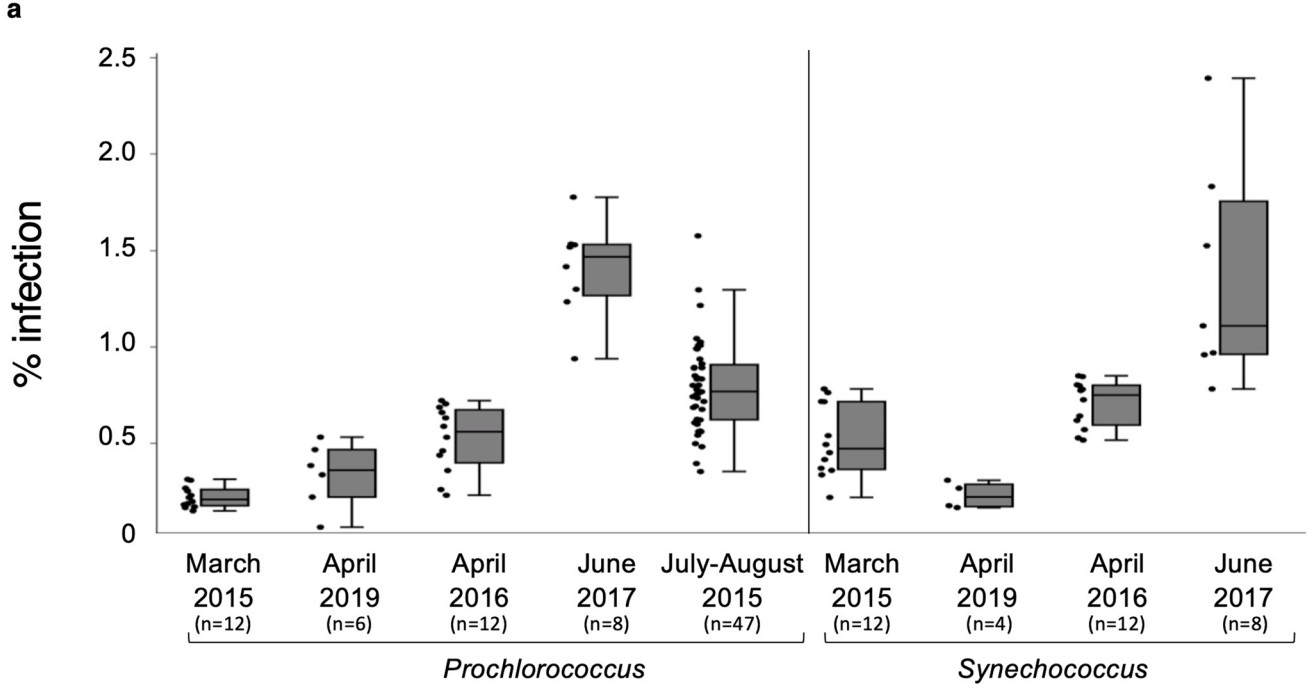

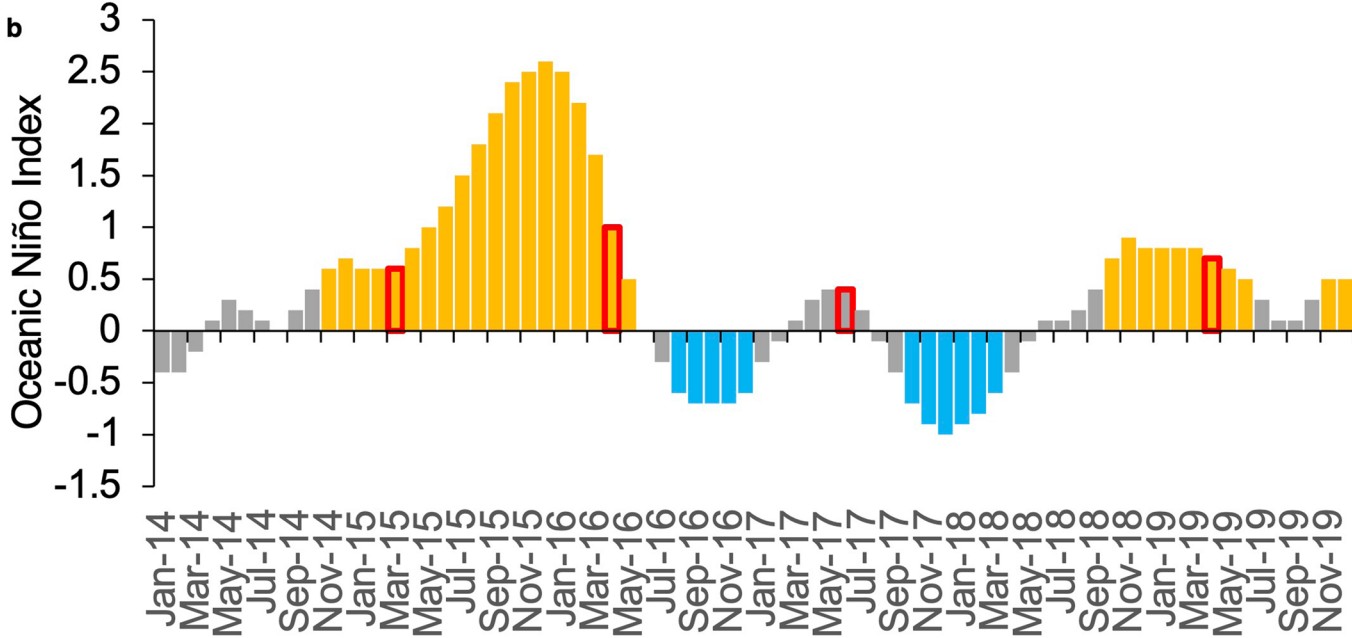

**Extended Data Fig. 9 | Seasonal and interannual infection dynamics.** (**a**) Seasonally increasing infection levels for *Prochlorococcus* (left) and *Synechococcus* (right) from cruises in March 2015, April 2019, April 2016, June 2017 (this study), and July-August 2015[39] in the subtropical gyre. Boxes show the median, 1st quartile, 3rd quartile, minimum and maximum are shown by box-and-whisker plots. Individual data points are shown to the left of each box. (**b**) Oceanic Niño Indices for 2014–2019 from the National Oceanographic and Atmospheric Administration[89]. Warm phases (>0.5) are indicated by orange, neutral phases (−0.5–0.5) are indicated by grey, and cool phases (<−0.5) are indicated by blue. Red boxes indicate the timing of the March 2015, April 2016, July 2017 and April 2019 cruises in this study. A record marine heatwave occurred in 2015 and 2016 when infection levels were low. The 2017 transects which had high viral infection occurred during a cool phase of the El Niño Oscillation. When El Niño conditions and the warm temperature anomaly returned in 2019, infection levels were low in the subtropical gyre in April, consistent with both seasonal and climatic hypotheses.

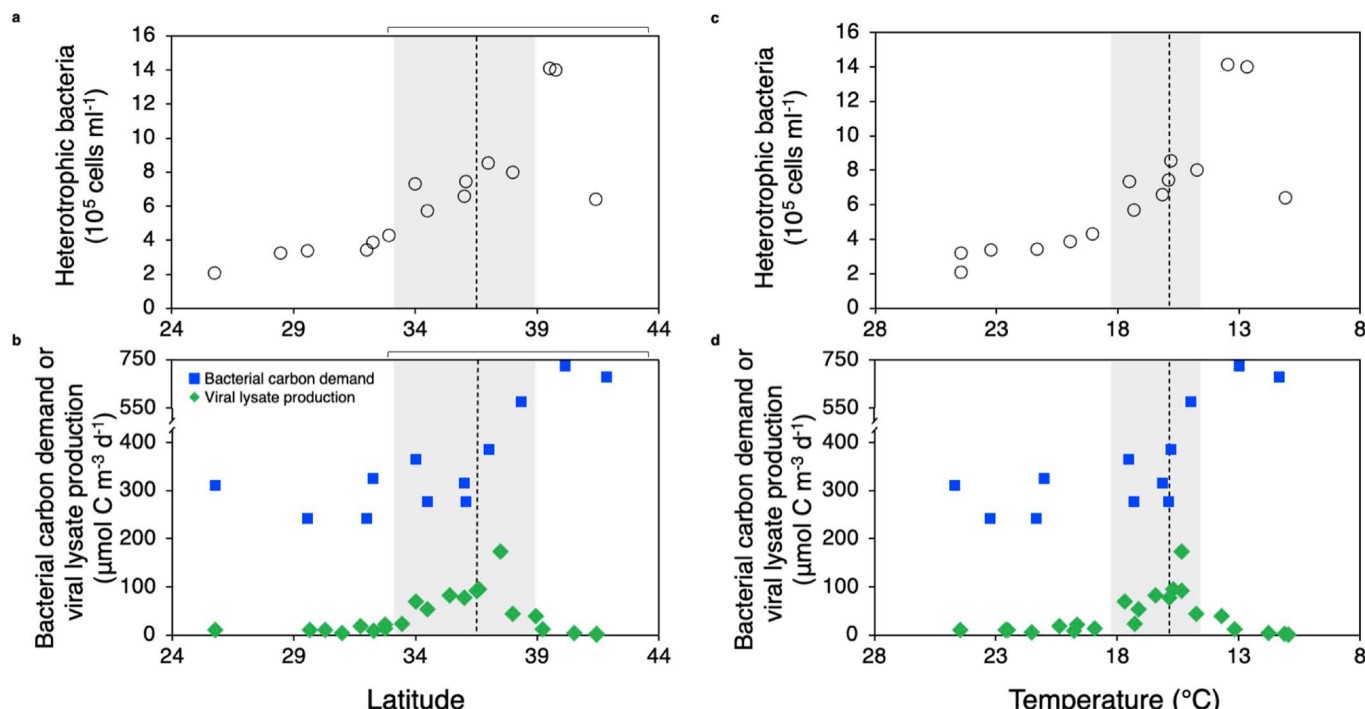

**Extended Data Fig. 10 | Virus-mediated organic matter production and heterotrophic bacteria abundances across the transects in June 2017.** (**a, c**) The abundances of heterotrophic bacteria along the June 2017 transects are reproduced from Extended Data Fig. 1 for ease of comparison. (**b, d**) Estimated bacterial carbon demand (blue) calculated from [3]H-Leucine incorporation and an assumed bacterial growth efficiency of 0.15 and virus-mediated lysate production (green) based on daily picocyanobacteria mortality. Dashed lines indicate the 0.2 mg·m$^{-3}$ chlorophyll front. Shaded regions indicate the position of the cyanophage hotspot and brackets indicate the transition zone.

# Reporting Summary

## Statistics

For all statistical analyses, confirm that the following items are present in the figure legend, table legend, main text, or Methods section.

| n/a | Confirmed | |
|---|---|---|
| ☐ | ☒ | The exact sample size (*n*) for each experimental group/condition, given as a discrete number and unit of measurement |
| ☐ | ☒ | A statement on whether measurements were taken from distinct samples or whether the same sample was measured repeatedly |
| ☐ | ☒ | The statistical test(s) used AND whether they are one- or two-sided <br> *Only common tests should be described solely by name; describe more complex techniques in the Methods section.* |
| ☒ | ☐ | A description of all covariates tested |
| ☒ | ☐ | A description of any assumptions or corrections, such as tests of normality and adjustment for multiple comparisons |
| ☐ | ☒ | A full description of the statistical parameters including central tendency (e.g. means) or other basic estimates (e.g. regression coefficient) AND variation (e.g. standard deviation) or associated estimates of uncertainty (e.g. confidence intervals) |
| ☐ | ☒ | For null hypothesis testing, the test statistic (e.g. *F*, *t*, *r*) with confidence intervals, effect sizes, degrees of freedom and *P* value noted <br> *Give P values as exact values whenever suitable.* |
| ☒ | ☐ | For Bayesian analysis, information on the choice of priors and Markov chain Monte Carlo settings |
| ☒ | ☐ | For hierarchical and complex designs, identification of the appropriate level for tests and full reporting of outcomes |
| ☐ | ☒ | Estimates of effect sizes (e.g. Cohen's *d*, Pearson's *r*), indicating how they were calculated |

*Our web collection on statistics for biologists contains articles on many of the points above.*

## Software and code

Policy information about availability of computer code

| Data collection | Continuous flow cytometery measurements were collected using SeaFlow software (Swalwell et al. 2011) <br> Discrete flow cytometry data were acquired using FACSDiva v8 or Spigot software |
|---|---|
| Data analysis | Continuous flow cytometry data were analyzed with the R package popcycle v1.1 <br> Discrete flow cytometry data were analyzed using FACSDiva v8 or Spigot software <br> Polony data were analyzed using GenePix Pro v 5.0 and ImageJ v 1.0 <br> Microscopy data were analyzed using Leica Application Suite X and ImageJ v 1.0 <br> Satellite data were analyzed using SeaDAS v7.5.3 and cyanophage map data was analyzed using R packages 'stats' v 3.6.2, 'mapdata' v 2.3.0, and 'ggplot' v 5.5.5 <br> Nutrient concentrations were analyzed using AQ software v 2.4.5 <br> RStudio v 1.2.5019 and Python v 3.8.2 were used to analyze discrete data |

For manuscripts utilizing custom algorithms or software that are central to the research but not yet described in published literature, software must be made available to editors and reviewers. We strongly encourage code deposition in a community repository (e.g. GitHub). See the Nature Portfolio guidelines for submitting code & software for further information.

# Data

Policy information about availability of data

All manuscripts must include a data availability statement. This statement should provide the following information, where applicable:

- Accession codes, unique identifiers, or web links for publicly available datasets
- A description of any restrictions on data availability
- For clinical datasets or third party data, please ensure that the statement adheres to our policy

The data that support the findings of this study are available at https://simonscmap.com/catalog/cruises/ in the directories KM1502, KOK1606, MGL1704, KM1906. Additionally, discrete data presented here are provided in Supplementary dataset file 1.

# Field-specific reporting

Please select the one below that is the best fit for your research. If you are not sure, read the appropriate sections before making your selection.

☐ Life sciences     ☐ Behavioural & social sciences     ☒ Ecological, evolutionary & environmental sciences

For a reference copy of the document with all sections, see nature.com/documents/nr-reporting-summary-flat.pdf

# Ecological, evolutionary & environmental sciences study design

All studies must disclose on these points even when the disclosure is negative.

| Study description | Latitudinal surveys of microbial and viral abundances and infection along 5 transects in 3 cruises between 2015-2017 in the North Pacific Ocean. Data from a fourth cruise in 2019 was used for validation of the multiple regression model built using data from the previous 3 cruises. |
|---|---|
| Research sample | Open ocean microbes and viruses and associated environmental conditions to investigate the effect of environmental conditions on cyanophage abundance and impact on picocyanobacteria. The samples represent the populations in the water masses they were collected in. |
| Sampling strategy | Sampling was performed at ~5m depth at high spatial resolution to assess the changes in microbial and viral dynamics along environmental gradients. Cruise tracks were designed to traverse the changing environment between the North Pacific Subtropical and Subpolar Gyres. Sampling schemes were designed such that the sample size collected in each regime would be statistically robust (n>=5) and occur at regularly spaced intervals. |
| Data collection | Samples were collected in the North Pacific Ocean from March 21 - March 29 2015 and April 10 - April 29 2019, from April 20 - May 4 2016, and from May 26th - June 13th 2017. Temperature and salinity were collected by shipboard sensors. Nutrients were collected into acid-cleaned high-density polyethylene bottles using a CTD rosette equipped with Niskin® bottles closed at 15 m depth. The samples were immediately frozen at -20 ºC until analysis in the laboratory. All samples used in virus analyses (VLPs, cyanophage abundances, and infection) were collected from each ship's flow-through seawater system located at ~5 m depth.

For analyses of viral abundance and infection levels, water was first filtered through a 20 μm mesh. Ten milliliter samples were collected for iPolony infected cell analysis, amended with glutaraldehyde (0.1% final concentration), incubated at 4 ºC in the dark for 15-30 minutes, snap frozen in liquid nitrogen, and stored at -80 ºC. Forty milliliter samples were filtered through a 0.2 μm syringe top filter to collect the virus-containing filtrate (Millipore, Durapore). For samples used in polony analyses of free cyanophages, the 0.2 μm filtrate was frozen as is at -80 ºC. For samples used in virus-like particle analyses, formaldehyde (2% final concentration) was added to the filtrate, incubated for 15-30 minutes in the dark, and stored at -80 ºC. Discrete samples used for validation of picocyanobacteria abundances determined by SeaFlow and for enumeration of heterotrophic bacteria abundances were collected and amended with glutaraldehyde (0.2% final concentration), incubated for 15-30 minutes in the dark, and snap frozen.

Data were recorded by Angelicque White, Michael Carlson, Francois Ribalet, Katie Watkins-Brandt, Nitzan Shamir, Julia Weissenbach, Ilia Maidanik, Sara Ferron, Bryndan Durham, Oscar Sosa, and Yotam Hulata. Data were recorded either automatically when sampled continuously in electronic files or in a sampling log. |
| Timing and spatial scale | Cruise tracks were designed to transit between the subtropical gyre to the subpolar gyre and cover the most ground in between given the ship time allotted. Each transect covered ~2000-2500 km. Underway flow cytometry and particulate carbon were sampled continuously. Discrete samples (for virus and cellular analyses and water chemistry) were taken every 4 or 8 hours over the sampling period. Bacterial uptake experiments were conducted so that experiments were performed at a spatial scale to evenly sample the environmental gradients on the 2017 cruise.

Dates of sampling and spatial scale of the transects were as follows:
March 21 – March 27, 2015, 3191.0 km transect
April 20 – May 1, 2016, 1591.0 km transect, 2 transect traversals
May 29 – June 10, 2017, 1839.5 km transect, 2 transect traversals
April 10 – April 27, 2019, 2188.7 km transect, 2 transect traversals |
| Data exclusions | No data were excluded from analyses. |

| Reproducibility | The 2016, 2017, and 2019 cruise were designed to traverse the same longitude line and were out-and-back transects. |
|---|---|

All polony reactions were performed with at least technical duplicates. Select polony samples were repeated by two different people to assess potential systematic bias (none found). Additionally, select polony reactions for infected cells were run in separate labs, using different cytometers and reagents to verify data reproducibility.

Bacterial and viral abundances analyzed via flow cytometry or epifluorescent microscopy were run with 2-3 technical replicates.

Bacterial uptake experiments were preformed in triplicate.

All attempts at reproducibility were successful.

| Randomization | Discrete measurements were analyzed in a random order. All samples were analyzed. For virus-like particle and polony measurements, fields of measurement were chosen at random. No experimental groups were needed in this analysis as no grouping existed. |
|---|---|

| Blinding | Blinding was performed to the extent that samples were given alphanumeric codes and thus analyzed without reference to collection site to minimize observer bias. No a priori expectations existed for these samples as they are the first to assess latitudinal gradients in cyanophage abundance and infection. |
|---|---|

Did the study involve field work?    ☒ Yes    ☐ No

# Field work, collection and transport

| Field conditions | The upper mixed layer of the subtropical and subpolar gyres, and the transition zone between them in the North Pacific Ocean. Environmental conditions ranged between:<br>Temperature: $10.8 - 26.3$ °C<br>Salinity: $32.5 - 35.5$<br>Nitrate+nitrate: $<0.009 - 5.87$ μM<br>Phosphate: $0.023 - 0.513$ μM<br>Particulate carbon: $0.67 - 18.3$ μM<br>Mixed layer depth: 10-116 m |
|---|---|

| Location | Sampling was conducted between 5-15m depth in the North Pacific Ocean. The cruise tracks transited between:<br>(year: southernmost lat, long - northernmost lat, long)<br>2015: 24.55N, -153.12W - 44.05 N, 127.27 W<br>2016: 23.49 N, 157.98 W - 37.84 N, 158.00 W<br>2017: 25.77 N, 158.02 W - 42.35 N, 158.00 W<br>2019: 22.46 N, 158.00 W - 42.20 N, 157.6 W |
|---|---|

| Access & import/export | Sampling location were accessed on the RVs Kilo Moana, Ka'imikai O Kanaloa, and Marcus G. Langseth. All materials were imported and exported in compliance with local and international laws. No permits were needed. |
|---|---|

| Disturbance | No disturbance was caused by these cruises. |
|---|---|

# Reporting for specific materials, systems and methods

We require information from authors about some types of materials, experimental systems and methods used in many studies. Here, indicate whether each material, system or method listed is relevant to your study. If you are not sure if a list item applies to your research, read the appropriate section before selecting a response.

### Materials & experimental systems

| n/a | Involved in the study |
|---|---|
| ☒ ☐ | Antibodies |
| ☒ ☐ | Eukaryotic cell lines |
| ☒ ☐ | Palaeontology and archaeology |
| ☒ ☐ | Animals and other organisms |
| ☒ ☐ | Human research participants |
| ☒ ☐ | Clinical data |
| ☒ ☐ | Dual use research of concern |

### Methods

| n/a | Involved in the study |
|---|---|
| ☒ ☐ | ChIP-seq |
| ☐ ☒ | Flow cytometry |
| ☒ ☐ | MRI-based neuroimaging |

# Flow Cytometry

## Plots

Confirm that:

☒ The axis labels state the marker and fluorochrome used (e.g. CD4-FITC).

☒ The axis scales are clearly visible. Include numbers along axes only for bottom left plot of group (a 'group' is an analysis of identical markers).

☒ All plots are contour plots with outliers or pseudocolor plots.

☒ A numerical value for number of cells or percentage (with statistics) is provided.

## Methodology

| | |
|---|---|
| Sample preparation | For analyses of infection levels, water was first filtered through a 20 μm mesh. Ten milliliter samples were collected for iPolony infected cell analysis, amended with glutaraldehyde (0.1% final concentration), incubated at 4 ºC in the dark for 15-30 minutes, snap frozen in liquid nitrogen, and stored at -80 ºC. Discrete samples used for validation of picocyanobacteria abundances determined by SeaFlow and for enumeration of heterotrophic bacteria abundances were collected and amended with glutaraldehyde (0.2% final concentration), incubated for 15-30 minutes in the dark, and snap frozen.<br><br>Prior to running, samples were thawed at 30 degrees C in a water bath in the dark and transferred to ice once thawed. All samples were amended with 1 μm yellow-green beads as an internal reference. Heterotrophic bacteria were analyzed from discrete samples by staining each sample with SYBR Green (1X final concentration) for 15 minutes in the dark on ice. |
| Instrument | SeaFlow or a BD Influx flow cytometer |
| Software | Continuous flow cytometery measurements were collected using SeaFlow software (Swalwell et al. 2011)<br>Discrete flow cytometry data were acquired using FACSDiva v8 or Spigot software |
| Cell population abundance | Prochlorococcus and Synechococcus cells were typically sorted at concentrations between 800-1000 cells per ul. Sorted cell populations were >99% pure as determined by fluorescence signatures of sorted cells. |
| Gating strategy | Synechococcus was gated and sorted based on their orange autofluorescence (phycoerythrin containing cells) and size based on forward scatter. Prochlorococcus was gated and sorted based on red autofluorescence (chlorophyll containing cells) and size based on forward scatter. Hierarchical gating based on orange then red fluorescence was employed to discriminate between Prochlorococcus and Synechococcus from mixed communities. This gating strategy has been extensively documented in previous literature (e.g. Chisholm et al. Nature 1988, Swalwell et al. Limnol. Oceanogr. Meth. 2011, Casey et al. Deep Sea. Res. II 2013, Thyssen et al. Front. Microbiol. 2014, Rii et al. Mar. Ecol. Prog. Ser. 2016). |

☒ Tick this box to confirm that a figure exemplifying the gating strategy is provided in the Supplementary Information.

