## [Peer Review File · Nature Microbiology]

Peer Review Information

Journal: Nature Microbiology

Manuscript Title: Viruses affect picocyanobacterial abundance and biogeography in the North Pacific Ocean

Corresponding author name(s): Professor Debbie Lindell

Editorial Notes:

**Redactions –
unpublished data**

**Redactions – published
data**

Reviewer Comments & Decisions:

Decision Letter, initial version:

9th August 2021

Dear Professor Lindell,

Thank you for your patience while your manuscript "A virus hotspot at ocean gyre boundaries limits the geographic range of *Prochlorococcus*" was under peer review at Nature Microbiology. It has now been seen by our referees, whose expertise and comments you will find at the end of this email. In the light of their advice, we have decided that we cannot offer to publish your manuscript in Nature Microbiology.

From the reports, you will see that while they find your work of some potential interest, the referees raise concerns about the methods (including the modelling and the polony), the sampling (different cruises), and the insufficient exclusion of competing hypothesis (other abiotic factors; other organisms, like grazers; etc). Unfortunately, these criticisms are sufficiently important as to preclude publication of your work in Nature Microbiology.

Although we cannot offer to publish your manuscript, I suggest that you consider transferring your manuscript to the Springer Nature journal, The ISME Journal. I have provided a link to automatically transfer your files in the footnote below, and no reformatting is required.

I am sorry that we cannot be more positive on this occasion, but hope that you find the referees' comments helpful when preparing your paper for resubmission elsewhere.

Yours sincerely,
{redacted}

Reviewer Expertise:

Referee #1: Marine cyanobacteria, cyanophage ecology

Referee #2: Phage identification and quantification methods/polony method

Referee #3: Mathematical Modelling

Reviewers Comments:

Reviewer #1 (Remarks to the Author):

This is a review of "A virus hotspot at ocean gyre boundaries limits the geographic range of *Prochlorococcus*" (NMICROBIOL-21061597) The manuscript presents data from multiple occupations of

2an area in the North Pacific Ocean and shows the patterns of cyanobacteria (*Synechococcus*-Syn and *Prochlorococcus*-Pro) and critically some of their dominant cyanophages. Overall the manuscript is well-written and –presented. As the manuscript discusses, there is a paucity of measurements in the ocean on agents of mortality in general and viruses/cyanophages in particular. Matching up host Syn/Pro and phage over major ocean environmental gradients is a great contribution in and of itself. Using a modeling (regression) approach, projecting the abundance of phages and T4/T7 across the ocean basin is also notable.

But aside from the general patterns of abundance, the major focus of finding of the manuscript focuses on the role of cyanophages have in shaping the community structure at the genera level. While the manuscript presents reasonable support for this hypothesis, it does not adequately exclude or address other hypotheses. First, there are many displays of something (cyanophage, Pro, Syn, etc.) versus temperature as the putative dominant driver of cyanobacteria community structure and it is sufficiently shown that the patterns don't align with temperature. However, other variables for which there are data: salinity, POC, phosphate, and nitrate+nitrate are not shown in this manner as to exclude these as potential drivers. Mixed layer depth or stratification have been implicated as an important driver of diversity and should be shown. <https://doi.org/10.1126/science.1122692> It's certainly possible that these will show nothing, but that would only burnish the arguments in this manuscript. Figure 2 shows the patterns of Pro and Syn vs. temperature and it is compelling that Pro drops off at ~18C, where it would normally be found, but it is also true that Syn is not found in great abundance in colder waters <13C where it 'should be' suggesting that other mechanisms may also be at work.

Second, there are some larger scale differences between 2015/2016 and 2017 that deserve more discussion and quite possibly may be the proximal driver. For example, on line 237 and forward the manuscript mentions the 'record marine heatwave', that affected 2015/2016 vs. 2017, but others have shown that this 'blob' more broadly affected the phytoplankton community (e.g. <https://doi.org/10.1002/lno.11056>) beyond just cyanobacteria. This suggests that something other than just cyanophage may be influencing the community. (It's notable that the abundance of Pro in warmer temperatures is lower in 2017 than in 2015/2016. Also the mortality shown in Fig S6). The 2017 transect was also done in June, whereas the other transects were in March/April. At this latitude this temporal difference can have an impact on sunlight - something like ~2 h. That difference may be affecting the rates of different groups - perhaps something worth constraining. More broadly, from the TS plot (fig 1d) it looks like the 2017 subtropical gyre has higher T and that the transition zone (as defined by S) is hotter. This may be due to the previous points, but does suggest that there are some water mass (i.e. environmental) things going on, which could be more thoroughly explored.

Third, there is somewhat of a cause/effect thing going on here that the data/discussion mentions, but is not really fully fleshed out. For example, if the increase in % infection of Pro in 2017 (Fig 4e) is the cause of their decline (Fig 2a), what is causing the increase in % infection. Could this be some other stress? Or relief from stress or other mechanisms referenced in 197-207. Looking at Fig 4 in 2015 there may be an increase in infection at the front, in 2016 there is an increase at the front, and in 2017 there is a more dramatic increase or 'hot spot' Thus, it's less clear that there is a persistent 'hot spot' but rather an area of enhanced infectivity, that at times may become a dominant process. Given the differential time of occupation and lags associated with processes it might be good to discuss (or speculate) on the persistence or development (time) of this feature. Oceanographically, most seasonal transition zones have "memory"/hysteresis/momentum in their response so the "direct" correlations measured here do not fully capture the cause/effect.

Fourth, there is no mention of the diversity within the Pro and Syn clades, but temperature plays a key role in defining the biogeography of the various strains/clades within Pro and Syn. Further, cyanophage have been shown to specific to different types of Pro or Syn. Thus, there is almost certainly a sub-genera diversity story here that if not measured, should at least be mentioned.

Overall there is a lot of great work here that should come out. There are also some areas, which if addressed, would provide critical context and potentially further insight into the fascinating patterns presented.

Comments by line

35: This statement is misleading – while maybe true for 2017, it is not true for other years. Further, there is no real evidence that the increase in 2017 is exceptional (e.g. 2015 v 2017 in fig 2b).

95: yes this is true, but misleading since it is likely not related to the front per se

104: Yes, but as above P, N+N, salinity, mixed layer/stratification are not specifically plotted

111: abiotic conditions *that were investigated* cannot ... (e.g. metals were not investigated and have been shown to be important in this broader region)

123-135: this is interesting, but it is not until 209 -224 that context is provided. Perhaps rearrange some sections?

168: reference 10 discusses abundance distributions in the Atlantic and has lab measured growth rates – this statement should be reworded to reflect the doubling rate is estimated

208-224: consider moving ahead of the data to provide context (see above)

227: yes! And perhaps this can be expanded?

255: “accurately predicted” – please provide a quantitative (stats?) evaluation of the model

Fig 1: The TS plot is great, but given that everything else is plotted vs. latitude there should also be a temperature vs. latitude plot

Fig 2 and 4: There is pretty clearly uneven data density vs. temperature, which is fine, but it makes this type of presentation potentially misleading. Perhaps binning using box plots would help to see the broad trends. These plots should still be in the supplementary material

Fig 2-5: Unless a style of NatureMicrobiology, the titles should reflect the data presented, rather than interpretation.

Fig 3: add years to panels a-c (they are different years?). Why are viral diversity data plotted vs. latitude (in fig 5 too), but other trends (total abundance) and the data in fig 4 plotted against temperature. It's very difficult to make connections between the trends.

Fig S2 – this figure really challenges the notion that it is the viral 'hotspot' that is causing the decline in Pro. 2015 and 2016 do not support this idea. 2017 does, but if this is a driving factor then why does it not occur in 2/3 of the observations?

Fig S6 – this figure is interesting and suggests that 2017 is different across the entire transect than 2015/2016. See above for more on differentiating the 'blob' years from the 2017

I hope that these remarks can help the authors improve the manuscript.

Reviewer #2 (Remarks to the Author):

The manuscript by Carlson et al. has an intriguing conclusion from an impressive collection of cruise data hinging on the Lindell lab's "polony" method that detects specific cyanophages. The cruises had two fundamentally different tracks, from Oregon to Hawaii in March 2015 (NE-SW track), and North-South from Hawaii in April 2016 and June 2017 (the March cruise extended several thousand km East of the other two). The results are combined here and it is reported that relatively high abundances of the polony-detectable viruses near the transition from the central gyre to the subarctic gyre generally correspond to the locations where Prochlorococcus abundance transitions from high (south) to low (north) while at the same time Synechococcus does the opposite. The authors basically conclude that the transition is a virus "hot spot" where particularly high virus infection of Prochlorococcus keeps the population low and thus *causes* the distribution pattern in that entire region; the causation is implicit in the title and stated outright in multiple places. It may be right, but maybe not. It is this grand claim of causation (implying it is the main cause, as opposed to something like grazers), and also several details of the underlying data, that concern me. However, I agree the manuscript has much valuable new information on the distributions of different cyanophages in places that have not before been studied.

I have two major concerns:

1. One important thing I did not see in this or other papers on the polony method (which is crucial to the results and conclusions here), is any evidence that the method definitely detects essentially *all* the important viruses infecting cyanobacteria throughout the regions these cruises cover, i.e. there are no important ones besides the T4-like, T7-like, TIM5-like ones that match the primers used here - significantly, including variants that simply do not hit the primers well. If the method only detects some of the cyanophages and misses others, then the "hot spot," relative to places with reported low cyanophages, could be in large part an artifact of detectability. That could negate, or at least significantly blunt, the conclusions. So where is that evidence – or even data providing strong support? Frankly it is quite hard to prove, even though it is an important underlying assumption for the conclusions drawn here. Do we really know that much about cyanophages in these remote ocean locations to be sure we know them all? I think there is still a lot we don't know. As a reviewer, I do not relish giving very difficult demands to authors, but when the conclusions really depend on a critical point,

I do not see an easy solution around it. This point is out of sight in the paper but nevertheless central.

2. Important: The data in Fig 4e (and elsewhere) indeed show that there is some correspondence between where viruses are high, and low Prochlorococcus. Correlation does not mean causation - it could be incidental, or both related to an important unmeasured third parameter. How can we know it is not at least in part from flagellates that eat both Syn and Pro "causing" enough Prochlorococcus mortality to prevent their population from increasing (because they grow slower than Syn). And maybe there is something about the transition zone that leads to higher small flagellates (or higher grazing from pre-existing mixotrophs) - with a reason just as unknown as why the viruses would be higher there. After all, predators are the explanation that Lindell coauthored with Biller, Chisholm et al. in their 2015 review, which said: "Furthermore, Synechococcus strains have higher maximum growth rates than Prochlorococcus and they are prey for many of the same predators. As the growth rate of predators is coupled to that of their prey, it may be impossible for Prochlorococcus to achieve net positive growth rates when Synechococcus is growing maximally - it would simply be 'grazed away' " This concern alone is enough to raise significant doubt about the title and main conclusions of this manuscript.

Other issues:

It may be problematic using the three cruises in different months, of different years, and two very different cruise tracks, with the transition zones thousands of miles apart. The huge North Pacific Gyre is not so uniform. For example, I recall seeing results from the Zehr lab about very different cyanobacteria and nitrogen fixer communities within the gyre along a transect between Hawaii and California. March may be hardly comparable to June. Lumping the cruises and comparing together is tricky (and iffy)- as is done in lines starting 102. Discussion of things like "expansion" of Synechococcus in 2017 (line 113, line 176) seems to be based on little data. "Expansion" related to what - expectations from another year? It all seems a little sloppy. And sometimes only 2 of the three years are referred to, e.g. line 154. At least more caveats are called for.

Perspective: The "massive decline" in Prochlorococcus (line 167), ascribed to viruses (the main point of the paper) seems to imply they were there and then disappeared over some short time (and were replaced by the "expansion" of Synechococcus). But there is no evidence of that here. It is a "massive decline" only from the perspective of a ship traveling North on their transect, quite an artificial perspective (it can be called a "massive increase" just as well, traveling South). Otherwise it is just like any distribution of an organism which is higher in some places and lower elsewhere, sometimes at a sharp boundary (like between gyres, or near the equator). The local history can only be guessed, without multiple measurements in a season. The causes are often very complex. Here the authors seem to look only at temperature, nutrients and viruses, and try to pin the blame completely in viruses. That is a narrow view, considering they did not even look at grazers.

I see that on line 169, the authors calculate viral induced Prochlorococcus mortality (with many underlying assumptions!), but don't show the corresponding calculation for Synechococcus mortality - is it consistent with the results/conclusions? I would think differential virus effects should be compared. One could imagine Syn might have an even higher mortality from viruses there, being more abundant, larger and growing faster.

Reviewer #3 (Remarks to the Author):

Thank you for the opportunity to review this paper. I have been asked by the editor to help in assessing the mathematical modeling aspects of the study only. As I am not a marine microbiologist expert, I will limit myself to that and ask for forgiveness if my comments are basic.

My two questions are about the choice of predictors for the multiple regression model, and about the use of a multiple regression model for predictions.

Based on model root mean square error, the authors choose chlorophyll and temperature as predictors

in their multiple regression model. At the same time, I understand that viruses abundance depends on the availability of bacteria hosts (line 47-49 or the manuscript). If that is the case, one would expect that bacteria abundance would be a better predictor. Indeed, the authors seem to say so on line 470-472. If chlorophyll can be used as a proxy for bacteria, this would be a better argument for the authors' choice of predictor than similar RMSE values.

My second question is about the use of a regression model itself. My understanding is that the authors argue for a balance between viruses and bacteria to be important, with the viruses depending on the availability of bacteria to reproduce, and the bacteria dying because of virus infection (lines 44-51). With the goal of making predictions, a population model, including abiotic forcing, would seem more appropriate than a regression model. A regression model neglects the interaction between the bacteria and viruses. That would be the case even if bacteria abundances were used as a predictor, as it would still be an independent variable.

Please let me stress again that I might be missing some very fundamental point that may be obvious to any microbiologist, and take my comments as a layman questions more than anything else.

Thank you for the opportunity to review this paper, I have learned a lot.

Best regards

Author Rebuttal to Initial comments

13 August 2021

Dear {redacted},

Thank you for detailing the main reasons for your decision on our manuscript though we are dismayed by the decision and many of the reviewers' comments. Most of the major concerns raised by the reviewers are based on significant misconceptions about the methods and aims while others seem completely reasonable to address in the text.

First, methodological concerns about primer biases and detection capabilities of the polony method raised by reviewer 2 have already been extensively addressed in previous manuscripts. Baran et al 2018, (*Nat. Microbiol.*) and Goldin et al. 2020 (*Front. Microbiol.*) both show that primers and probes, which were designed using both isolate and environmental sequences, capture environmental virus genotypes across the diversity of the cyanophage lineages examined in each paper and applied in this current study. Furthermore, both Baran et al. 2018 (*Nat. Microbiol.*) and Mwurur, Carlson et al. 2021 (*ISME J*) bioinformatically verified that these primers and probes matched cyanophage sequences from metagenomic datasets. Specifically, Mwurur, Carlson et al. 2021 (*ISME J*) compared polony primers and probes to the diversity of cyanophages in 44 metagenomic samples collected in the North Pacific Subtropical Gyre in 2015 and found that 93% of cyanophage reads would be detected. It is notable that the relative cyanophage community composition in the metagenomes was similar to that using absolute

quantification with the polony method. Also note that this cruise was just 4 months after one of the cruises presented in this manuscript. We strongly feel that our manuscript should not be rejected or faulted due to misconceptions and clear lack of thorough knowledge of the work done to validate the polony method shown by Reviewer 2. Despite already having published such methodological verifications in previous manuscripts, if the Nature Microbiology editors think it necessary, we could evaluate the extent to which our primers and probes match the sequence diversity of cyanophages across these transects to definitively address this issue for the current study, as we have sequenced metagenomes that were collected in tandem with the polony samples presented in this manuscript.

Second, Reviewer 3 did not understand our aim for the modeling which was to predict virus abundances over large ocean expanses. We reported the strong correlation between virus abundance and infected cell abundances in the manuscript and agree that developing models with cyanobacterial abundances would be the way to go if these data were as available at anywhere near the resolution of chlorophyll data. Cyanobacterial abundances can be measured only from samples collected during a ship visit to a particular site at a particular point in time. In contrast, chlorophyll concentrations are measured by satellite and are thus collected remotely and continuously at very high spatial and temporal resolution, and the data are freely available. It is important to mention that chlorophyll is directly linked to the abundances of photosynthetic organisms which includes the cyanobacteria (Bouman et al. 2012 *Science*). In reading this section of our manuscript again, we acknowledge that the reasons for using satellite derived chlorophyll data and not cyanobacterial counts were not as clearly laid out as needed for those less familiar with large-scale ecology or oceanography. This issue can be easily clarified in a revised manuscript.

Third, the repeated sampling of the transition zone from multiple cruises strengthens the ability to determine which of the phenomena reported are widespread and recurrent in the North Pacific Ocean, such as the virus hotspot we present in this manuscript. It is shocking that reproducibility at such a wide range of times scales (days, weeks, months, years) and spatial scales (km to thousands of km) could be considered 'problematic' as stated by reviewer 2. If sampling done at different times and places cannot be compared, then no generalizable ecosystem features would ever be discovered. In fact, only through the collection of data on multiple cruises over diverse sets of environmental conditions and in different seasons have the determinants of phytoplankton distributions and abundances been clarified as in Ustick et al. 2021 (*Science*), Flombaum et al. 2013 (*PNAS*) etc. Additionally, this reviewer's criticisms of our 'lumping' of cruises together are clearly misguided. We plainly show each cruise separately and then compare the similarities and differences between the transects. In fact, only through such comparisons did we identify that the decline of *Prochlorococcus* at temperatures considerably higher than expected was accompanied by high virus infection on one of the cruises, and that this was a departure from the norm in relation to the other cruises where *Prochlorococcus* declined at expected temperatures and viral infection was low. Thus, such multi-cruise comparison is as an advantage and not a problem for both identifying recurrent phenomena and those that are unusual. We agree, however, that textual clarifications are likely needed to bring these points out more.

Fourth, our work is not simply based on abundance correlations between cells and viruses as implied by reviewer 2. While these correlations exist, we specifically measured direct cell infection which could not be achieved prior to our newly developed method (Mwurat, Carlson et al. 2021 *ISME J*) and is a key novelty of this study. Because infection by lytic viruses results in cell death, this can be used to determine the direct cause of mortality. Our assertions are based both on the magnitudes of infection and the corresponding patterns between increased infection and decreased cyanobacteria abundances. We agree entirely that mortality in environmental systems is the sum of many factors including viral lysis, grazing, and abiotic factors which we mention several times throughout the manuscript (lines 134, 170, 301). We show that viruses contributed substantially to *Prochlorococcus* mortality (up to 51%) and propose that this high level of virus-induced mortality, on top of the “normal” mortality levels caused by multiple factors that usually maintain the steady state between growth and death, is what brought about the sharp decline in *Prochlorococcus* populations in 2017 (see lines 170-172). We agree that grazing could also be involved in the “additional” mortality and the unusual decline in 2017. Nonetheless, this does not take away from our finding of the clear involvement of viral infection in this phenomenon. We acknowledge that the text didn’t reflect the possible involvement of grazing with sufficient emphasis and agree that we need to give it more weight in the text and to also modify the title to better reflect this as suggested by the reviewer.

Finally, we agree with reviewer 1 that it would be useful to present comparisons between the abundances of cyanobacteria and cyanophages and other abiotic parameters as a means to show that other factors were not involved in the shifting distributions. These were examined but were not significant. We will now include these comparisons to exclude these as potential drivers and to more clearly justify our conclusions, as suggested by reviewer 1.

Given that most of the major criticisms of the manuscript are due to significant misconceptions by reviewers and that the remaining comments are well within reason to address in the text, we ask you to allow us to respond fully to all of the criticisms raised by the reviewers and to submit a revision of our manuscript. Thank you very much for your consideration on this matter.

Sincerely,

Debbie Lindell and Michael Carlson

Decision Letter, first revision:

20th August 2021

Dear Debbie,

Thank you for your letter asking us to reconsider our decision on your Article entitled "A virus hotspot at ocean gyre boundaries limits the geographic range of Prochlorococcus". After careful consideration we have decided that we would be willing to consider a revised version of your manuscript.

Editorially, we feel it will be important to add an evaluation of the extent to which our primers and probes match the sequence diversity of cyanophages across the transects.

Along with your revised manuscript, you should also submit a separate point-by-point response to all of the concerns raised by the referees, in each case describing what changes have been made to the manuscript or, alternatively, if no action has been taken, providing a compelling argument for why that is the case. If we feel that a substantial attempt has been made to address the referees' comments, this response will be sent back to the referees - along with the revised manuscript - so that they can judge whether their concerns have been addressed satisfactorily or otherwise.

I should stress, however, that we would be reluctant to engage our referees again unless we thought that their comments had been addressed in full.

- ensure it complies with our format requirements for Letters as set out in our guide to authors at www.nature.com/nmicrobiol/authors/index.html

- state in a cover note the length of the text, methods and legends; the number of references and the number of display items.

Please ensure that all correspondence is marked with your Nature Microbiology reference number in the subject line.

Please use the following link to submit your revised manuscript:

{redacted}

We hope to receive your revised paper within four weeks. If you cannot send it within this time, please let us know so that we can close your file. In this event, we will still be happy to reconsider your paper at a later date so long as nothing similar has been accepted for publication at Nature Microbiology or published elsewhere in the meantime. Should you miss the four-week deadline and your paper is eventually published, the received date will be that of the revised, not the original, version.

I would appreciate it if you could tell me if you think you will be able to submit a revised manuscript, and also the likely timescale.

I look forward to hearing from you soon.

Yours sincerely,
{redacted}

Author Rebuttal, first revision:

Dear {redacted}

We would like to thank you for giving us the opportunity to respond to the reviewers' comments and submit a revised version of the manuscript. As requested by the editor and one of the reviewers, we have analyzed viral metagenomes collected at high-resolution across the 2016, 2017, and 2019 transects to assess the ability of the primers and probes used in polony methods to capture the sequence diversity of cyanophages. We now report that our methods match >93% of assembled cyanophage sequences in these metagenomes in the Methods and provide alignments used in this evaluation as supplementary material. We have added additional supplemental figures and

significantly expanded the text addressing alternative hypotheses, which show that none of the abiotic factors measured could explain the unexpected decline in *Prochlorococcus*. Additionally, we now point out that the loss of *Prochlorococcus* but a lack of decline in total bacteria suggest a mortality process specific to picocyanobacteria, providing further support for viral infection as a major cause. We have better emphasized and discussed the potential role of mortality by grazers and other factors in addition to the significant virus-mediated mortality. We have reworked the text to clarify and bolster comparisons from the multiple cruises presented here and to previous work on picocyanobacterial distributions in the oceans. Finally, we have outlined the aims of our modeling more clearly. Our point-by-point responses are outlined below and we provide the revised manuscript with changes shown as an additional file for the reviewers.

We thank the reviewers for their comments. The revisions made to address the comments have significantly improved the manuscript. We hope that this revised manuscript now meets the high standards of *Nature Microbiology* and will be accepted for publication.

Sincerely,

Debbie Lindell and Michael Carlson

*** Reviewer

Expertise:

Referee #1: Marine cyanobacteria, cyanophage ecology

Referee #2: Phage identification and quantification methods/polony method Referee

#3: Mathematical Modelling

Reviewers Comments:

Reviewer #1 (Remarks to the Author):

This is a review of "A virus hotspot at ocean gyre boundaries limits the geographic range of *Prochlorococcus*" (NMICROBIOL-21061597) The manuscript presents data from multiple occupations of an area in the North Pacific Ocean and shows the patterns of cyanobacteria (*Synechococcus*-Syn and *Prochlorococcus*-Pro) and critically some of their dominant

cyanophages. Overall the manuscript is well-written and –presented. As the manuscript discusses, there is a paucity of measurements in the ocean on agents of mortality in general and viruses/cyanophages in particular. Matching up host Syn/Pro

and phage over major ocean environmental gradients is a great contribution in and of itself. Using a modeling (regression) approach, projecting the abundance of phages and T4/T7 across the ocean basin is also notable.

We thank the reviewer for commending both the contribution of the science and the quality of the manuscript and for the constructive criticisms and suggestions presented below.

But aside from the general patterns of abundance, the major focus of finding of the manuscript focuses on the role of cyanophages have in shaping the community structure at the genera level. While the manuscript presents reasonable support for this hypothesis, it does not adequately exclude or address other hypotheses. First, there are many displays of something (cyanophage, Pro, Syn, etc.) versus temperature as the putative dominant driver of cyanobacteria community structure and it is sufficiently shown that the patterns don't align with temperature. However, other variables for which there are data: salinity, POC, phosphate, and nitrate+nitrite are not shown in this manner as to exclude these as potential drivers. Mixed layer depth or stratification have been implicated as an important driver of diversity and should be

shown. <https://doi.org/10.1126/science.1122692> It's certainly possible that these will show nothing, but that would only burnish the arguments in this manuscript.

We agree with the reviewer that adding such comparisons will help illustrate that other abiotic factors could be excluded as responsible for the cyanobacterial patterns in 2017. We had previously carried out many of the analyses but did not present them as figures. We have now included a supplemental figure (Figure S3) showing *Prochlorococcus* and *Synechococcus* abundances versus these other variables (nitrate+nitrite, phosphate, salinity, mixed layer depth and particulate carbon). This clearly shows that *Prochlorococcus* populations thrived across a range of phosphate, nitrate+nitrite concentrations in the 2015 and 2016 transects but declined in 2017 at relatively high nutrient levels at which they normally thrive in and are thus cannot be considered the cause of their decline, similar to what we saw with temperature. *Prochlorococcus* populations in 2017 declined at similar values of salinity and POC compared to observations on the 2016 transect, whereas *Prochlorococcus* populations thrived across a wider range of values for these variables in 2015. Thus, neither of these factors appears to be differentially influencing the 2017 distributions of *Prochlorococcus*, nor have either of them been implicated previously as important in limiting picocyanobacterial populations.

We have also calculated the mixed layer depth from CTD cast depth profiles from the 2016 and 2017 transects and from the March climatology based on Argo float profiles for the 2015 cruise, since no shipboard data was taken to assess MLD on this transect. MLD could not explain the 2017 decline in *Prochlorococcus*. We added these comparisons to the supplemental figure (Fig. S3).

We have added a more explicit discussion of the testing and exclusion of these alternative hypotheses to the main text in addition to the supplemental figure.

Fig. S3. Comparison of picocyanobacterial abundances to abiotic factors. *Prochlorococcus* (left) and *Synechococcus* (right) abundances plotted against concentrations of phosphate (a,b), nitrate+nitrite (c,d), particulate carbon (e,f), salinity (g,h), and mixed layer depth (i,j) for the March 2015 (purple), April 2016 (blue), and June 2017 (orange) transects.

Main text (lines 110-122):

A suite of abiotic variables beyond temperature, many of which are considered important determinants for *Prochlorococcus*' biogeography^{11-16,34}, were assessed for their potential role in restricting the geographic distribution of *Prochlorococcus* on the 2017 transect (Fig. S3).

Macronutrient (phosphate and nitrite+nitrate, Fig. S3a,c) and micronutrient (iron²⁷)

concentrations were within the range for *Prochlorococcus*' optimal growth³⁵, and lead (Pb) concentrations²⁷ were below levels toxic to *Prochlorococcus*³⁶. *Prochlorococcus* populations in 2017 declined at similar values of salinity (Fig. S3e) and particulate carbon (Fig. S3g) as the 2016 transect, while thriving across a wider range of values for these variables in 2015.

Furthermore, a mixed layer depth threshold was not associated with the decline in *Prochlorococcus* (Fig. S3i). Collectively, the abiotic conditions observed in 2017 in the region of the *Prochlorococcus* decline, supported large populations of *Prochlorococcus* in 2015 and 2016 (Fig. S3). Thus, none of the physical or chemical factors investigated here can explain the unexpected decline in *Prochlorococcus* in 2017.

Figure 2 shows the patterns of Pro and Syn vs. temperature and it is compelling that Pro drops off at ~18C, where it would normally be found, but it is also true that Syn is not found in great abundance in colder waters <13C where it 'should be' suggesting that other mechanisms may also be at work.

We interpret the decline in *Synechococcus* abundances at the northern end of the transect differently than the reviewer. While we agree that there is a temperature difference in their decline, being somewhat higher in 2017 than in 2015 and 2016 (~2 degrees C for *Synechococcus*, compared to ~5 degrees C for *Prochlorococcus*), *Synechococcus* distribution patterns appear to be more related to other phytoplankton than directly to temperature. This may be due to competition or conditions that lead to spatial succession between them. First, for all three cruises the northern increase in *Synechococcus* is positioned "after" the decline in *Prochlorococcus* as we move north (Fig. S2 of the manuscript). Second, the subsequent northern decline at the northern edge of the transect occurred on all three cruises when picoeukaryotes reached ~20,000 cells ml⁻¹ (see below figure and now added to Fig. S2 in the revised manuscript).

This suggests that competition with picoeukaryotes may be a factor in the northern decline of *Synechococcus* rather than temperature, or that other environmental conditions are responsible for the switch between *Synechococcus* and the picoeukaryotes in this region. It is important to note that temperature is considered to be much less of a driver for *Synechococcus* distribution patterns than for *Prochlorococcus*, at the genus level, at least at the temperature ranges

relevant here (Zinser et al. 2007 *Limnol. Oceanogr.*, Zwirgmaier et al. 2007, 2008 *Environ. Microbiol.*,

Flombaum et al. 2013 *PNAS*, Pittera et al. 2014 *ISME J*). Note also that *Synechococcus* abundances declined but they were still present at $1-2 \times 10^4$ cells ml^{-1} at a temperature of 13 degrees C and below (in comparison to the drop in *Prochlorococcus* to ~ 2000 cells ml^{-1}). We have added a discussion on this in the Supplementary Discussion to clarify and differentiate this pattern more explicitly.

Supplementary Discussion:

Biotic and abiotic controls on *Synechococcus*' distribution

Synechococcus populations decreased at warmer temperatures in June 2017 compared to the 2015 and 2016 transects (Fig. 2b). However, the shift in the 2017 decline occurred with a ~ 2 °C difference relative to the previous years, whereas the shift observed for the 2017 *Prochlorococcus* decline had a ~ 5 °C difference. The northernmost decline in *Synechococcus* corresponded to the region where picoeukaryotes reached $\sim 2 \times 10^4$ cells $\cdot \text{ml}^{-1}$ in all three cruises (Fig. S2), suggesting that competition with picoeukaryotes, or as yet unknown abiotic conditions that lead to the spatial succession between them, are responsible for this decline. It is important to note that temperature is a less significant driver of the overall distribution range of the *Synechococcus* genus than for the *Prochlorococcus* genus^{11,12,46,90}, at least at the temperature ranges relevant for this region.

Second, there are some larger scale differences between 2015/2016 and 2017 that deserve more discussion and quite possibly may be the proximal driver. For example, on line 237 and forward the manuscript mentions the 'record marine heatwave', that affected 2015/2016 vs. 2017, but others have shown that this 'blob' more broadly affected the phytoplankton community (e.g. <https://doi.org/10.1002/lno.11056>) beyond just cyanobacteria. This suggests that something other than just cyanophage may be influencing the community. (It's notable that the abundance of Pro in warmer temperatures is lower in 2017 than in 2015/2016. Also the mortality shown in Fig S6). The 2017 transect was also done in June, whereas the other transects were in March/April. At this latitude this temporal difference can have an impact on sunlight - something like ~ 2

h. That difference may be affecting the rates of different groups - perhaps something worth constraining. More broadly, from the TS plot (fig 1d) it looks like the 2017 subtropical gyre has higher T and that the transition zone (as defined by S) is hotter. This may be due to the previous points, but does suggest that there are some water mass (i.e. environmental) things going on, which could be more thoroughly explored.

We fully agree that these large-scale climate dynamics are likely to play a role in impacting phytoplankton dynamics as noted in the previous version of the manuscript on lines 191-193. We found higher *Prochlorococcus* abundances in 2015 and 2016 than in 2017 across the transect (Fig. 2a, Fig. S2), in line with findings by Pena et al. 2018 *Limnol. Oceanogr.* in the

Pacific Ocean (the manuscript mentioned by the reviewer) and Larkin et al. 2020 *Plos One* in the Southern California Bight during the heatwave.

However, the drop in *Prochlorococcus* abundances from $\sim 130,000 \text{ ml}^{-1}$ to $\sim 2000 \text{ ml}^{-1}$ at 18 degrees C in 2017 after the heatwave is well beyond the consistently lower abundances found in other years at this temperature. Furthermore, in many previous years with no heatwave, *Prochlorococcus* distribution patterns were still limited by temperatures at the lower bounds of 12-13 degrees C (Bouman et al. 2006 *Science*, Johnson et al. 2006 *Science*, Zinser et al. 2007 *Limnol. Oceanog.*, Flombaum et al. 2013 *PNAS*, Larkin et al. 2016 *ISME J*) and not the 17-18 degrees C found in this study in 2017. Thus, the lack of a heatwave cannot be considered the proximal reason for the decline of *Prochlorococcus* at higher temperatures than normal. Rather, these data implicate mortality as the reason for the decline, and the high percentage of cyanophage infection points to this being a proximal cause of the *Prochlorococcus* decline in 2017.

Regarding season, in June there would be more sunlight and therefore a longer period of time for phytoplankton to grow, so we'd expect the opposite trend of more *Prochlorococcus* in June than in March/April if it was the proximal cause for the differences.

Having said that, we agree that it is feasible that either large-scale climate dynamics or season resulted in higher infection in 2017 and may thus be the ultimate cause(s) rather than the proximal cause(s) in the unusual *Prochlorococcus* decline. This is the argument we presented briefly in the previous version of the manuscript (on lines 233-243). We agree with the reviewer regarding the higher temperatures in the subtropical gyre and transition zone in 2017, as seen from the TS plot, and also agree that these are likely related to the points raised of climate dynamics and season. We now clarify and strengthen these points in the revised version of the manuscript (lines 268-286 see below).

We also agree that day length may be important in altering the patterns observed at different times of year. Day lengths in March 2015, April 2016, and June 2017 at 35 degrees N (a latitude where the transition zone overlapped in all three cruises) were

~ 12.5 , 13.5 , and 14.5 h long respectively. We had previously incorporated a correction for the number of infection cycles that cyanophages could complete per day based on the day length (line 438). For picocyanobacteria, day length was found to be positively correlated to the relative abundance of the *Prochlorococcus* HL I.2 ecotype in Larkin et al. 2020 *ISME J*. Furthermore, the division rates used in calculations in this manuscript were based on cultures grown on a 14:10 h light:dark cycle. Thus, we do not expect day length to affect our calculations much for the 2017 transect.

In summary, we discussed these points briefly in the previous version of the manuscript and now expand this discussion. We describe in the main text that these large-scale

climate patterns are more likely to impact *Prochlorococcus* mortality through virus infection or grazing than directly affecting their growth.

Main text (lines 268-286):

Despite consistent features in cyanophage distributions across the North Pacific Ocean, cyanophage infection was higher (Fig. 4, Fig. S7), while *Prochlorococcus* abundances were consistently lower (Fig. 2a), across the June 2017 transect relative to the March 2015 and April 2016 transects. Seasonality and/or climate variability could explain this interannual variability, although the data currently available to assess this are sparse. Viral infection of picocyanobacteria in the subtropical gyre increased from early spring to summer suggesting a potential seasonal pattern that may extend across the transect (Fig. S9a). In addition, the June 2017 transect occurred during a neutral to negative El Niño phase with lower sea surface temperatures relative to the 2015 and 2016 transects which were in years of a record marine heatwave, followed by a strong El Niño⁵⁵ (Fig. S9b). In 2015 and 2016, *Prochlorococcus* abundances were found to be higher than usual in the North Pacific Ocean in this (Fig. 2a) and in other studies^{56,57}. Irrespective of the underlying drivers for the observed interannual variability, we speculate that an ecosystem tipping point was reached in the prevailing conditions in June 2017. In this scenario, picocyanobacterial populations were subjected to high infection levels that resulted in an accumulation of cyanophages, initiating a stronger than usual positive feedback loop between infection and virus production and precipitating the unexpected *Prochlorococcus* decline. Continued observations in the North Pacific Ocean are needed to evaluate the potential link between seasonality and/or large-scale climate forcing as ultimate drivers affecting virus-host interactions.

Third, there is somewhat of a cause/effect thing going on here that the data/discussion mentions, but is not really fully fleshed out. For example, if the increase in % infection of Pro in 2017 (Fig 4e) is the cause of their decline (Fig 2a), what is causing the increase in

% infection. Could this be some other stress? Or relief from stress or other mechanisms referenced in 197-207.

We agree with the reviewer that it seems likely that some underlying mechanism facilitated enhanced infection. We discussed two possibilities which were related in the increased availability of nutrients and decreased decay of infectivity at colder temperatures in the previous version. We have taken the reviewer's suggestion and added further discussion about stress and other mechanisms that may be at play in the main text.

Main text (lines 215-232)

The cyanophage hotspot in the transition zone is a ridge of high virus activity that separates the subtropical and subpolar gyres. The reproducibility of our observations, which were separated by days to weeks within cruises (2016 and 2017) and by years among the three cruises (Fig. S4),

indicates that this virus hotspot is a recurrent feature at the boundary of these two major gyres in the North Pacific Ocean. This suggests that the hotspot forms due to the distinctive environment

of the inter-gyre transition zone creating conditions that enhance infection of picocyanobacteria and proliferation of cyanophages. *Prochlorococcus* in the transition zone may be prone to stress due to existing near the edge of their temperature range^{5,6}, which has the potential to increase susceptibility to viral infection. Alternatively, there may be temperature-dependent tradeoffs between virus decay and production that lead to replication optima within a narrow temperature range⁴⁸. Cyanophage infectivity has been observed to decay more slowly at colder temperatures⁴⁹ which may allow for the accumulation of infective viruses leading to increased infection.

Additionally, cyanophages may have more productive infections due to enhanced nutrient supply in the transition zone²⁷ (Fig. 1h, i) relative to the subtropics since the cyanophages replicate in hosts with presumably greater intracellular nutrient quota and obtain more extracellular nutrients, both of which may increase progeny production^{9,10}. Thus, a putative cyanophage replication optimum may reflect combined effects of temperature and nutrient conditions that are intrinsically linked to the oceanographic forces that shape the transition zone itself.

Looking at Fig 4 in 2015 there may be an increase in infection at the front, in 2016 there is an increase at the front, and in 2017 there is a more dramatic increase or ‘hot spot’ Thus, it’s less clear that there is a persistent ‘hot spot’ but rather an area of enhanced infectivity, that at times may become a dominant process. Given the differential time of occupation and lags associated with processes it might be good to discuss (or speculate) on the persistence or development (time) of this feature. Oceanographically, most seasonal transition zones have “memory”/hysteresis/momentum in their response so the “direct” correlations measured here do not fully capture the cause/effect.

We agree with the reviewer that the magnitude of infectivity or host susceptibility is variable over the different years. To be clear, we define the hotspot based on cyanophage abundances (Fig. 3 in original submission and Fig. 2c in the revised manuscript). We have explicitly stated this in the revised manuscript for clarity, “Within the transition zone, we observed a steep latitudinal increase in abundances of cyanophages each year, which we define as a cyanophage hotspot (Fig. 2c, Fig. S2, S4).” (lines 153-154). This is indeed a very interesting aspect of our findings. Our out-and-back crossings show the hotspot persists for at least the time scale of weeks. The repeated yearly observations suggest that the hotspot is recurrent at least for the spring and summer. Finally, our previous work in the North Pacific Subtropical Gyre estimated the turnover time of the cyanophage population to be at least a month based on calculations of how many infection cycles would be needed to replace the free-living virus pool (Mruwat, Carlson et al 2021), indicating this hotspot may extend beyond the seasons of sampling.

Unfortunately, we do not have sufficient data yet to provide a justifiable discussion on the period of development, persistence and decline of such a feature. It is for this reason that we describe the feature as recurrent in the manuscript rather than persistent. While we fully agree with the reviewer that this is of great interest, and wished we had sufficient data to justify such a discussion, we feel that such a discussion would be too speculative based on the current dataset.

Fourth, there is no mention of the diversity within the Pro and Syn clades, but temperature plays a key role in defining the biogeography of the various strains/clades within Pro and Syn. Further, cyanophage have been shown to specific to different types of Pro or Syn. Thus, there is almost certainly a sub-genera diversity story here that if not measured, should at least be mentioned.

We agree that there may be a diversity story that underpins our population-level observations. We briefly mentioned the potential role of changing cyanobacterial community composition in the previous manuscript. As requested by the reviewer, we have now expanded this to include a discussion on how such changes could affect viral abundance and infection patterns. We now also provide more details of the expected picocyanobacteria diversity changes in the Supplemental Discussion. This is quite a speculative discussion, however, and direct testing of these hypotheses awaits further investigation. In addition, it is important to note that the specificity of cyanophages is not at the ecotype level, but rather at the genotype level, often related to cell-surface properties or resistance mechanisms that often vary within an ecotype/clade (Avrani et al 2011 *Nature*, Zborowsky & Lindell 2019 *PNAS*).

Main text (lines 255-266):

Changes in temperature and nutrients occurring in the transition zone are expected to result in shifts in picocyanobacterial diversity at the sub-genus level (see Supplementary Discussion) which we speculate may affect community susceptibility to viral infection. One mechanism for this may be that the picocyanobacteria that thrive in the transition zone are intrinsically more susceptible to viral infection. Another scenario may be related to tradeoffs associated with the evolution of resistance to viral infection. The horizontal advection of nutrient-rich waters to the transition zone²⁸ may select for rapidly growing cells adapted for efficient resource utilization. Viral resistance in picocyanobacteria often incurs the cost of reduced growth rates^{53,54}. Thus, competition for nutrients in this region may favor cells with faster growth rates, but with increased susceptibility to viral infection. Thus, it is likely that cyanophage distributions do not always follow cyanobacterial patterns (Fig. S2) because of complex interactions between lineage-specific cyanophage traits, host community structure, and environmental variables.

Supplementary Text:

Expected shifts in picocyanobacterial diversity along environmental gradients

Picocyanobacterial community composition is expected to undergo changes in the vicinity of the transition zone. While, the high light (HL) II *Prochlorococcus* ecotype dominates in the subtropics^{12-15,30,33}, the HLI ecotype is expected to become the most abundant *Prochlorococcus* ecotype due to higher growth rates at the temperature range of 16-18 °C found in the transition

zone^{12,15,30}. There may also be changes in *Prochlorococcus* diversity within an ecotype, such as the enrichment of *Prochlorococcus* HLI.2 ecotypes during summer months³⁰. *Synechococcus*

clades are likely to undergo similar reorganizations in their community structure in the transition zone. The oligotrophic specialist *Synechococcus* clades II and III are expected to be succeeded by the cold-water adapted clades I and/or IV at ~15 °C based on previous findings for the Pacific and Atlantic Oceans³³. Additionally, the low iron adapted clade CRD1 has been previously observed to thrive in the North Pacific inter-gyre transition zone region³³.

Overall there is a lot of great work here that should come out. There are also some areas, which if addressed, would provide critical context and potentially further insight into the fascinating patterns presented.

We thank the reviewer for their encouraging comments about this manuscript. Comments by line 35: This statement is misleading – while maybe true for 2017, it is not true for other years. Further, there is no real evidence that the increase in 2017 is exceptional (e.g. 2015 v 2017 in fig 2b).

We have qualified this sentence by adding “At these times” so that it is clear that the extended reach of *Synechococcus* applies to the times where *Prochlorococcus* distribution was limited (i.e., up and back in 2017). We also toned down the statement by exchanging the word “expanded” to “were found” in the abstract on lines 34-35.

“At these times, large *Synechococcus* populations were found in waters typically dominated by *Prochlorococcus*.”

We agree with the reviewer that in terms of maximal abundances, *Synechococcus* populations in 2017 were not significantly higher than in 2015. However, our statement refers to the latitudinal breadth of the increased *Synechococcus* abundances in 2017 which was much broader than in 2015 and 2016 as well as relative to 9 other cruises reported in Gainer et al. 2017 Plos One, Juranek et al. 2012 *J. Geophys. Res.*, Sohm et al. 2015 *ISME J*, which we now clarify in the main text. Additionally, integrated abundances across this region in 2017 were significantly higher than in 2015. This has been clarified in the text and these additional citations and integrated abundance calculations have been added to the manuscript.

Main text (line 96-101):

In contrast, the geographic range of *Synechococcus* was broader by ~3 degrees of latitude (~330 km) and their integrated abundances across the transect were 2-fold higher than observed in 2015 and 2016 (Fig. 2b, Fig. S2). Thus, a more southern decline in *Prochlorococcus* and a broader distribution of *Synechococcus* (Supplementary Text) was observed in 2017 relative to the 2015

and 2016 transects (Fig. 2, Fig S2) and to 9 previous transects across this transition zone conducted in different years and seasons³¹⁻³³.

95: yes this is true, but misleading since it is likely not related to the front per se

We agree that it is not related to the front and make this exact argument later. Our aim here is describe the distribution pattern of *Prochlorococcus* and point out that it is different between the years. We use the chlorophyll front as this is an historical marker for the transition zone (Polovina et al. 2001 *Progr. Oceanogr.*) and an indicator that the decline in 2017 wasn't in the same relative position over the three cruises with respect to the trophic status of the water column, rather than using latitude which is more arbitrary with respect to water column conditions due to shifts related to season and climate.

104: Yes, but as above P, N+N, salinity, mixed layer/stratification are not specifically plotted

We have now plotted abundances versus abiotic parameters as suggested and now mention this in the main and supplementary text and refer the reader to the new figure showing the plots (Figure S3). See response above.

111: abiotic conditions *that were investigated* cannot ... (e.g. metals were not investigated and have been shown to be important in this broader region)

We have added this qualification on line 121 as requested by the reviewer. We also note that metal concentrations were quantified by colleagues on the 2016 and 2017 cruises. The majority of metal concentrations were similar between the 2016 and 2017 cruises, with the exception of cadmium which was lower in 2017, and is toxic at high levels to *Prochlorococcus* (Echeveste et al. 2012 *Environ. Tox. Chem.*).

{redacted}

.

Iron, manganese, and lead concentrations have been published

for the 2017 cruise which we cite (Pinedo-Gonzalez et al. 2020 *PNAS*). At this time, iron concentrations were well within the range for optimal growth by *Prochlorococcus* (Thompson et al. 2011 *ISME J*) and lead was below levels observed to be toxic/inhibitory to *Prochlorococcus*

(Echeveste et al. 2012 *Environ. Tox. Chem.*). We have added a more explicit statement regarding this to the manuscript and refer the reader to the published data for the 2017 cruise.

Main text (lines 113-115):

Macronutrient (phosphate and nitrite+nitrate, Fig. S3a,c) and micronutrient (iron²⁷) concentrations were within the range for *Prochlorococcus*' optimal growth³⁵, and lead (Pb) concentrations²⁷ were below levels toxic to *Prochlorococcus*³⁶.

123-135: this is interesting, but it is not until 209 -224 that context is provided. Perhaps rearrange some sections?

We understand the reviewer's desire for context at this position in the manuscript. However, we think that this section is most appropriate as a discussion to explain differences in cyanophage community composition across ocean regimes. The text on lines 123-153 (previous version) discusses the findings for one of these regimes (the subtropical gyre), while the discussion on 209-224 (previous version) makes the comparison between the different regimes (subtropical, transition zone, and subpolar). Thus, it relies on the prior reporting of the community composition in all three regimes and not for just one regime.

For these reasons, we prefer that this paragraph remains after the presentation of the data from all three regions to serve as a discussion for the cumulative at the end of the section. We note that some context for this is provided in the introduction which we have now modified somewhat to more directly suggest that lineage-specific traits may influence abundances under different environmental conditions. We also now state "(see below)" on line 178, after the presentation of the results for the transition zone to make it clear to the reader that a discussion on this is coming.

Main text (lines 58-62):

Picocyanobacteria are infected by several lineages of viruses belonging to the order *Caudovirales*¹⁸⁻²⁰. Each lineage has distinct traits with infection characteristics that lie on a spectrum in virulence and host range and differ in the core replication and morphogenesis genes they encode, as well as in the genes captured from their hosts^{10,18,19,21-24}. Such traits affect their fitness and presumably influence their abundance under different environmental conditions⁹.

168: reference 10 discusses abundance distributions in the Atlantic and has lab measured growth rates – this statement should be reworded to reflect the doubling rate is estimated

We have modified this sentence accordingly (lines 186-188):

Since *Prochlorococcus* is estimated to double every 2.8 ± 0.8 days at the low temperatures in this region¹², we estimate that between 21-51% of the population was infected and killed in the interval prior to cell division.

208-224: consider moving ahead of the data to provide context (see above) Please

see response to the above comment.

227: yes! And perhaps this can be expanded?

As requested by the reviewer here and in a previous comment, we have now expanded the discussion on how cyanobacterial clade diversity could affect cyanophage abundance and infection (lines 255-266 see above). The sentence mentioned by the reviewer now comes after the discussion on how phage lineage-specific differences and cyanobacterial community composition may be driving their different distributions.

255: “accurately predicted” – please provide a quantitative (stats?) evaluation of the model

Table S2 shows the Root Mean Squared Error of our observations compared to the model. We apologize for forgetting to cite the table here to provide the appropriate statistical justification for our statement. We have now added this citation (line 296).

Fig 1: The TS plot is great, but given that everything else is plotted vs. latitude there should also be a temperature vs. latitude plot

Temperature and salinity have now been plotted vs latitude in Figure 1 to address this comment (see Fig. 1e).

Fig 2 and 4: There is pretty clearly uneven data density vs. temperature, which is fine, but it makes this type of presentation potentially misleading. Perhaps binning using box plots would help to see the broad trends. These plots should still be in the supplementary material

Showing the high-resolution data allows the reader to assess the true variability in abundances at finer scales than would be possible if the data were binned. A binning approach might be misleading as it requires subjective decisions as to the temperature bins to choose. Picrocyanobacterial abundance data is plotted in four separate figures, with Figures S4 and S5 showing the out and back transects without overlapping. We are concerned that adding an additional figure of picrocyanobacterial abundances would lead to excessive redundancy and prefer not to include such a plot.

Fig 2-5: Unless a style of NatureMicrobiology, the titles should reflect the data presented, rather than interpretation.

We have changed figure legend titles in accordance with this comment.

Fig 3: add years to panels a-c (they are different years?). Why are viral diversity data plotted vs. latitude (in fig 5 too), but other trends (total abundance) and the data in fig 4 plotted against temperature. It's very difficult to make connections between the trends.

Years have been added to the panels as suggested. As requested, we have plotted the distributions of cyanophage lineages by temperature. For Fig. 5, we have left the 2019 abundances plotted against latitude as those are intended to be compared to the maps below.

Fig S2 – this figure really challenges the notion that it is the viral ‘hotspot’ that is causing the decline in Pro. 2015 and 2016 do not support this idea. 2017 does, but if this is a driving factor then why does it not occur in 2/3 of the observations?

We fully agree with the reviewer that the hotspot is not causing the decline of *Prochlorococcus* in 2015 and 2016 and stated as much on lines 81-83 of the previous submission. Rather the declines in 2015 and 2016 occur at temperatures consistent with thermal limits on *Prochlorococcus* growth. We do, however, see some evidence of smaller magnitude dips in *Prochlorococcus*' abundance in the region of the hotspot on the 2015 and 2016 cruises, as noted on lines 168-170 and shown in Fig. 4 and Fig. S5.

We would also like to know why infection was so much higher in 2017, and, thus, why the hotspot had a larger influence on *Prochlorococcus* populations at this time than in 2015 and 2016. The 25% higher abundances of cyanophages in the hotspot in 2017 (which we now mention in the manuscript on line 156) may be a cause of the higher infection. However, this is a bit of a chicken and egg argument as infection affects cyanophage abundances, and cyanophage abundances affect infection. We now discuss plausible scenarios related to climate variability and/or seasonality that may have differentiated 2017 from the previous years. Irrespective of whether these are the drivers for the observed differences, we now discuss the possibility of the system having reached a tipping point in 2017 (lines 279-284) resulting in a stronger than usual positive feedback loop between infection and viral abundances.

Main text (lines 268-286)

Despite consistent features in cyanophage distributions across the North Pacific Ocean, cyanophage infection was higher (Fig. 4, Fig. S7), while *Prochlorococcus* abundances were consistently lower (Fig. 2a), across the June 2017 transect relative to the March 2015 and April 2016 transects. Seasonality and/or climate variability could explain this interannual variability, although the data currently available to assess this are sparse. Viral infection of

picocyanobacteria in the subtropical gyre increased from early spring to summer suggesting a potential seasonal pattern that may extend across the transect (Fig. S9a). In addition, the June

2017 transect occurred during a neutral to negative El Niño phase with lower sea surface temperatures relative to the 2015 and 2016 transects which were in years of a record marine heatwave, followed by a strong El Niño⁵⁵ (Fig. S9b). In 2015 and 2016, *Prochlorococcus* abundances were found to be higher than usual in the North Pacific Ocean in this (Fig. 2a) and in other studies^{56,57}. Irrespective of the underlying drivers for the observed interannual variability, we speculate that an ecosystem tipping point was reached in the prevailing conditions in June 2017. In this scenario, picocyanobacterial populations were subjected to high infection levels that resulted in an accumulation of cyanophages, initiating a stronger than usual positive feedback loop between infection and virus production and precipitating the unexpected *Prochlorococcus* decline. Continued observations in the North Pacific Ocean are needed to evaluate the potential link between seasonality and/or large-scale climate forcing as ultimate drivers affecting virus-host interactions.

Fig S6 – this figure is interesting and suggests that 2017 is different across the entire transect than 2015/2016. See above for more on differentiating the ‘blob’ years from the 2017

Yes, we agree and had mentioned this briefly in line 232-243 (previous version). We now emphasize this point by adding the words “across the June 2017 transect” on line 270 of the revised manuscript and by adding further discussion to the text (lines 268-286, see above). It seems to go hand in hand with the lower *Prochlorococcus* abundances in the entire transect Fig. 2a). See above for revised section of the manuscript.

I hope that these remarks can help the authors improve the manuscript.

We thank the reviewer as we think these suggestions have significantly improved the manuscript.

Reviewer #2 (Remarks to the Author):

The manuscript by Carlson et al. has an intriguing conclusion from an impressive collection of cruise data hinging on the Lindell lab’s “polony” method that detects specific cyanophages. The cruises had two fundamentally different tracks, from Oregon to Hawaii in March 2015 (NE-SW track), and North-South from Hawaii in April 2016 and June 2017 (the March cruise extended several thousand km East of the other two). The results are combined here and it is reported that relatively high abundances of the polony-detectable viruses near the transition from the central

gyre to the subarctic gyre generally correspond to the locations where *Prochlorococcus* abundance transitions from high (south) to low (north) while at the same time *Synechococcus* does the

opposite. The authors basically conclude that the transition is a virus “hot spot” where particularly high virus infection of *Prochlorococcus* keeps the population low and thus

causes the distribution pattern in that entire region; the causation is implicit in the title and stated outright in multiple places. It may be right, but maybe not. It is this grand claim of causation (implying it is the main cause, as opposed to something like grazers), and also several details of the underlying data, that concern me. However, I agree the manuscript has much valuable new information on the distributions of different cyanophages in places that have not before been studied.

We appreciate the acknowledgment by the reviewer of the value of the information in the manuscript. While we understand the reviewer’s concerns about primer and probe designs, these were extensively addressed in previous manuscripts, and we have evaluated them again here in response to these continued concerns. We strongly disagree, however, that combining cruises across different years and tracks should be considered a negative as they allow us to determine which phenomena are widespread and recurrent and which are unusual across a variety of environmental conditions. We agree that we need to more explicitly discuss grazing as a potential additional cause of the unusual *Prochlorococcus* decline. We also agree that it is necessary to change the title of the manuscript, although we did not and do not try to claim that viruses are the sole cause for cyanobacterial distribution patterns in the entire region, but rather the main driver for the unusual decline in 2017. We more fully address these points as they are raised below.

I have two major concerns:

1. One important thing I did not see in this or other papers on the polony method (which is crucial to the results and conclusions here), is any evidence that the method definitely detects essentially *all* the important viruses infecting cyanobacteria throughout the regions these cruises cover, i.e. there are no important ones besides the T4-like, T7-like, TIM5-like ones that match the primers used here - significantly, including variants that simply do not hit the primers well. If the method only detects some of the cyanophages and misses others, then the “hot spot,” relative to places with reported low cyanophages, could be in large part an artifact of detectability. That could negate, or at least significantly blunt, the conclusions. So where is that evidence – or even data providing strong support? Frankly it is quite hard to prove, even though it is an important underlying assumption for the conclusions drawn here. Do we really know that much about cyanophages in these remote ocean locations to be sure we know them all? I think there is still a lot we don’t know. As a reviewer, I do not relish giving very difficult demands to authors, but when the conclusions really depend on a critical

point, I do not see an easy solution around it. This point is out of sight in the paper but nevertheless central.

We understand the reviewer's concerns regarding a primer-based method for quantifying phage abundances and infection in natural populations for the two reasons stated above: 1) whether primers and probes are adequate at detecting the variability of known virus families in complex communities; and 2) whether other groups of viruses are important (abundant and contribute significantly to infection) that are not targeted and detected by the currently used sets of primer.

In regards to the first issue – methodological concerns about primer biases and detection capabilities of the polony method. These have been extensively addressed in previous manuscripts that report the development of the method for virus ecology (Baran et al 2018, *Nat. Microbiol.*, Goldin et al. 2020 *Front. Microbiol.*). The primers and probes, reported in those papers and applied in this current study, were designed using both cyanophage isolates and environmental sequences and both papers show that they capture environmental virus genotypes across the diversity of the cyanophage lineages examined. Furthermore, both Baran et al. 2018 *Nat. Microbiol.* (for T7-like cyanophages) and Mruwat, Carlson et al. 2021 *ISME J* (for T4-like cyanophages) bioinformatically verified that these primers and probes matched cyanophage sequences from metagenomic datasets. For example, Mruwat, Carlson et al. 2021 *ISME J* compared polony primers and probes to the diversity of cyanophages in 44 metagenomic samples collected in the North Pacific Subtropical Gyre in 2015 (this cruise was just 4 months after one of the cruises presented in this manuscript) and found that 93% of cyanophage reads would be detected. It is notable that the relative cyanophage community composition in the metagenomes was similar to that using absolute quantification with the polony method. The relevant text and figure from Mruwat, Carlson et al. 2021 *ISME J* is copied below.

“Metagenomes and viromes were also used to assess the extent to which our degenerate primers and probes captured the major cyanophage types in these waters. The degenerate primers and probes which target the T4-like cyanophage *g20* gene captured the sequence variation in all 11 dominant T4-like cyanophage contigs (46) (Supplementary Fig. 5a). Sequence variation in the individual reads indicated that assemblies represented the dominant genotypes of cyanophages in the water and that primers and probes used in this study were compatible with the diversity of at least 93% of the individual reads (Supplementary Fig. 5b). Furthermore, sequenced amplicons from free cyanophages collected at 25 m and 75 m depths during this cruise indicated that the primers captured a diverse set of T4-like cyanophage from within the population (34). Therefore, any underestimation for cyanophage abundances and infection is likely to be minor and within the threshold of detection, unless due to a presently unknown, nonetheless abundant, cyanophage genotype.”

Supplementary Fig. 6. Comparison of environmental T4-like cyanophage *g20* sequences to degenerate primers and probes. (a) The alignment shows the regions of the *g20* gene that are targeted by the degenerate primers and probes used in the T4-like cyanophage polony method (34). The top row is the primer and probe sequences (sequences shown for the probe and reverse primer are the reverse complements). Degenerate bases in the primer/probe sequence are shown by stacked nucleotides. Cyanophage scaffolds assembled from viromes collected on the same 2015 cruise and a reference cultured T4-like cyanophage (Syn9) are shown below. Pink boxes highlight bases that are nucleotide mismatches to the primer and probe sequences. Note that single nucleotide mismatches were observed for 5 sequences and these were in the middle of the reverse primer and probe which would not interfere with amplification or detection. Empirical testing of the probe indicated that it could tolerate 3 mismatches (Goldin, personal communication). (b) Sequence logos show the relative frequency of bases at each position in the primers and probe based on the alignment of individual T4-like cyanophage reads from the 2015 viromes. The percent of reads aligned to the forward and reverse primers with two or less mismatches was 94% and 97%, respectively. Ninety three percent of reads aligned to the probe sequence with 3 or less mismatches.

In response to reviewer concerns, we have further analyzed the extent to which the primers and probes used in this study capture the diversity of cyanophages from 68 viromes collected across the gradients detailed in the manuscript and taken in coordination with the polony samples on the 2016, 2017, and 2019 cruises. We used a phylogenetic-based approach to identify assembled sequences from the dominant viruses in the water at the time. We then identified the subset of sequences related to cyanophages and compared these sequences to the primer and probe sequences. These methods are previously published in Aylward et al. 2017 *PNAS* and Mruwat, Carlson et al. 2021 *ISME J*.

The following table details the number of putative cyanophage contigs and the number of mismatches with primers and probes. These data indicate that the polony method detects more than 93% of the sequence variation between cyanophages along the gradient transects at the time of sampling, taking into consideration up to two mismatches. Empirical testing indicates that the polony assay can tolerate this level of mismatches in primers with probes capable of detecting variants with up to 3 mismatches. In addition, this analysis does not take into consideration that sequencing error may underlie some of the observed sequence variability. We therefore consider our calculations as conservative estimates, and our primers and probes may actually

detect more than 93% of the sequence variants. Underestimation by this method is, therefore, likely to be minimal. This information has now been added to the Methods section of the manuscript (lines 418-428, see below).

Table 1. Likelihood of detection of metagenomic sequences with polony primers and probes.

	g20 – T4-like cyanophages			polA – T7-like cyanophages		
	Forward	Reverse	Probe	Forward	Reverse	Probe
% match	99%	93%	96%	98%	95%	98%
(# sequences w/ ≤ 2 mismatches/total)	(1233/1238)	(296/317)	(368/385)	(391/397)	(160/167)	(276/283)

With respect to whether currently unknown groups of viruses are abundant and major contributors to picocyanobacterial mortality that were not assessed with the polony method. As implied by the reviewer, it will never be possible to definitively rule out the contribution of yet unknown virus types. However, there are several lines of evidence suggesting that we have captured the dominant players in this system and that other known phage lineages that we did not quantify likely contribute little to cyanobacterial mortality. First, as we showed in Mruwat, Carlson et al. 2021 *ISME J*, based on data in Aylward et al. 2017 *PNAS*, lineages such as cyanosiphoviruses are minor components of the cyanophage community (<10%) in the surface ocean in the North Pacific Subtropical Gyre. Similarly, other groups of non-T4-like or T7-like cyanophages (Sabehi et al. 2012 *PNAS*, Mizuno et al. 2013 *Plos Genetics*, Chenard et al. 2015 *ISME J*, Xu et al. 2018 *Env. Microbiol.*, Flores et al. 2019 *Env. Micro. Rep.*) have all been noted to be significantly less abundant than the T4-like and T7-like cyanophages in metagenomic datasets from global expeditions (e.g. TARA) (Nishimura et al. 2017 *mSphere*). Our own recent discovery of TIM5-like viruses, a non-T4-like cyanomyophage lineage, is a prime example. It was found to be widespread in the oceans using metagenomes (Sabehi et al. 2012 *PNAS*), but quantitative measurements here clearly reveal that they are in very low abundances relative to T4-like and T7-like cyanophages and infect no more than 0.05% of picocyanobacteria even at their observed maximum abundances (unpublished data for the Red Sea) despite their ubiquity.

Second, of the hundreds of single cell genomes of *Prochlorococcus* and *Synechococcus*, many of which were from samples we collected on the 2016 cruise across the transect, only a handful had any evidence of viral infection (Berube et al. 2018 *Sci. Data*). Those cells were infected by T4-like and T7-like phages. It is important to note that if novel yet abundant cyanophage lineages did exist, this untargeted approach of assessing viral infection would be expected to detect them. These independent lines of evidence suggest that cyanophages are a well described group (probably the best in the oceans

currently) and that while novel virus types may well remain unknown at this stage, they are likely to be rare.

Third, the magnitude of hypothetical underestimation of cyanophages suggested by the reviewer would have to be massive in the NPSG to lead to an erroneous hotspot in the transition zone. Cyanophage abundances in the hotspot were 3-10x greater than in the subtropics. If the hotspot is an artifact, we must have underestimated cyanophage abundances in the subtropics by a similar magnitude. This would mean cyanophages would average between $1.7-5.7 \times 10^6$ in the subtropics. Comparing this hypothetical abundance of cyanophages to the abundances of all dsDNA viruses determined from VLP analysis ($5-20 \times 10^6$ viruses ml^{-1}) suggests that at several times cyanophages were

>100% of the dsDNA virus community. This hypothetical scenario seems far outside the bounds of reality for this system where picocyanobacteria are not even the most abundant microbe. Furthermore, as mentioned above, of the regimes described in this manuscript, the NPSG is the best described metagenomically with 44 metagenomes from August 2015 and 12 over the seasonal cycle, which showed that virus assemblages are stable and persistent year-round (Aylward et al. 2017 *PNAS*).

In summary, we have shown that the polony method can detect the diversity of T4-like and T7-like cyanophages across these transects, that multiple independent reports indicate that these are the major cyanophage types in these waters and that other cyanophage types are likely to be minor components. We further show that hypothetical scenarios of our methods underestimating cyanophage abundances in the NPSG lead to unrealistic scenarios. Taken altogether, these lines of evidence provide the strong support the reviewer is requesting, that our methods do indeed capture the major components of the cyanophages in the North Pacific Ocean.

We recognize that other readers may have similar concerns as Reviewer 1. Thus, we have added a paragraph detailing the % mismatch analysis to the methods and have added the data as alignments in supplemental files. We have also added a few sentences explaining that we targeted the major cyanophage lineages and acknowledge those we did not investigate (lines 132-137). We further qualified that our discussion of cyanophage makeup is for the cyanophage lineages we measured (line 144).

Main text

Lines 132-137

We targeted the T7-like clade A and clade B cyanopodoviruses and the T4-like cyanomyoviruses, three major cyanophage lineages based on isolation studies^{18-20,24,40}, single-cell genomics⁴¹, and global metagenomic surveys^{25,26,42-44} as well as a more recently discovered

group, the TIM5-like cyanomyoviruses^{42,44}. Cyanophages from other lineages that are less common in metagenomic surveys^{39,42,43} were not investigated.

Lines 144-145

Together, these two clades constituted >80% of cyanophages measured, with the remainder consisting of T7-like clade A and TIM5-like cyanophages (Fig. 3, Fig. S4).

In the Methods: Lines 418-428

Viral metagenomes (n=68) were collected in parallel with polony samples across the 2016, 2017, and 2019 transects and used to assess the extent to which the degenerate primers and probes used to target T4-like and T7-like cyanophages captured the major cyanophage types in these waters. We used a phylogenetic-based approach to classify assembled sequences (Supplementary files 1-

6) that represent the dominant viruses in the water at the time of sampling, as described previously³⁹. The polony assay was able to capture 99% (1233/1238), 93% (296/317), and 96% (368/385) of T4-like cyanophage sequences with ≤ 2 mismatches to the forward primer, reverse primer, and probe sequences, respectively. Similarly, the polony assay was able to capture 98% (391/397), 95% (160/167), and 98% (276/283) of T7-like cyanophage sequences with ≤ 2 mismatches to the forward primer, reverse primer, and probe sequences, respectively. No assemblies were detected for TIM5-like cyanophages, likely due to their low abundance.

2. Important: The data in Fig 4e (and elsewhere) indeed show that there is some correspondence between where viruses are high, and low Prochlorococcus. Correlation does not mean causation - it could be incidental, or both related to an important unmeasured third parameter. How can we know it is not at least in part from flagellates that eat both Syn and Pro "causing" enough Prochlorococcus mortality to prevent their population from increasing (because they grow slower than Syn). And maybe there is something about the transition zone that leads to higher small flagellates (or higher grazing from pre-existing mixotrophs) – with a reason just as unknown as why the viruses would be higher there. After all, predators are the explanation that Lindell coauthored with Biller, Chisholm et al. in their 2015 review, which said: "Furthermore, Synechococcus strains have higher maximum growth rates than Prochlorococcus and they are prey for many of the same predators. As the growth rate of predators is coupled to that of their prey, it may be impossible for Prochlorococcus to achieve net positive growth rates when Synechococcus is growing maximally — it would simply be 'grazed away' " This concern alone is enough to raise significant doubt about the title and main conclusions of this manuscript.

We agree that mortality in environmental systems is the sum of many factors including viral lysis, grazing, and abiotic factors which we mention several times throughout the manuscript (lines 134, 170, 301 of the initial submission), although perhaps this was not emphasized sufficiently in that version of the manuscript. Our work goes beyond correlations and

specifically measures cell infection by lytic viruses (Fig. 4a-f) in addition to cell and virus abundances, indicating that viruses contributed substantially to *Prochlorococcus* mortality (estimated to be up to 51%) and propose that this high level

of virus-induced mortality, beyond the “normal” mortality levels caused by multiple factors that usually maintain the steady state between growth and death, is what brought about the sharp decline in *Prochlorococcus* populations in 2017 (see lines 170- 172 of the original manuscript and lines 192-196 of the revised manuscript). We also agree that grazing or other mortality factors could be involved in the “additional” mortality and the unexpected decline of *Prochlorococcus* in 2017. However, overall grazing in the transition zone does not appear to have been elevated in 2017 as total bacterial abundances were similar between all three cruises and were even higher south of the chlorophyll front where *Prochlorococcus* abundances were unusually low in 2017, suggesting that a mortality process specific to picocyanobacterial was in play (see lines 84-86, 103-108 and 125-129 shown below). Importantly, a potential role for grazing does not detract from our findings that show the involvement of viral infection in this phenomenon. Based on the reviewer’s comments, we modified the text to give more weight and emphasis to grazing on lines 192-198, and also specifically qualify our statements regarding the role of viruses by stating that we consider virus-mediated mortality in 2017 to be “a significant factor” in limiting the geographic range of *Prochlorococcus* (line 201).

The title of the manuscript has also been modified in line with the reviewer’s comment.

A note regarding the citation mentioned by the reviewer from Biller et al. 2015 *Nat. Rev. Microbiol.*: This comment while made in context of grazing is not limited to it but can be expanded to any mortality factor that affects both *Prochlorococcus* and *Synechococcus*. This statement could easily apply to viruses as well and would be supported by our data. Some T4-like cyanophages can infect both *Prochlorococcus* and *Synechococcus*. We observed increased infection by T4-like cyanophages in the transition zone in both *Prochlorococcus* and *Synechococcus*. Just as with grazing, the mortality by such broad host range phages could reduce slower growing *Prochlorococcus* populations while *Synechococcus* maintained net positive growth. Note that this original statement was not specifically aimed at viruses at that time (in 2015) as there was no evidence to suggest viruses would have such an effect on *Prochlorococcus*’ geographic distribution, and in fact, we were skeptical of this prior to seeing the data for 2017.

Main text

Lines 84-86:

Prochlorococcus abundances remained high in the southern region of the transition zone in 2015 and 2016, decreasing precipitously to less than ~ 2000 cells·ml⁻¹ north of the chlorophyll front, generally constituting <5% of total bacteria (Fig. 2, S1b, S2).

Lines 103-108:

Total bacterial abundances were stable in the subtropics and increased 1.4-3.2-fold in the transition zone on all three cruises (Fig. S1a). This increase occurred north of the chlorophyll front in 2015 and 2016 whereas they increased south of this feature in June 2017. Thus, the 2017 increase in total bacteria was despite the anomalous loss of *Prochlorococcus*, which made up only 5% of the total bacteria south of the chlorophyll front relative to 20-30% in the equivalent region in 2015 and 2016 (Fig. S1b).

Lines 125-129:

The lack of an identifiable abiotic variable differentiating the 2017 transect from the other two transects and the overall high abundances in total bacteria for all three transects (Fig. S1a) led us to hypothesize that a mortality factor specific to picocyanobacteria, such as infection by viruses, played a role in precipitating the observed shifts in picocyanobacterial geographic ranges.

Lines 192-203:

Under quasi steady state conditions, abiotic controls on the growth rate of *Prochlorococcus* are balanced by mortality due to viral lysis, grazing, and other mortality agents^{39,45,47}. Based on both the high levels of virus-mediated mortality and the parallel pattern between *Prochlorococcus*' death and viral infection, we propose that enhanced viral infection in 2017 disrupted this balance, leading to the unexpected decline in *Prochlorococcus* populations. Grazing and other mortality agents not investigated here could also have contributed to additional mortality beyond the steady state, resulting in further losses of *Prochlorococcus*. In contrast to *Prochlorococcus*, *Synechococcus* maintained large populations despite high levels of infection (Fig 4f), presumably because of faster growth rates enabling them to maintain positive net growth despite enhanced mortality. These findings suggest that virus-mediated mortality in 2017 was a significant factor in limiting the geographic range of *Prochlorococcus* that resulted in a massive loss of habitat of approximately 550 km.

Other issues:

It may be problematic using the three cruises in different months, of different years, and two very different cruise tracks, with the transition zones thousands of miles apart. The huge North Pacific Gyre is not so uniform. For example, I recall seeing results from the Zehr lab about very different cyanobacteria and nitrogen fixer communities within the gyre along a transect between Hawaii and California. March may be hardly comparable to June. Lumping the cruises and comparing together is tricky (and iffy)— as is done in lines starting 102. Discussion of things like “expansion” of *Synechococcus* in 2017 (line 113, line 176) seems to be based on little data. “Expansion” related to what - expectations from another year? It all seems a little sloppy. And sometimes only 2 of the three years are referred to, e.g. line 154. At least more caveats are called for.

The repeated sampling of the transition zone from multiple cruises strengthens the ability to determine which of the phenomena reported are widespread and recurrent in the North Pacific Ocean, such as the virus hotspot we present in this manuscript. The observed reproducibility of the hotspot at such a wide range of time scales (days, weeks,

months, years) and spatial scales (km to thousands of km) on 4 different cruises is remarkable, especially considering the observed variability in abiotic and biotic parameters in the North Pacific Ocean. We are very surprised that the reviewer would consider this to be ‘problematic’ and ‘sloppy’ as such repeated sampling at different times and places allows for discovery of generalizable ecosystem features (see for example Ustick et al. 2021 *Science*, Flombaum et al. 2013 *PNAS*). Rather than “lump” cruises together, as suggested by the reviewer, we show each cruise separately and then compare the similarities and differences between the transects. This approach uncovered the decline of *Prochlorococcus* at temperatures considerably higher than expected along with high virus infection, and that this was a departure from the norm in relation to the other two cruises where viral infection was low and *Prochlorococcus* declined at expected temperatures (as observed by others in the same region, such as Larkin et al. 2016 *ISME J*). Thus, our multi-cruise comparison identified both recurrent phenomena and those that are out of the ordinary.

Based on reviewer suggestions, we have revised our language to be more explicit when comparing cruises and to more carefully outline our basis for the expansion of the *Synechococcus* range, which is based on the other cruises in this manuscript as well as 9 additional cruises previously reported in Gainer et al. 2017 *Plos One*, Sohm et al. 2016 *ISME J* and Juranek et al. 2012 *JGR*. These changes can be found in the abstract on lines 34-35 and in the main text, specifically on lines 96-101.

Abstract lines 34-35:

At these times, large *Synechococcus* populations were found in waters typically dominated by *Prochlorococcus*.

Main text (lines 96-101)

In contrast, the geographic range of *Synechococcus* was broader by ~3 degrees of latitude (~330 km) and their integrated abundances across the transect were 2-fold higher than observed in 2015 and 2016 (Fig. 2b, Fig. S2). Thus, a more southern decline in *Prochlorococcus* and a broader distribution of *Synechococcus* (Supplementary Text) was observed in 2017 relative to the 2015 and 2016 transects (Fig. 2, Fig S2) and to 9 previous transects across this transition zone conducted in different years and seasons³¹⁻³³.

Perspective: The "massive decline" in *Prochlorococcus* (line 167), ascribed to viruses (the main point of the paper) seems to imply they were there and then disappeared over some short time (and were replaced by the “expansion” of *Synechococcus*). But there is no evidence of that here.

It is a "massive decline" only from the perspective of a ship traveling North on their transect, quite an artificial perspective (it can be called a "massive increase" just as well, traveling South). Otherwise it is just like any distribution

of an organism which is higher in some places and lower elsewhere, sometimes at a sharp boundary (like between gyres, or near the equator). The local history can only be guessed, without multiple measurements in a season. The causes are often very complex. Here the authors seem to look only at temperature, nutrients and viruses, and try to pin the blame completely in viruses. That is a narrow view, considering they did not even look at grazers.

The reviewer comments illustrate the power of conducting our studies against a backdrop of extensive distribution data for both *Prochlorococcus* and *Synechococcus*. Our discussion of a decline is compared to that expected with respect to the temperature range *Prochlorococcus* is known to inhabit based on a large body of work reported by many other scientists well before us (Johnson et al. 2006 *Science*, Bouman et al. 2006 *Science*, Zinser et al. 2007 *Limnol. and Oceanogr.*, Zwirgmaier et al. 2008 *Environ. Microbiol.*, Juranek et al. 2012 *J. Geophys. Res.*, Flombaum et al. 2013 *PNAS*, Sohm et al. 2015 *ISME J*, Larkin et al. 2016 *ISME J*, Gainer et al. 2017 *PlosOne*).

Thousands of field observations from numerous previous field campaigns, our own findings for the 2015 and 2016 cruises, and laboratory studies all suggest *Prochlorococcus* should, and did occupy waters at similar temperatures in abundances near 10^5 cells ml^{-1} . Thus, the temperature related distribution of *Prochlorococcus* is well known both for this region (Juranek et al. 2012 *J. Geophys. Res.*, Sohm et al. 2015 *ISME J*, Larkin et al. 2016 *ISME J*, Gainer et al. 2017 *Plos One*) and in general. We now cite these additional papers for reference.

In the text we state that the observed massive decline in *Prochlorococcus* is relative to the expected habitat based on temperature (lines 86-88 and 94-95, see below). We intentionally compared abundances on the different cruises to the location of the chlorophyll front as an indicator of the transition zone in order to provide a point of reference relative to trophic status. Importantly, we do not make claims regarding the time scale of the *Prochlorococcus* decline relative to temperature and our use of the term decline is not to imply a time scale as we cannot know what that would be from our data.

The high degree of viral infection in 2017 indicates a major role of viruses in shaping the distribution of *Prochlorococcus* on this occasion. Abiotic conditions and grazing are noted throughout the manuscript as important factors impacting phytoplankton distribution in general as well as in the context of the unexpected decline (lines 57-59, 134-135, 170-171, and 305-307 in the previous version of the manuscript). In particular, we point out our conclusions paragraph where we explicitly stated that the virus hotspot “served as an additional limit superimposed on the gradients in abiotic conditions” (lines 306-307 of previous version and line 330 of the revised manuscript). Temperature is the

widely accepted parameter that controls *Prochlorococcus* distributions (Johnson et al. 2006 *Science*, Bouman et al. 2006 *Science*, Zinser et al. 2007 *Limnol. and Oceanogr.*, Zwirgmaier et al. 2008 *Environ. Microbiol.*, Flombaum et al. 2013 *PNAS*, Biller et al. 2016 *Nat. Rev. Microbiol.*) with nutrient limitation often implicated as well (Saito et al. 2014 *Science*, Browning et al. 2017 *Nature*, Kent et al. 2019 *ISME J*, Ustick et al. 2021 *Science*). Thus, we test hypotheses regarding these accepted drivers, which is now more explicitly demonstrated in the text and figures in response to comments by reviewer 1 (lines 110- 122). Prior to this study there was scant evidence that mortality (whether due to viral infection or grazing) is involved in shaping the biogeography of picocyanobacteria, which we mention in the introduction on lines 54-56. Our data provide evidence for the role of viral infection. As discussed above, we agree that grazing could also have played a role in addition to virus infection, and now more explicitly discuss this possibility on lines 192-203 as discussed above.

Main text:

Lines 54-56:

Mortality factors, such as grazing and viral infection, are regarded as important regulators of picocyanobacterial abundances and diversity^{8,17}, but are seldom considered as factors impacting cyanobacterial biogeography.

Lines 84-101:

Prochlorococcus abundances remained high in the southern region of the transition zone, decreasing precipitously to less than ~ 2000 cells \cdot ml⁻¹ north of the chlorophyll front, generally constituting $<5\%$ of total bacteria (Fig. 2, S1b, S2). This decline occurred at water temperatures of ~ 12 °C (Fig. 2) and is consistent with thermal limits on *Prochlorococcus*' growth determined for cultures and numerous field observations^{11,12,15,30}. Conversely, *Synechococcus* was 10-100-fold more abundant in the transition zone relative to the subtropics and gradually decreased northward towards subpolar waters (Fig 2, Fig. S2).

Picocyanobacterial abundance patterns differed dramatically in June 2017. The decline in *Prochlorococcus* occurred ~ 2 degrees south (~ 220 km) of the chlorophyll front (Fig. S1b, Fig. S2) where water temperatures were nearly 18 °C (Fig. 2a). This implicated factors other than temperature as responsible for significantly restricting the geographic distribution of *Prochlorococcus*. In contrast, the geographic range of *Synechococcus* was broader by ~ 3 degrees of latitude (~ 330 km) and their integrated abundances across the transect were 2-fold higher than observed in 2015 and 2016 (Fig. 2b, Fig. S2). Thus, a more southern decline in *Prochlorococcus* and a broader distribution of *Synechococcus* (Supplementary Text) was observed in 2017 relative to the 2015 and 2016 transects (Fig. 2, Fig S2) and to 9 previous transects across this transition zone conducted in different years and seasons³¹⁻³³.

Lines 110-122

nature portfolio

A suite of abiotic variables beyond temperature, many of which are considered important determinants for *Prochlorococcus*' biogeography^{11-16,34}, were assessed for their potential role in

restricting the geographic distribution of *Prochlorococcus* on the 2017 transect (Fig. S3). Macronutrient (phosphate and nitrite+nitrate, Fig. S3a,c) and micronutrient (iron²⁷) concentrations were within the range for *Prochlorococcus*' optimal growth³⁵, and lead (Pb) concentrations²⁷ were below levels toxic to *Prochlorococcus*³⁶. *Prochlorococcus* populations in 2017 declined at similar values of salinity (Fig. S3e) and particulate carbon (Fig. S3g) as the 2016 transect, while thriving across a wider range of values for these variables in 2015.

Furthermore, a mixed layer depth threshold was not associated with the decline in *Prochlorococcus* (Fig. S3i). Collectively, the abiotic conditions observed in 2017 in the region of the *Prochlorococcus* decline, supported large populations of *Prochlorococcus* in 2015 and 2016 (Fig. S3). Thus, none of the physical or chemical factors investigated here can explain the unexpected decline in *Prochlorococcus* in 2017.

Lines 192-203

Under quasi steady state conditions, abiotic controls on the growth rate of *Prochlorococcus* are balanced by mortality due to viral lysis, grazing, and other mortality agents^{39,45,47}. Based on both the high levels of virus-mediated mortality and the parallel pattern between *Prochlorococcus*' death and viral infection, we propose that enhanced viral infection in 2017 disrupted this balance, leading to the unexpected decline in *Prochlorococcus* populations. Grazing and other mortality agents not investigated here could also have contributed to additional mortality beyond the steady state, resulting in further losses of *Prochlorococcus*. In contrast to *Prochlorococcus*, *Synechococcus* maintained large populations despite high levels of infection (Fig 4f), presumably because of faster growth rates enabling them to maintain positive net growth despite enhanced mortality. These findings suggest that virus-mediated mortality in 2017 was a significant factor in limiting the geographic range of *Prochlorococcus* and resulted in a massive loss of habitat of approximately 550 km.

I see that on line 169, the authors calculate viral induced *Prochlorococcus* mortality (with many underlying assumptions!), but don't show the corresponding calculation for *Synechococcus* mortality – is it consistent with the results/conclusions? I would think differential virus effects should be compared. One could imagine *Syn* might have an even higher mortality from viruses there, being more abundant, larger and growing faster.

The assumptions used in our calculations of viral-induced mortality are well constrained, based on well-documented, independent observations, and result in conservative estimates of mortality. For example, using fewer assumptions such as not adjusting for temperature or not considering day length (which reviewer 1 recommended was important to consider), results in higher levels of mortality (66% vs 51% reported in the manuscript). The assumptions used for these calculations are provided in the Methods so the reader can evaluate them and we now also mention their conservative nature (lines 437-450).

We appreciate the reviewer's suggestion to provide similar estimates for *Synechococcus*. We used temperature-related growth rates from Pittera et al. 2014 *ISME J* for clade I which is a clade expected to be found in these waters and for which sufficient data is available. We estimated that *Synechococcus* would double approximately once per day compared to *Prochlorococcus* which was estimated to double every ~2.8 days. Given these generation times, the relative impact of viruses was less for *Synechococcus* infection before division and ranged between 9-31% compared to *Prochlorococcus* which ranged between 21-51%. Note that the latent periods used are based on empirical data (Table S2) and are different for *Synechococcus* and *Prochlorococcus* for the T7-like clade B phages. These calculations support the idea that *Synechococcus* maintains higher abundances in the transition zone when viral infection is high due to higher growth rates. We have added this discussion to the manuscript (lines 183-190 and 198-201).

Main text Lines

183-190:

We estimate that viruses killed between 10-30% of *Prochlorococcus* and *Synechococcus* cells each day at these high instantaneous levels of infection (Fig. S6) based on the expected number of infection cycles cyanophages were able to complete at the light and temperature conditions in the transition zone (see Methods). Since *Prochlorococcus* is estimated to double every 2.8 ± 0.8 days at the low temperatures in this region¹², we estimate that between 21-51% of the population was infected and killed in the interval prior to cell division. *Synechococcus* is expected to have faster growth rates at these temperatures^{12,46} and we estimate that less of the *Synechococcus* population, 9-31%, was killed prior to division.

Lines 198-201

In contrast to *Prochlorococcus*, *Synechococcus* maintained large populations despite high levels of infection (Fig 4f), presumably because of faster growth rates enabling them to maintain positive net growth despite enhanced mortality.

Lines 437-450

Thus, we further refined these estimates to be more conservative by adjusting for the impact of day length, light levels, and temperature. First, we applied a correction for temperature, assuming that for every 3 °C decrease in temperature from 21 °C, the latent period lengthened by 25% and likewise shortened for increasing temperatures (unpublished data). Second, cyanophage infections in high light intensities ($210 \mu\text{mol photons} \cdot \text{m}^{-2} \cdot \text{s}^{-1}$) are 40% shorter than those conducted at low light intensities ($15 \mu\text{mol photons} \cdot \text{m}^{-2} \cdot \text{s}^{-1}$)⁶⁶. Third, we applied this light correction such that cyanophages had 60% shorter latent periods during daylight hours and that no lysis occurred during nighttime as suggested previously³⁹. These assumptions yielded estimates of ~3-4 cycles per day for the dominant cyanophage types in the subtropical gyre³⁹, and ~2-3 infection cycles per day considering the cooler temperatures and shorter day length in the transition zone. These assumptions yield conservative estimates in mortality. Using fewer

assumptions such as not adjusting for temperature or not considering day length and light levels, resulted in higher levels of mortality (a maximum of 66% vs 51% reported here).

Reviewer #3 (Remarks to the Author):

Thank you for the opportunity to review this paper. I have been asked by the editor to help in assessing the mathematical modeling aspects of the study only. As I am not a marine microbiologist expert, I will limit myself to that and ask for forgiveness if my comments are basic.

My two questions are about the choice of predictors for the multiple regression model, and about the use of a multiple regression model for predictions.

Based on model root mean square error, the authors choose chlorophyll and temperature as predictors in their multiple regression model. At the same time, I understand that viruses abundance depends on the availability of bacteria hosts (line 47-49 or the manuscript). If that is the case, one would expect that bacteria abundance would be a better predictor. Indeed, the authors seem to say so on line 470-472. If chlorophyll can be used as a proxy for bacteria, this would be a better argument for the authors' choice of predictor than similar RMSE values.

We apologize that the rationale behind our choice to use chlorophyll and temperature was not clearly stated for those outside of oceanography/marine microbiology. Our aim was to predict virus abundances over large ocean expanses because virus abundance data is often reported at low spatial resolution. We reported the strong correlation between virus abundance and infected cell abundances in the manuscript and agree that developing models with cyanobacterial abundances would be the way to go if these data were as available at anywhere near the resolution of chlorophyll data.

Cyanobacterial abundances can be measured only from samples collected on a ship which can only go to one particular site at one particular point in time. In contrast, chlorophyll concentrations and temperature are measured by satellite and are thus collected remotely and continuously at very high spatial (hundreds of kilometers) and temporal resolution (multiple swaths in days), and the data are freely available. It is also important to mention that chlorophyll is directly linked to the abundances of photosynthetic organisms which includes the

cyanobacteria (Bouman et al. 2012 *Science*). In reading this section again, we acknowledge the reasons for using satellite derived chlorophyll data and not cyanobacterial counts were not as clearly laid out as

needed for those less familiar with large-scale ecology or oceanography. We have added these clarifications on to the main text.

Main text (lines 289-294)

Measurements of cyanobacterial and cyanophage abundances rely on discrete sample collection from shipboard oceanographic expeditions, which limits the geographical and seasonal extent of available data. Therefore, we developed a multiple regression model based on high-resolution satellite data of temperature and chlorophyll to predict cyanophage abundances, a key proxy of cyanobacterial infection (Pearson's $r=0.61$, two-sided $p\text{-value}=1.7\times 10^{-8}$, $df=68$, $n=70$). We used the model to estimate the geographic extent of the virus hotspot.

My second question is about the use of a regression model itself. My understanding is that the authors argue for a balance between viruses and bacteria to be important, with the viruses depending on the availability of bacteria to reproduce, and the bacteria dying because of virus infection (lines 44-51). With the goal of making predictions, a population model, including abiotic forcing, would seem more appropriate than a regression model. A regression model neglects the interaction between the bacteria and viruses. That would be the case even if bacteria abundances were used as a predictor, as it would still be an independent variable.

{redacted}

Surveys such as ours aim, in part, at being able to provide the data necessary for the inclusion of viruses in future population models. We have been involved in such projects using our previously published datasets of virus abundances and infection (Mruwat, Carlson et al. 2021 *ISME J*) to develop and finetune population models for the North Pacific Subtropical Gyre, the efforts of which are still ongoing (Beckett et al. 2021 BioArxiv). Development of such a population model for the larger North Pacific Ocean that makes use of our data is a future of aim of ours but is beyond the scope of this current study. We have added a perspective addressing this in the conclusions to highlight the need for continued progress to include viruses in population models (lines 334-338).

Even though our data-driven regression model does not include details of the specific interactions between bacteria and viruses, it successfully predicted the position of the virus hotspot in the transition zone. Thus, while we are not yet able to incorporate our data into a population model, it provides insight into virus distribution patterns on a regional scale for the North Pacific Ocean which was not possible previously.

Main text (lines 334-338)

Expansion of this model for other ocean regions, determination of the population traits that lead to these ecosystem features, and the development of population models for cyanophages and other autotroph-virus systems, will allow us to gain a global view of the impacts of viruses on marine ecosystems.

{redacted}

Please let me stress again that I might be missing some very fundamental point that may be obvious to any microbiologist, and take my comments as a layman questions more than anything else.

Thank you for the opportunity to review this paper, I have learned a lot. Best

regards

We thank the reviewer for the comments and particularly appreciate feedback from those not immersed in marine microbiology. We think these perspectives will help make the manuscript more accessible to a broad audience.

References:

1. Avrani S, Wurtzel O, Sharon I, Sorek R & Lindell D (2011) Genomic island variability facilitates *Prochlorococcus*-virus coexistence. *Nature* **474**: 604-608.
2. Aylward FO, Boeuf D, Mende DR, Wood-Charlson EM, Vislova A, Eppley JM, Romano AE & DeLong EF (2017) Diel cycling and long-term persistence of viruses in the ocean's euphotic zone. *Proc Natl Acad Sci USA* **114**: 11446-11451.
3. Baran N, Goldin S, Maidanik I & Lindell D (2018) Quantification of diverse virus populations in the environment using the polony method. *Nat Microbiol* **3**: 62-72.
4. Beckett SJ, Demory D, Coenen AR, *et al.* (2021) Diel population dynamics and mortality of *Prochlorococcus* in the North Pacific Subtropical Gyre. *bioRxiv* 2021.2006.2015.448546.
5. Berube PM, Biller SJ, Hackl T, *et al.* (2018) Single cell genomes of *Prochlorococcus*, *Synechococcus*, and sympatric microbes from diverse marine environments. *Sci Data* **5**: 180154.
6. Biller SJ, Berube PM, Lindell D & Chisholm SW (2015) *Prochlorococcus*: the structure and function of collective diversity. *Nat Rev Microbiol* **13**: 13-27.
7. Bouman HA, Ulloa O, Scanlan DJ, *et al.* (2006) Oceanographic basis of the global surface distribution of *Prochlorococcus* ecotypes. *Science* **312**: 918-921.
8. Browning TJ, Achterberg EP, Rapp I, Engel A, Bertrand EM, Tagliabue A & Moore CM (2017) Nutrient co-limitation at the boundary of an oceanic gyre. *Nature* **551**: 242-246.
9. Chenard C, Chan AM, Vincent WF & Suttle CA (2015) Polar freshwater cyanophage S-EIV1 represents a new widespread evolutionary lineage of phages. *ISME J* **9**: 2046-2058.
10. Echeveste P, Agustí S & Tovar-Sánchez A (2012) Toxic thresholds of cadmium and lead to oceanic phytoplankton: cell size and ocean basin-dependent effects. *Environ Toxicol Chem* **31**: 1887-1894.
11. Flombaum P, Gallegos JL, Gordillo RA, *et al.* (2013) Present and future global distributions of the marine cyanobacteria *Prochlorococcus* and *Synechococcus*. *Proc Natl Acad Sci USA* **110**: 9824-9829.
12. Flores-Urbe J, Filosof A, Sharon I, Fridman S, Larom S & Béjà O (2019) A novel uncultured marine cyanophage lineage with lysogenic potential linked to a putative marine *Synechococcus* 'relic' prophage. *Environ Microbiol Rep* **11**: 598-604.
13. Follows MJ, Dutkiewicz S, Grant S & Chisholm SW (2007) Emergent biogeography of microbial communities in a model ocean. *Science* **315**: 1843-1846.
14. Gainer PJ, Pound HL, Larkin AA, LeCleir GR, DeBruyn JM, Zinser ER, Johnson ZI & Wilhelm SW (2017) Contrasting seasonal drivers of virus abundance and production in the North Pacific Ocean. *PLoS ONE* **12**: e0184371.

15. Goldin S, Hulata Y, Baran N & Lindell D (2020) Quantification of T4-like and T7-like cyanophages using the polony method show they are significant members of the viroplankton of the North Pacific Subtropical Gyre. *Front Microbiol* **11**: 1210.
16. Johnson ZI, Zinser ER, Coe A, McNulty NP, Woodward EMS & Chisholm SW (2006) Niche partitioning among *Prochlorococcus* ecotypes along ocean-scale environmental gradients. *Science* **311**: 1737-1741.

17. Juranek LW, Quay PD, Feely RA, Lockwood D, Karl DM & Church MJ (2012) Biological production in the NE Pacific and its influence on air-sea CO₂ flux: Evidence from dissolved oxygen isotopes and O₂/Ar. *J Geophys Res Oceans* **117**.
18. Kent AG, Baer SE, Mouginito C, Huang JS, Larkin AA, Lomas MW & Martiny AC (2019) Parallel phylogeography of *Prochlorococcus* and *Synechococcus*. *ISME J* **13**: 430- 441.
19. Larkin AA, Moreno AR, Fagan AJ, Fowlds A, Ruiz A & Martiny AC (2020) Persistent El Niño driven shifts in marine cyanobacteria populations. *PLoS ONE* **15**: e0238405.
20. Larkin AA, Blinebry SK, Howes C, Lin Y, Loftus SE, Schmaus CA, Zinser ER & Johnson ZI (2016) Niche partitioning and biogeography of high light adapted *Prochlorococcus* across taxonomic ranks in the North Pacific. *ISME J* **10**: 1555-1567.
21. Mizuno CM, Rodriguez-Valera F, Kimes NE & Ghai R (2013) Expanding the marine virosphere using metagenomics. *PLoS Genet* **9**: e1003987.
22. Mruwat N, Carlson MCG, Goldin S, *et al.* (2021) A single-cell polony method reveals low levels of infected *Prochlorococcus* in oligotrophic waters despite high cyanophage abundances. *ISME J* **15**: 41-54.
23. Nishimura Y, Watai H, Honda T, *et al.* (2017) Environmental viral genomes shed new light on virus-host interactions in the ocean. *mSphere* **2**.
24. Pinedo-González P, Hawco NJ, Bundy RM, Armbrust EV, Follows MJ, Cael BB, White AE, Ferrón S, Karl DM & John SG (2020) Anthropogenic Asian aerosols provide Fe to the North Pacific Ocean. *Proc Natl Acad Sci USA* **117**: 27862-27868.
25. Pittera J, Humily F, Thorel M, Grulois D, Garczarek L & Six C (2014) Connecting thermal physiology and latitudinal niche partitioning in marine *Synechococcus*. *ISME J* **8**: 1221-1236.
26. Polovina JJ, Howell E, Kobayashi DR & Seki MP (2001) The transition zone chlorophyll front, a dynamic global feature defining migration and forage habitat for marine resources. *Prog Oceanogr* **49**: 469-483.
27. Sabehi G, Shaulov L, Silver DH, Yanai I, Harel A & Lindell D (2012) A novel lineage of myoviruses infecting cyanobacteria is widespread in the oceans. *Proc Natl Acad Sci USA* **109**: 2037-2042.
28. Saito MA, McIlvin MR, Moran DM, Goepfert TJ, DiTullio GR, Post AF & Lamborg CH (2014) Multiple nutrient stresses at intersecting Pacific Ocean biomes detected by protein biomarkers. *Science* **345**: 1173-1177.
29. Sohm JA, Ahlgren NA, Thomson ZJ, Williams C, Moffett JW, Saito MA, Webb EA & Rocap G (2016) Co-occurring *Synechococcus* ecotypes occupy four major oceanic regimes defined by temperature, macronutrients and iron. *ISME J* **10**: 333-345.
30. Thompson AW, Huang K, Saito MA & Chisholm SW (2011) Transcriptome response of high- and low-light-adapted *Prochlorococcus* strains to changing iron availability. *ISME J* **5**: 1580-1594.
31. Ustick LJ, Larkin AA, Garcia CA, Garcia NS, Brock ML, Lee JA, Wiseman NA, Moore JK & Martiny AC (2021) Metagenomic analysis reveals global-scale patterns of ocean nutrient limitation. *Science* **372**: 287-291.
32. Xu Y, Zhang R, Wang N, Cai L, Tong Y, Sun Q, Chen F & Jiao N (2018) Novel phage– host interactions and evolution as revealed by a cyanomyovirus isolated from an estuarine environment. *Environ Microbiol* **20**: 2974-2989.
33. Zborowsky S & Lindell D (2019) Resistance in marine cyanobacteria differs against specialist and generalist cyanophages. *Proc Natl Acad Sci USA* **116**: 16899-16908.

34. Zinser ER, Johnson ZI, Coe A, Karaca E, Veneziano D & Chisholm SW (2007) Influence of light and temperature on *Prochlorococcus* ecotype distributions in the Atlantic Ocean. *Limnol Oceanogr* **52**: 2205-2220.
35. Zwirgmaier K, Heywood JL, Chamberlain K, Woodward EMS, Zubkov MV & Scanlan DJ (2007) Basin-scale distribution patterns of picocyanobacterial lineages in the Atlantic Ocean. *Environ Microbiol* **9**: 1278-1290.
36. Zwirgmaier K, Jardillier L, Ostrowski M, Mazard S, Garczarek L, Vaulot D, Not F, Massana R, Ulloa O & Scanlan DJ (2008) Global phylogeography of marine *Synechococcus* and *Prochlorococcus* reveals a distinct partitioning of lineages among oceanic biomes. *Environ Microbiol* **10**: 147-161.

Decision Letter, second revision:

9th November 2021

Dear Debbie,

Thank you for your patience while your manuscript "Enhanced infection of picocyanobacteria in a virus hotspot between ocean gyres" was under peer-review at Nature Microbiology. It has now been seen by 4 referees, whose expertise and comments you will find at the of this email. You will see from their comments below that while they find your work of interest, some important points are raised. We are very interested in the possibility of publishing your study in Nature Microbiology, but would like to consider your response to these concerns in the form of a revised manuscript before we make a final decision on publication.

Referee #1 feels that data presentation could still be improved (making it more accessible for the reader), that the data also points to alternative mechanisms for the decline in cyanobacteria, and says that "Nevertheless, while I think the data show that there can be an area of enhanced viral abundance and % infected cyanobacteria, it less clear that the viruses are driving to the decline.". Editorially, we feel this will need to be discussed in the text. Referee #2 feels that coverage of the polony method could be better explained, and that the manuscript "would benefit from the more detailed explanations as written in responses to reviews". Referee #4 also has "slight worries about the governing processes driving Pro dynamics", but feels this can be addressed in the Discussion. The rest of the referees' reports are clear and the remaining issues should be straightforward to address.

If you have not done so already please begin to revise your manuscript so that it conforms to our Article format instructions at <http://www.nature.com/nmicrobiol/info/final-submission/>

The usual length limit for a Nature Microbiology Article is six display items (figures or tables) and 3,000 words. We have some flexibility, and can allow a revised manuscript at 3,500 words, but please consider this a firm upper limit. There is a trade-off of ~250 words per display item, so if you need more space, you could move a Figure or Table to Supplementary Information.

Some reduction could be achieved by focusing any introductory material and moving it to the start of your opening 'bold' paragraph, whose function is to outline the background to your work, describe in a sentence your new observations, and explain your main conclusions. The discussion should also be limited. Methods should be described in a separate section following the discussion, we do not place a word limit on Methods.

Nature Microbiology titles should give a sense of the main new findings of a manuscript, and should not contain punctuation. Please keep in mind that we strongly discourage active verbs in titles, and that they should ideally fit within 90 characters each (including spaces).

We strongly support public availability of data. Please place the data used in your paper into a public

data repository, if one exists, or alternatively, present the data as Source Data or Supplementary Information. If data can only be shared on request, please explain why in your Data Availability Statement, and also in the correspondence with your editor. For some data types, deposition in a public repository is mandatory - more information on our data deposition policies and available repositories can be found at <https://www.nature.com/nature-research/editorial-policies/reporting-standards#availability-of-data>.

Please include a data availability statement as a separate section after Methods but before references, under the heading "Data Availability". This section should inform readers about the availability of the data used to support the conclusions of your study. This information includes accession codes to public repositories (data banks for protein, DNA or RNA sequences, microarray, proteomics data etc...), references to source data published alongside the paper, unique identifiers such as URLs to data repository entries, or data set DOIs, and any other statement about data availability. At a minimum, you should include the following statement: "The data that support the findings of this study are available from the corresponding author upon request", mentioning any restrictions on availability. If DOIs are provided, we also strongly encourage including these in the Reference list (authors, title, publisher (repository name), identifier, year). For more guidance on how to write this section please see:

<http://www.nature.com/authors/policies/data/data-availability-statements-data-citations.pdf>

To improve the accessibility of your paper to readers from other research areas, please pay particular attention to the wording of the paper's opening bold paragraph, which serves both as an introduction and as a brief, non-technical summary in about 150 words. If, however, you require one or two extra sentences to explain your work clearly, please include them even if the paragraph is over-length as a result. The opening paragraph should not contain references. Because scientists from other sub-disciplines will be interested in your results and their implications, it is important to explain essential but specialised terms concisely. We suggest you show your summary paragraph to colleagues in other fields to uncover any problematic concepts.

If your paper is accepted for publication, we will edit your display items electronically so they conform to our house style and will reproduce clearly in print. If necessary, we will re-size figures to fit single or double column width. If your figures contain several parts, the parts should form a neat rectangle when assembled. Choosing the right electronic format at this stage will speed up the processing of your paper and give the best possible results in print. We would like the figures to be supplied as vector files - EPS, PDF, AI or postscript (PS) file formats (not raster or bitmap files), preferably generated with vector-graphics software (Adobe Illustrator for example). Please try to ensure that all figures are non-flattened and fully editable. All images should be at least 300 dpi resolution (when figures are scaled to approximately the size that they are to be printed at) and in RGB colour format. Please do not submit Jpeg or flattened TIFF files. Please see also 'Guidelines for Electronic Submission of Figures' at the end of this letter for further detail.

Figure legends must provide a brief description of the figure and the symbols used, within 350 words, including definitions of any error bars employed in the figures.

When submitting the revised version of your manuscript, please pay close attention to our [href="https://www.nature.com/nature-research/editorial-policies/image-integrity">Digital Image Integrity Guidelines.](https://www.nature.com/nature-research/editorial-policies/image-integrity) and to the following points below:

Please include a statement before the acknowledgements naming the author to whom correspondence and requests for materials should be addressed.

Finally, we require authors to include a statement of their individual contributions to the paper -- such as experimental work, project planning, data analysis, etc. -- immediately after the acknowledgements. The statement should be short, and refer to authors by their initials. For details please see the Authorship section of our joint Editorial policies at http://www.nature.com/authors/editorial_policies/authorship.html

- * include a point-by-point response to any editorial suggestions and to our referees. Please include your response to the editorial suggestions in your cover letter, and please upload your response to the referees as a separate document.

- * ensure it complies with our format requirements for Letters as set out in our guide to authors at www.nature.com/nmicrobiol/info/gta/

- * state in a cover note the length of the text, methods and legends; the number of references; number and estimated final size of figures and tables

- * resubmit electronically if possible using the link below to access your home page:
{redacted}

- *This url links to your confidential homepage and associated information about manuscripts you may have submitted or be reviewing for us. If you wish to forward this e-mail to co-authors, please delete this link to your homepage first.

Please ensure that all correspondence is marked with your Nature Microbiology reference number in the subject line.

Nature Microbiology is committed to improving transparency in authorship. As part of our efforts in this direction, we are now requesting that all authors identified as 'corresponding author' on published papers create and link their Open Researcher and Contributor Identifier (ORCID) with their account on the Manuscript Tracking System (MTS), prior to acceptance. This applies to primary research papers only. ORCID helps the scientific community achieve unambiguous attribution of all scholarly contributions. You can create and link your ORCID from the home page of the MTS by clicking on 'Modify my Springer Nature account'. For more information please visit www.springernature.com/orcid.

We hope to receive your revised paper within three weeks. If you cannot send it within this time, please let us know.

Yours sincerely,

{redacted}

Reviewer Expertise:

Referee #1: Marine cyanobacteria, cyanophage ecology

Referee #2: Phage identification and quantification methods/polony method

Referee #3: Mathematical Modelling

Referee #4: Mathematical Modelling, marine microbiology

Reviewers Comments:

Reviewer #1 (Remarks to the Author):

This is a re-review of "Enhanced infection of picocyanobacteria in a virus hotspot between ocean gyres" (NMICROBIOL-21061597B). Overall the manuscript has improved and several areas of concern have been addressed. Indeed the response is very long! However, some areas were not addressed perhaps from misunderstandings. For example, while there are plots that include environmental variables (in the supplemental), it's not clear why the revision does not show the variables vs. temperature like it does in the main text. Or at least have all of the variables plotted against Latitude. In its present form it is very frustrating for this reader to not be able to make visual comparisons between the different parameters measured. A similar presentation challenge is the different scales on the x-axis (temperature) between figure 2 and figure 3, yet width of the 'shaded region showing the virus hot spot' is the same (and thus differentially defined). I appreciate that the manuscript now includes plots of other variables, including mixed layer depth. There is low data density for low cell concentrations, namely for Pro, but for Syn I would argue there is a pattern that low MLD are associated with higher Syn concentrations. As before (and below), I don't think this detracts from the message that viruses are more abundant at these frontal / gyre transition zones, but it does present alternative mechanisms for the decline in cyanobacteria.

Some other comments were misinterpreted (perhaps my fault for not explaining in detail) – in particular the effect of the 'blob' in 2017. The point was not that the waters were warmer and that temperature was proximal, but that the waters were different in many ways from out years (e.g. little Syn in waters <13C, overall lower abundances in Pro, etc.). It's notable (to me) that the viral abundances are much more consistent across years (fig 2 and 3), but that the % infectivity is much higher in 2017 perhaps indicating that it's a numerator (i.e. host) change. Just a competing hypothesis, not necessarily the only one.

Those comments aside, overall the revised version has toned down the cause/effect arguments, which dramatically helps the manuscript. Nevertheless, while I think the data show that there can be an area of enhanced viral abundance and % infected cyanobacteria, it less clear that the viruses are driving to the decline. I think fig S2 pretty clearly shows this concept – there is no spatial relationship between cyanophage and cell concentrations. My guess is that if the manuscript directly plotted cyanophage (or various forms T4, etc.) vs. Pro or Syn concentrations, there would be no correlation. (If I'm wrong, I would strongly encourage that plot included!). The point here is that the manuscript acknowledges there are a host of factors that play into these patterns and that the drivers are not simple. But it also makes a case that cyanophage deserve full consideration as a driver.

Small comment, the title is much better, but I preferred 'gyre boundaries' to 'between ocean gyres'

Reviewer #2 (Remarks to the Author):

I have carefully gone over the responses to reviews and the revisions, and found that the authors have done a very good job of responding. While I still have some mild skepticism about a few issues I had raised, I am satisfied with the revisions and the version of the manuscript as is now, including the additional caveats made in responses to reviews. So I now support publication in this journal. I think that in multiple instances the authors had in mind a great deal of background information when they analyzed their data and wrote it up, yet they presented or cited only a fraction of that, resulting in someone like myself (not so fully versed in all that) feel that some aspects were not fully explained. That is better now, but I still think readers in the specialty would benefit from the more detailed explanations as written in responses to reviews (of course keeping in mind space limitation in the main paper). For example, the authors clearly explained the evidence on the coverage of the polony method, which was based on multiple papers as well as some new analyses. Readers would have to make quite an effort to themselves read and synthesize the multiple cited papers plus new data, so I suggest that perhaps a version of that fuller explanation be included in supplementary material. Though what is now in suppl is better than before, a clear exposition like this would be very welcome.

Reviewer #3 (Remarks to the Author):

I would like to thank the authors for taking my comments into account. I still think a predictive model including the interaction between viruses and bacterias would be better suited --- in particular because this interaction is the main focus of the paper. At the same time, I do understand how this is today apparently not possible, and how the linear regression against temperature and chlorophyll (as a proxy for cyanobacteria) can provide qualitative results about virus abundances.

Thank you for your time.

Best regards

Gianluca Meneghello

Reviewer #4 (Remarks to the Author):

Summary

The authors present a comprehensive assessment of picocyanobacterial and phage dynamics along transects in the North Pacific in three separate years. They also report a large suite of auxiliary oceanographic variables (salinity, temperature, nutrient concentration, etc.) for all three years. They ask which factors may drive declines in *Prochlorococcus* abundance that occur at high-latitude. To assess whether abiotic factors, most notably temperature, drive the decline, they make use of prior knowledge of *Prochlorococcus* growth sensitivity to a suite of environmental variables, which is now extremely well characterized. In two years, 2015 and 2016, they find that the decline in *Prochlorococcus* at high latitude is indeed consistent with thermal sensitivity of Pro to cold water. In 2017 however, there is a large region with low Pro density where Pro are expected to grow well, suggesting mortality is likely to play a role. The authors present an enumeration of cyanophages in the region, along with calculations of rates of infectivity, and infer that declines in Pro mortality could be due to viral infection.

Main comments

This is a highly commendable piece of work; the data is at the cutting edge of anything I have seen in this area. I also commend the thorough reviews and particularly the comments of reviewer #1 urging these authors to consider alternative explanations. I feel Carlson and co. have addressed these comments thoroughly. This, along with the addition of the metagenomic comparisons referenced, provides an impressively rigorous assessment of host-virus dynamics along a large ocean range.

I feel this work will be well received by the community as it will undoubtedly help to spur new avenues of research. I do have a few points of confusion about a few of the conclusions. I hope that these will be interpreted by the authors and the editors as opportunities for further clarification, or potentially to point toward future interesting lines of inquiry.

Issue 1: reconciling phage titers with Pro abundances and infection kinetics. In Figure 2c, the cyanophage abundance appears, to first order, to be strikingly similar across all three years. The authors note in line 237 that there is ~25% increase in cyanophages in 2017. However, this difference pales in comparison to the drop in Pro from $\sim 10^5$ to $\sim 10^3$ within the transition zone between 2015/2016 vs. 2017. I'm struggling to reconcile this with the claim that cyanophage drive significant mortality in 2017, but not in 2015 or 2016. If the infection kinetics were similar across all three transects, wouldn't one expect the increase in phage titer in 2017 vs. 2015/2016 to be commensurate with the drop in host abundance between years? I can think of a few ways to reconcile this apparent mismatch, e.g. by invoking enhanced removal of cyanophages in 2017 and/or differences in infection rates due to differences in picocyanobacterial and cyanophage genotype between years. I apologize if this was discussed, and I missed it. If it wasn't discussed, could the authors comment on this?

Issue 2: Can the authors comment on potential limitations of assuming 'average' values of infection parameters? Here and in Mruwat et al., 2020, the authors take average representative values for infection parameters such as the latent period and the burst size (Table S1 of this study and main text of Mruwat et al., 2020). In fact, there is striking variability in these numbers. For example, some T7-like phage have astonishingly short latent periods (~ 1 -3 hours, Table S4 of Mruwat et al., 2020), and some T4-like phage have much longer latent periods (~ 12 -15 hours, Table S4 of Mruwat et al., 2020). This represents an order of magnitude range in latent period, and there are similar ranges for other parameters such as burst size and per-cell rates of viral production. I worry that both here and in the Mruwat study, no attempt was made to assess uncertainty in these parameters, and the impact this may have on predictions of infection % and rates of virus induced mortality. I wish to be clear, that I personally see this issue as an exciting avenue of further inquiry, e.g. by working to understand at higher resolution which parameters from within this range are most likely, based upon the genetic composition of hosts and viruses in a given region.

Taken together, I will admit that issues #1 and #2 leave me with a slightly shaky understanding of the governing processes driving Pro dynamics along these transects. Personally, I see this as interesting and worthy of interrogation by the broader community. However, the authors may wish to put some thought into discussing these issues in the current manuscript, to invite readers to continue this interesting work.

Minor comments

Line 173-174: "Macronutrient ... optimal growth" the sentence references phosphate and nitrite+nitrate ranges for optimal growth, but refs 27 and 35 are only for iron. Can the authors either edit this to remove reference to macronutrients, or provide more relevant citations?

Lines 632-638: "Our modeling ... marine ecosystems" I'm not sure I agree that this regression model "provides a framework for incorporating viruses into ecosystem models" as per the discussion with reviewer #3, regression models and population models are very different. There are a whole bunch of

additional issues with ecosystem models, such as quantification of rates of virus adsorption and infection, rates of viral decay, etc. that are not considered here. I suggest toning down this claim.

Figure 1 caption: I don't get what is meant by "color bars at the top" I only see one colorbar showing chlorophyll false color scale. Can the authors clarify?

Figure 2 caption: It does not say what the gray shaded regions are. I see from other captions it marks the hot-spot, but I suggest saying that explicitly here

Author Rebuttal, second revision:

Reviewers Comments:

Reviewer #1 (Remarks to the Author):

This is a re-review of "Enhanced infection of picocyanobacteria in a virus hotspot between ocean gyres" (NMICROBIOL-21061597B). Overall the manuscript has improved and several areas of concern have been addressed. Indeed the response is very long! However, some areas were not addressed perhaps from misunderstandings. For example, while there are plots that include environmental variables (in the supplemental), it's not clear why the revision does not show the variables vs. temperature like it does in the main text. Or at least have all of the variables plotted against Latitude. In its present form it is very frustrating for this reader to not be able to make visual comparisons between the different parameters measured. A similar presentation challenge is the different scales on the x-axis (temperature) between figure 2 and figure 3, yet width of the 'shaded region showing the virus hot spot' is the same (and thus differentially defined).

Thank you for this clarification as we had not understood that this was the request initially. We have now added additional plots to the supplementary figures so that all the data is now presented both as a function of temperature and of latitude.

-In Fig S1 all data are now also plotted against temperature.

-In Fig S7 the data are now also plotted against latitude.

-In Fig S8 the data are now also plotted against temperature. (Note that in preparing this figure, we found a small error in part c which we have now fixed.)

-In Fig S11 all data are now also plotted against temperature.

We have also added 2 new supplemental figures. Fig. S12 shows abiotic parameters in Fig 1 plotted against temperature and Fig. S13 shows cyanophage lineages in Fig 3 plotted against latitude. These latter two figures are referred to from the figure legend of Fig. 1 and Fig. 3, respectively.

In addition, we have adjusted the x-axis scale on Fig. 3 to be the same as Fig. 2 as suggested by the reviewer. We would like to point out, however, that the width of the hotspot in both figures was defined in the same way as stated in the methods and covered the range from 14.7 to 18.4 °C.

I appreciate that the manuscript now includes plots of other variables, including mixed layer depth. There is low data density for low cell concentrations, namely for Pro, but for Syn I would argue there is a pattern that low MLD are associated with higher Syn concentrations. As before (and below), I don't think this detracts from the message that viruses are more abundant at these frontal / gyre transition zones, but it does present alternative mechanisms for the decline in cyanobacteria.

We agree with the reviewer that additional mechanisms, such as limitation by environmental parameters, change in mixed layer depth (MLD), and competition with picoeukaryotes can also play a role in the decline in *Synechococcus*. We now more explicitly mention MLD as a possible factor in the decline of picocyanobacteria in a new paragraph on the controls of *Prochlorococcus*' distribution in the supplement which we added in response to a comment by reviewer 2. We have also modified the concluding sentence on lines 130-131 to add a caveat that alternative mechanisms that we are unaware of may exist. We want to reiterate that we fully agree that multiple factors contributed to the patterns observed here (see lines 255-257, 267-271) and that it is the combined effect of viruses and environment that caused the decline in 2017. This has now been emphasized and stated more explicitly by revising the wording, particularly in the conclusions. (Note updated text below is marked in purple).

Lines 128-131:

“Thus, none of the physical or chemical factors investigated here can alone explain the unexpected decline in *Prochlorococcus* in 2017. However, we cannot rule out that a unique combination of these factors, or additional abiotic factors, led to the decline in *Prochlorococcus*.”

Lines 374-380:

This hotspot is superimposed on gradients in abiotic conditions, and together they influence important processes that shape the ecological succession of major marine primary producers and the cycling of organic matter in this region. The formation of the hotspot and the variation within was likely a result of distinct combinations of environmental conditions that ensued at different times, potentially having differential effects on virus diversity, infectivity and production as well as on host diversity and susceptibility to co-occurring viruses.

Supplement lines 941-942:

" Temperature is the widely accepted parameter that controls *Prochlorococcus* distributions^{11-13,15,90} with nutrient limitation^{4,16,34,91} and mixed layer depth^{14,92} implicated as well."

Supplement lines 958-964:

"The northernmost decline in *Synechococcus* corresponded to the region where picoeukaryotes reached $\sim 2 \times 10^4$ cells·ml⁻¹ in all three cruises (Fig. S2), suggesting that competition with picoeukaryotes may contribute to this decline and the spatial succession between these phytoplankton. Although none of the environmental parameters measured correlated with the decline in *Synechococcus* in 2017, factors such as mixed layer depth or as of yet unknown abiotic conditions may also be important in the decline in *Synechococcus*."

Some other comments were misinterpreted (perhaps my fault for not explaining in detail) – in particular the effect of the 'blob' in 2017. The point was not that the waters were warmer and that temperature was proximal, but that the waters were different in many ways from out years (e.g. little Syn in waters <13C, overall lower abundances in Pro, etc.). It's notable (to me) that the viral abundances are much more consistent across years (fig 2 and 3), but that the % infectivity is much higher in 2017 perhaps indicating that it's a numerator (i.e. host) change. Just a competing hypothesis, not necessarily the only one.

We thank the reviewer for clarifying this point. As we mentioned above, we fully agree with the reviewer that abiotic conditions and potentially host and virus diversity and interactions were different in 2017 and have stated this in the paragraphs on lines 226, 251, 273, and 294. We are unsure if the reviewer means % infectivity or % infection in their comment since we did not measure % infectivity. Changes in infectivity or increased host susceptibility due to changes in population structure, could lead to higher infection as mentioned on lines 255-257, 259-263, 294-296, 301-302. We have now added an additional clarification sentence to the conclusion which serves to summarize these in the one place to provide a plausible and coherent hypothesis. See revisions shown above for lines 377-380.

Indeed, % infection was much higher in 2017 across the whole transect and more total picocyanobacteria cells were infected in the hotspot in 2017 (Fig. S8). While we agree with the reviewer that the trends of increased cyanophage abundances are consistent between years, we consider the difference between the years to be quite significant. The 25% increase in cyanophage abundances in 2017 is an additional half a million cyanophages above those in 2016, 3-fold the number of *Prochlorococcus* in 2017 just north of their decline. We propose that this increase superimposed on a smaller picocyanobacterial population size brought the system to a tipping point and was a major factor in the observed decline. We now point this cyanophage increase out in numbers and also discuss now more clearly in the manuscript that it is likely that additional factors also contributed to the collapse of *Prochlorococcus*.

Those comments aside, overall the revised version has toned down the cause/effect arguments, which dramatically helps the manuscript. Nevertheless, while I think the data show that there can be an area of enhanced viral abundance and % infected cyanobacteria, it is less clear that the viruses are driving the decline. I think fig S2 pretty clearly shows this concept – there is no spatial relationship between cyanophage and cell concentrations. My guess is that if the manuscript directly plotted cyanophage (or various forms T4, etc.) vs. Pro or Syn concentrations, there would be no correlation. (If I'm wrong, I would strongly encourage that plot included!). The point here is that the manuscript acknowledges there are a host of factors that play into these patterns and that the drivers are not simple. But it also makes a case that cyanophage deserve full consideration as a driver.

We agree that the relationship between the abundances of cells and viruses is often difficult to interpret due to non-linear interactions between hosts, parasites, and the environment. This is because parasites such as lytic viruses are both dependent on their hosts and detrimentally affect their abundance and both are affected by environmental conditions. Indeed, the linear regression between total cyanobacteria and total cyanophages across all regimes is not significant (see below figure). However, analysis of the data for the hotspot shows a significant anticorrelation between cyanophage and cyanobacterial abundances across all three cruises (Pearson's $r = -0.56$, $p\text{-value} = 0.0005$) and even more so for the 2017 cruise (Pearson's $r = -0.65$, $p\text{-value} = 0.004$). As mentioned above, we hypothesize that the key difference in 2017 was a combination of high virus and low picocyanobacteria abundances that lead to a tipping point and the *Prochlorococcus* decline. As suggested by the reviewer we have added the below figure and a discussion on these points. We thank the reviewer for the suggestion of making this figure. This data together with the measurements of % infection, provide a strong basis for the role of viruses in the decline in this transition zone in 2017 but not in the other regimes. Nevertheless, we fully agree with the reviewer that this finding does not exclude other factors as important contributors to mortality in addition. As mentioned and shown above we have made changes that more explicitly discuss the combined role of cyanophage and abiotic conditions in the conclusions section of the manuscript and trust we have found the right balance in reporting the role of virus-induced mortality in combination with environmental factors (see lines 41, 128-131, 255-257, 267-271, 374-380, Supplement lines 941-942, 958-964 noted and shown above).

Figure 2d. The relationship between total picocyanobacteria and cyanophages. There was no relationship across data from all regimes (Pearson's $r = -0.008$, two-sided p -value = 0.9, $n = 87$). Picocyanobacteria were positively correlated with cyanophages in the subtropics (triangles) (Pearson's $r = 0.54$, two-sided p -value = 0.02, $n = 26$). In the hotspot (gray shaded dots), picocyanobacteria abundances were anticorrelated with cyanophages across all three cruises (Pearson's $r = -0.56$, two-sided p -value = 0.0005, $n = 34$). There was no relationship found in subpolar region (diamonds) (Pearson's $r = 0.2$, two-sided p -value = 0.2, $n = 27$).

Lines 182-192:

“To begin assessing whether cyanophages negatively affected cyanobacterial populations in the hotspot we tested the relationship between cyanophage and total cyanobacteria abundances. This showed a clear and significant anticorrelation between cyanophage and cyanobacterial abundances across all three cruises (Pearson's $r = -0.56$, two-sided p -value = 0.0005, $n = 34$). This relationship was particularly distinct in 2017 when cyanobacteria were at their overall lowest abundances and cyanophages at their highest (Pearson's $r = -0.65$, two-sided p -value = 0.004, $n = 18$). This suggests that viruses are one of the key regulators of picocyanobacteria in the region of the hotspot. However, no significant correlation was found across all regimes and all years (Pearson's $r = -0.008$, two-sided p -value = 0.9, $n = 87$) (Fig. 2d) indicating that factors other than viruses are likely more important in regulating cyanobacterial abundances in other regimes.”

Lines 194-201:

“Our single-cell infection measurements allowed us to directly evaluate active viral infection and its impact on picocyanobacteria in the transition zone. Viral infection spiked in this region each year with infection levels that were, on average, 2- to 9-fold higher than those in the subtropical gyre (Fig. 4, Fig. S5-S6, S8). Infection peaked within the temperature range of 12-18 °C and was associated with a concomitant dip in *Prochlorococcus* abundances in all three cruises (Fig. 4, Fig. S5). These findings provides independent support for the strong anticorrelation between cell and virus abundances (Fig. 2d) being the result of virus-induced mortality.”

Small comment, the title is much better, but I preferred 'gyre boundaries' to 'between ocean gyres'

The title has been modified accordingly.

Reviewer #2 (Remarks to the Author):

I have carefully gone over the responses to reviews and the revisions, and found that the authors have done a very good job of responding. While I still have some mild skepticism about a few issues I had raised, I am satisfied with the revisions and the version of the manuscript as is now, including the additional caveats made in responses to reviews. So I now support publication in this journal.

I think that in multiple instances the authors had in mind a great deal of background information when they analyzed their data and wrote it up, yet they presented or cited only a fraction of that, resulting in someone like myself (not so fully versed in all that) feel that some aspects were not fully explained. That is better now, but I still think readers in the specialty would benefit from the more detailed explanations as written in responses to reviews (of course keeping in mind space limitation in the main paper). For example, the authors clearly explained the evidence on the coverage of the polony method, which was based on multiple papers as well as some new analyses. Readers would have to make quite an effort to themselves read and synthesize the multiple cited papers plus new data, so I suggest that perhaps a version of that fuller explanation be included in supplementary material. Though what is now in suppl is better than before, a clear exposition like this would be very welcome.

We thank the reviewer for the positive remarks. As suggested by the reviewer, we have significantly expanded in the supplemental discussion on two topics that were discussed at length in the previous response to reviewers. One is an exposition on the polony method and its capacity to detect the cyanophage lineages we targeted as well as our perspective on the contribution of other and as yet unknown cyanophage types to overall mortality. The second is a more detailed summary of the body of literature that describes controls on *Prochlorococcus*' biogeography, particularly in the North Pacific Ocean, than appears in the main manuscript. We also discuss the reasons for the use of the chlorophyll front as a point of reference across cruises and clarify our use of the term decline relative to the expected abundance of *Prochlorococcus* based on its known temperature range. These sections can now be found in one place for readers in the supplementary material under the titles "**Controls on the biogeography of *Prochlorococcus***" (Supplemental discussion lines 940-952), "**Single-virus and single-cell infection quantification of the dominant cyanophage lineages**" (Supplemental discussion lines 969-1016). We thank the reviewer for this suggestion.

Reviewer #3 (Remarks to the Author):

I would like to thank the authors for taking my comments into account. I still think a predictive model including the interaction between viruses and bacterias would be better suited --- in particular because this interaction is the main focus of the paper. At the same time, I do understand how this is today apparently not possible, and how the linear regression against temperature and chlorophyll (as a proxy for cyanobacteria) can provide qualitative results about virus abundances.

Thank you for your time.

Best regards

Gianluca Meneghello

We thank the reviewer for his comments and look forward to creating population-based models using such data in the future. In light of this comment and comments by reviewer 4, we have modified our wording about the contribution of our model to population models.

Lines 380-386:

“Our modeling enabled predictions of viruses and their potential impact on picocyanobacterial distributions and biogeochemistry at a large geographic scale. Expansion of this model for other ocean regions, determination of population traits that lead to these ecosystem features, and the development of population models for cyanophages and other autotroph-virus systems, will allow us to gain a global view of the impacts of viruses on marine ecosystems in both present-day and future oceans^{11,12,14-16,30}.”

Reviewer #4 (Remarks to the Author):

Summary

The authors present a comprehensive assessment of picocyanobacterial and phage dynamics along transects in the North Pacific in three separate years. They also report a large suite of auxiliary oceanographic variables (salinity, temperature, nutrient concentration, etc.) for all three years. They ask which factors may drive declines in *Prochlorococcus* abundance that occur at high-latitude. To assess whether abiotic factors, most notably temperature, drive the decline, they make use of prior knowledge of *Prochlorococcus* growth sensitivity to a suite of environmental variables, which is now extremely well characterized. In two years, 2015 and 2016, they find that the decline in *Prochlorococcus* at high latitude is indeed consistent with thermal sensitivity of *Pro* to cold water. In 2017 however, there is a large region with low *Pro* density where *Pro* are expected to grow well, suggesting mortality is likely to play a role. The authors present an enumeration of cyanophages in the

region, along with calculations of rates of infectivity, and infer that declines in Pro mortality could be due to viral infection.

Main comments

This is a highly commendable piece of work; the data is at the cutting edge of anything I have seen in this area. I also commend the thorough reviews and particularly the comments of reviewer #1 urging these authors to consider alternative explanations. I feel Carlson and co. have addressed these comments thoroughly. This, along with the addition of the metagenomic comparisons referenced, provides an impressively rigorous assessment of host-virus dynamics along a large ocean range.

I feel this work will be well received by the community as it will undoubtedly help to spur new avenues of research.

We thank the reviewer for the positive assessment of the research presented in this manuscript.

I do have a few points of confusion about a few of the conclusions. I hope that these will be interpreted by the authors and the editors as opportunities for further clarification, or potentially to point toward future interesting lines of inquiry.

Issue 1: reconciling phage titers with Pro abundances and infection kinetics. In Figure 2c, the cyanophage abundance appears, to first order, to be strikingly similar across all three years. The authors note in line 237 that there is ~25% increase in cyanophages in 2017. However, this difference pales in comparison to the drop in Pro from $\sim 10^5$ to $\sim 10^3$ within the transition zone between 2015/2016 vs. 2017. I'm struggling to reconcile this with the claim that cyanophage drive significant mortality in 2017, but not in 2015 or 2016. If the infection kinetics were similar across all three transects, wouldn't one expect the increase in phage titer in 2017 vs. 2015/2016 to be commensurate with the drop in host abundance between years? I can think of a few ways to reconcile this apparent mismatch, e.g. by invoking enhanced removal of cyanophages in 2017 and/or differences in infection rates due to differences in picocyanobacterial and cyanophage genotype between years. I apologize if this was discussed, and I missed it. If it wasn't discussed, could the authors comment on this?

We see a significant anticorrelation between cyanophage and cyanobacterial abundances in the hotspot across all three years, based on new analysis suggested by reviewer #1 and now shown in Fig. 2d. Thus, it is possible that cyanophages are impacting cyanobacteria in 2016, especially as there is a small decline in *Prochlorococcus* abundances in the hotspot (Fig. 4c), but there are likely other, more significant factors, at play in determining the distribution of *Prochlorococcus* in 2016. Our reasons for not claiming that viruses drive significant mortality in 2016 are three-fold. First, the decline in *Prochlorococcus* in 2016 is commensurate with the

expected temperature limitation on their growth and geographic limits. Second, the decline in *Prochlorococcus* abundances occurs after the virus hotspot. Third, although % infection is elevated in the hotspot, it is not high enough to explain the decline based on our calculations of mortality from the observed % infection.

The increase in cyanophages in 2017, considering the decrease in picocyanobacteria, is within reasonable bounds for this system. For example, observing a 25% increase in virus abundances (~500,000 viruses/ml) compared to a decrease of ~32,000-77,000 *Prochlorococcus* cells/ml lost due to viruses based on our estimated mortality rates of 21%-51% mortality for ~150,000 cells/ml, suggests ~7-16 viruses produced per cell which encompasses the range of the burst sizes of 12 for the two T4-like cyanophages that infect *Prochlorococcus* for which the burst size has been determined (see Table S1). Perhaps our wording of “cyanophages were ~25% more abundant” does not sufficiently provide an understanding of the magnitude of the increase, especially since 500,000 cyanophages/ml is 3-fold more than the 150,000 *Prochlorococcus* cells/ml found just north of the decline in that year. Thus, we have now added the increase in cyanophages as absolute abundances in parentheses to this sentence for clarification.

We agree with the reviewer that our data suggest that some underlying factor(s) likely changed to either increase virus production or reduce cyanophage removal in 2017 relative to other years. We had indeed discussed a number of factors potentially important in the formation of the hotspot in the paragraph beginning on line 251 which include temperature-related decay, nutrient availability, host and virus diversity (lines 294-307). In response to this comment and a related comment by reviewer 1 we have also added a sentence to the manuscript that summarizes these possibilities both in the paragraph discussing the formation of the hotspot (beginning on line 251) and in the conclusion section. In addition, the reviewer raises an important point that we had not conveyed originally; that the transition zone conditions may vary from year to year and lead to differential impacts on cyanophage production and decay at different times. We are actively exploring mechanisms that influence cyanophage production and decay *in situ* because of such dynamics. We have now included this important point in the manuscript.

Lines 267-269:

The environmental factors influencing the production and removal of viruses are likely to vary in intensity at different times leading to variability in cyanophage abundances and infection levels.

Lines 377-380:

The formation of the hotspot and the variation within was likely a result of distinct combinations of environmental conditions that ensued at different times, potentially having differential effects on virus diversity, infectivity, and production as well as on host diversity and susceptibility to co-occurring viruses.

Issue 2: Can the authors comment on potential limitations of assuming 'average' values of infection parameters? Here and in Mruwat et al., 2020, the authors take average representative values for infection parameters such as the latent period and the burst size (Table S1 of this study and main text of Mruwat et al., 2020). In fact, there is striking variability in these numbers. For example, some T7-like phage have astonishingly short latent periods (~1-3 hours, Table S4 of Mruwat et al., 2020), and some T4-like phage have much longer latent periods (~12-15 hours, Table S4 of Mruwat et al., 2020). This represents an order of magnitude range in latent period, and there are similar ranges for other parameters such as burst size and per-cell rates of viral production. I worry that both here and in the Mruwat study, no attempt was made to assess uncertainty in these parameters, and the impact this may have on predictions of infection % and rates of virus induced mortality. I wish to be clear, that I personally see this issue as an exciting avenue of further inquiry, e.g. by working to understand at higher resolution which parameters from within this range are most likely, based upon the genetic composition of hosts and viruses in a given region.

We thank the reviewer for this pertinent comment. We agree that characterizing the variability in life history traits (such as burst size, latent periods, etc) and the impact of environmental conditions, is an important step in further refining estimates of cyanophage infection and mortality in natural systems. We are currently working towards this challenge and have characterized the traits of several phages belonging to the two sister clades of T7-like cyanophages (clades A and B). We found clade A phages have much shorter latent periods than clade B phages, as well as have higher burst sizes and are more virulent, and that these differences appear to differentially impact the ecology of the two clades (Maidanik et al. subm). This work is currently under review.

As the reviewer points out, we employ two calculations in this work. 1) calculating instantaneous % infection from iPolony infected cell measurements, and 2) estimating % mortality from this % infection data. Both calculations are carried out using parameters specific to the different cyanophage families and different host genera separately (i.e. T4-like cyanophages with *Prochlorococcus*, T4-likes with *Synechococcus*, etc.). As such the variability is 3-4-fold rather than 10-fold within each category of cyanophage family and host genus.

As per Mruwat & Carlson et al (2020), the calculation of % infection from iPolony data uses the relative proportion of the latent period, divided into three bins representing the periods prior to, during, and after phage genome replications which have different detection efficiencies (single phage genome copies prior to replication are detected less efficiently than multiple genome copies during and after genome replication inside cells). Although data are limited, the variability in the proportion of the latent period of these three stages is relatively minor. In response to this comment we tested the uncertainty of our estimates to these values by lengthening and shortening the duration of the bins by one standard deviation. This changes the infection values reported in this manuscript by an average of 0.0006% for T4-like cyanophages, for example.

Thus, the variability in the latent period appears to have a minimal impact on our calculation of % infection. More variability, would result, however, if cyanophage infections were highly synchronized and our measurements captured infections that were all prior to genome replication or all after genome replication. This would result in ~2x variability in infection values between the lower and upper bounds. Since this is quite significant, we do show these bounds of uncertainty in infection values as error bars both in Mruwat & Carlson et al. (2020) and in this manuscript in Fig. S5 and Fig. S6.

Conversion of instantaneous measurements of infection to mortality estimates relies on the accuracy of the number of infection cycles cyanophages can complete in a day, which is a function of the latent period. Indeed, there is variability in reported latent period values. We use the average because we assume that environmental populations do not skew towards cyanophages at either end of the latent period time spectrum. As mentioned above, the maximum difference for the length of the latent period within each virus-host group is 3-4-fold, rather than the 10-fold mentioned by the reviewer across all families and both host genera. If populations skewed towards one end, the most significant impact would be increased mortality estimates as the current average is closer to the longer end of the latent period distribution. This suggests that only if cyanophage community latent periods were highly skewed towards the longest latent periods, infections were highly synchronized in the field and we continuously captured them in the latest stages of infection, would our estimates represent a significant overestimation of mortality.

To make the reader more aware of the uncertainties in our estimates we have now added a section in the supplementary text describing the known variability in cyanophage latent periods, our assumption regarding the suitability of using average latent periods for environmental populations and the implications for our estimates of mortality to increase the understanding of our calculations and their uncertainties and to facilitate discussion on genotypic variation of latent periods as suggested by the reviewer. Please see discussions in the supplementary text on lines 1018-1047 in the section “**Estimating infection and mortality from iPolony measurements.**”

Taken together, I will admit that issues #1 and #2 leave me with a slightly shaky understanding of the governing processes driving Pro dynamics along these transects. Personally, I see this as interesting and worthy of interrogation by the broader community. However, the authors may wish to put some thought into discussing these issues in the current manuscript, to invite readers to continue this interesting work.

In summary, we have now significantly expanded the discussion of our methods and assumptions in calculating mortality in order to make uncertainties clearer to the reader (issue #2). Additionally, while we had previously discussed the topics mentioned in issue #1, we have added further discussion about how environmental conditions may impact virus decay and host

susceptibility. We further discussed the variability observed in the hotspot (lines. 267-269, 303-307, 377-380) and have added a figure and discussion to the main text that shows the negative relationship between viruses and cyanobacteria in the hotspot. This enhances and reinforces our inferences that viruses are important factors in the mortality of *Prochlorococcus* not just in 2017, but likely in all years, even though they did not cause an unexpected decline in these other years. We have further added more discussion on the potential reasons for the differences in *Prochlorococcus* decline in the different years. We thank the reviewer for these comments, which have improved the manuscript.

Minor comments

Line 173-174: “Macronutrient ... optimal growth” the sentence references phosphate and nitrite+nitrate ranges for optimal growth, but refs 27 and 35 are only for iron. Can the authors either edit this to remove reference to macronutrients, or provide more relevant citations?

This sentence has been reworded to be more accurate and to provide more suitable citations.

Lines 119-120:

“*Prochlorococcus* populations in 2017 were low compared to previous observations at similar macronutrient levels (phosphate and nitrite+nitrate, Fig. S3a,c)¹¹. Micronutrient (iron²⁷)...”

Lines 632-638: “Our modeling ... marine ecosystems” I’m not sure I agree that this regression model “provides a framework for incorporating viruses into ecosystem models” as per the discussion with reviewer #3, regression models and population models are very different. There are a whole bunch of additional issues with ecosystem models, such as quantification of rates of virus adsorption and infection, rates of viral decay, etc. that are not considered here. I suggest toning down this claim.

We have reworded this sentence to tone down this statement as suggested.

Lines 380-386:

“Our modeling enabled predictions of viruses and their potential impact on picocyanobacterial distributions and biogeochemistry at a large geographic scale. Expansion of this model for other ocean regions, determination of population traits that lead to these ecosystem features, and the development of population models for cyanophages and other autotroph-virus systems, will allow us to gain a global view of the impacts of viruses on marine ecosystems in both present-day and future oceans^{11,12,14-16,30}.”

Figure 1 caption: I don't get what is meant by "color bars at the top" I only see one colorbar showing chlorophyll false color scale. Can the authors clarify?

This was a remnant of a previous figure legend. This text has been removed.

Figure 2 caption: It does not say what the gray shaded regions are. I see from other captions it marks the hot-spot, but I suggest saying that explicitly here

This information has now been added to the Fig. 2 legend.

Decision Letter, third revision:

Our ref: NMICROBIOL-21061597C

11th January 2022

Dear Debbie,

Thank you for submitting your revised manuscript "Enhanced infection of picocyanobacteria in a virus hotspot at ocean gyre boundaries" (NMICROBIOL-21061597C). It has now been seen by one of the two original referees that still had some concerns, and their comments are below. The reviewers find that the paper has improved in revision, and therefore we'll be happy in principle to publish it in Nature Microbiology, pending minor revisions to satisfy the referees' and editors' final requests and to comply with our editorial and formatting guidelines.

Thank you again for your interest in Nature Microbiology Please do not hesitate to contact me if you have any questions.

Best wishes,
{redacted}

Reviewer #4 (Remarks to the Author):

Thank you for your clarifying remarks - I have nothing further to add

Decision Letter, final checks:

Dear Debbie,

Thank you for your patience as we've prepared the guidelines for final submission of your Nature Microbiology manuscript, "Enhanced infection of picocyanobacteria in a virus hotspot at ocean gyre boundaries" (NMICROBIOL-21061597C). Please carefully follow the step-by-step instructions provided in the attached file, and add a response in each row of the table to indicate the changes that you have made. Please also check and comment on any additional marked-up edits we have proposed within the text. Ensuring that each point is addressed will help to ensure that your revised manuscript can be swiftly handed over to our production team.

We would like to start working on your revised paper, with all of the requested files and forms, as soon as possible (preferably within a week). Please get in contact with us if you anticipate delays.

In recognition of the time and expertise our reviewers provide to Nature Microbiology's editorial process, we would like to formally acknowledge their contribution to the external peer review of your manuscript entitled "Enhanced infection of picocyanobacteria in a virus hotspot at ocean gyre boundaries". For those reviewers who give their assent, we will be publishing their names alongside the published article.

Nature Microbiology offers a Transparent Peer Review option for new original research manuscripts submitted after December 1st, 2019. As part of this initiative, we encourage our authors to support increased transparency into the peer review process by agreeing to have the reviewer comments, author rebuttal letters, and editorial decision letters published as a Supplementary item. When you submit your final files please clearly state in your cover letter whether or not you would like to participate in this initiative. Please note that failure to state your preference will result in delays in accepting your manuscript for publication.

Cover suggestions

As you prepare your final files we encourage you to consider whether you have any images or illustrations that may be appropriate for use on the cover of Nature Microbiology.

Nature Microbiology has now transitioned to a unified Rights Collection system which will allow our Author Services team to quickly and easily collect the rights and permissions required to publish your work. Approximately 10 days after your paper is formally accepted, you will receive an email in providing you with a link to complete the grant of rights. If your paper is eligible for Open Access, our Author Services team will also be in touch regarding any additional information that may be required to arrange payment for your article.

Please note that *Nature Microbiology* is a Transformative Journal (TJ). Authors may publish their research with us through the traditional subscription access route or make their paper immediately open access through payment of an article-processing charge (APC). Authors will not be required to make a final decision about access to their article until it has been accepted. [Find out more about Transformative Journals](https://www.springernature.com/gp/open-research/transformative-journals)

Authors may need to take specific actions to achieve [compliance](https://www.springernature.com/gp/open-research/funding/policy-compliance-faqs) with funder and institutional open access mandates. For submissions from January 2021, if your research is supported by a funder that requires immediate open access (e.g. according to [Plan S principles](https://www.springernature.com/gp/open-research/plan-s-compliance)) then you should select the gold OA route, and we will direct you to the compliant route where possible. For authors selecting the subscription publication route our standard licensing terms will need to be accepted, including our [self-archiving policies](https://www.springernature.com/gp/open-research/policies/journal-policies). Those standard licensing terms will supersede any other terms that the author or any third party may assert apply to any version of the manuscript.

Please use the following link for uploading these materials:
{redacted}

Final Decision Letter:

Dear Debbie,

I am pleased to accept your Article "Viruses affect picocyanobacterial abundance and biogeography in the North Pacific Ocean" for publication in Nature Microbiology. Thank you for having chosen to submit your work to us and many congratulations.

Acceptance of your manuscript is conditional on all authors' agreement with our publication policies (see <https://www.nature.com/nmicrobiol/editorial-policies>). In particular your manuscript must not be published elsewhere and there must be no announcement of the work to any media outlet until the publication date (the day on which it is uploaded onto our website).

Please note that *Nature Microbiology* is a Transformative Journal (TJ). Authors may publish their research with us through the traditional subscription access route or make their paper immediately open access through payment of an article-processing charge (APC). Authors will not be required to make a final decision about access to their article until it has been accepted. [Find out more about Transformative Journals](https://www.springernature.com/gp/open-research/transformative-journals)

Authors may need to take specific actions to achieve [compliance](https://www.springernature.com/gp/open-research/funding/policy-compliance-faqs) with funder and institutional open access mandates. For submissions from January 2021, if your research is supported by a funder that requires immediate open access (e.g. according to [Plan S principles](https://www.springernature.com/gp/open-research/plan-s-compliance)) then you should select the gold OA route, and we will direct you to the compliant route where possible. For authors selecting the subscription publication route our standard licensing terms will need to be accepted, including our [self-archiving policies](https://www.springernature.com/gp/open-research/policies/journal-policies). Those standard licensing terms will supersede any other terms that the author or any third party may assert apply to any version of the manuscript.

nature portfolio

Congratulations once again and I look forward to seeing the article published.